# COSA: CONTEXT-AWARE OUTPUT-SPACE ADAPTER FOR TEST-TIME ADAPTATION IN TIME SERIES FORE-CASTING

**Jeonghwan Im**
Department of Data Science
Seoul National University of Science and Technology
24520018@seoultech.ac.kr

**Hyuk-Yoon Kwon**[*]
Department of Data Science
Seoul National University of Science and Technology
hyukyoon.kwon@seoultech.ac.kr

## ABSTRACT

Deployed time-series forecasters suffer performance degradation under non-stationarity and distribution shifts. Test-time adaptation (TTA) for time-series forecasting differs from vision TTA because ground truth becomes observable shortly after prediction. Existing time-series TTA methods typically employ dual input/output adapters that indirectly modify data distributions, making their effect on the frozen model difficult to analyze. We introduce the *Context-aware Output-Space Adapter* (COSA), a minimal, plug-and-play adapter that directly corrects predictions of a frozen base model. COSA performs residual correction modulated by gating, utilizing the original prediction and a lightweight context vector that summarizes statistics from recently observed ground truth. At test time, only the adapter parameters (linear layer and gating) are updated under a leakage-free protocol, using observed ground truth with an adaptive learning rate schedule for faster adaptation. Across diverse scenarios, COSA demonstrates substantial performance gains versus baselines without TTA (13.91∼17.03%) and SOTA TTA methods (10.48∼13.05%), with particularly large improvements at long horizons, while adding a reasonable level of parameters and negligible computational overhead. The simplicity of COSA makes it architecture-agnostic and deployment-friendly. Source code: `https://github.com/bigbases/COSA_ICLR2026`

## 1 INTRODUCTION

Time-series forecasting serves as the foundation for critical decision-making across diverse domains, including finance (Chen et al., 2023), supply chain management (Aamer et al., 2020), energy grids (Di Piazza et al., 2021), and predictive maintenance (Makridis et al., 2020). Modern forecasting models, including Transformer-based architectures (Zhou et al., 2021; Liu et al., 2023; 2022), typically achieve high accuracy. However, they suffer performance degradation in real deployment settings due to non-stationarity and distribution shifts (Du et al., 2021; Chen et al., 2024a). Time series exhibit inherent non-stationarity, with changing temporal patterns and statistical characteristics over time, resulting in distributions at training that typically differ from those encountered after deployment.

To address this challenge, various approaches have been proposed, including online learning, continual learning, and domain adaptation. Online and continual learning methods adapt by updating model parameters directly to streaming data (Du et al., 2021; Zhang et al., 2024; Kirkpatrick et al., 2017; Rolnick et al., 2019; Giannini et al., 2023; Pham et al., 2022), but these approaches incur additional computational costs, memory requirements, catastrophic forgetting issues, and plasticity. Furthermore, these methods typically require labeled data or explicit knowledge of task boundaries, making them unsuitable for scenarios where only unlabeled streaming data is available during deployment. Domain adaptation methods learn robust representations by reducing source–target dis-

---

[*]Corresponding author: hyukyoon.kwon@seoultech.ac.kr

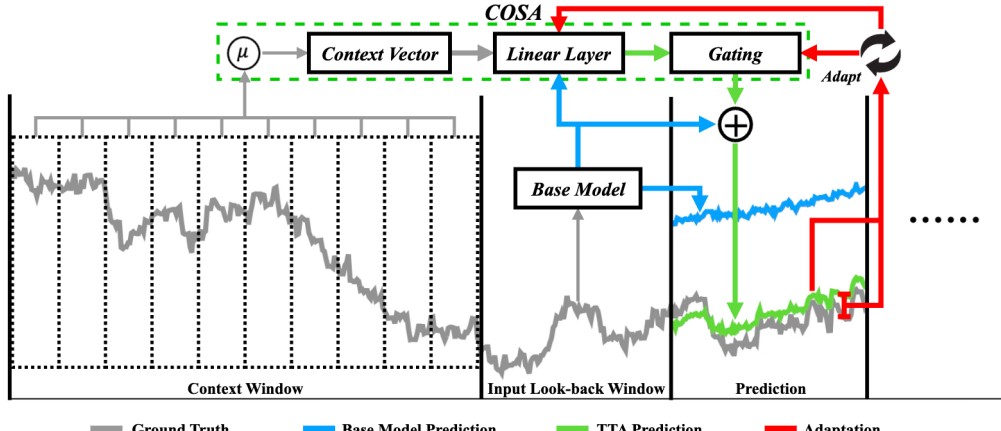

Figure 1: Overview of COSA operation showing the context-aware gated linear adapter architecture with input processing, linear transformation, gating mechanism, and output correction for test-time adaptation.

tribution differences (Wilson et al., 2020; Jin et al., 2022), but they rely on explicit target domain data and boundary definitions.

Test-time adaptation (TTA) offers an alternative approach that adapts to distribution changes by updating only lightweight modules using unlabeled test streams after deployment. TTA methods have evolved mainly in the vision domain through batch normalization coefficient optimization and entropy minimization (Wang et al., 2020), self-supervised/contrastive learning combined with pseudo-labeling (Liang et al., 2021; Chen et al., 2022; Gong et al., 2025), single-sample multi-augmentation-based adaptation (Zhang et al., 2022), and long-term adaptation stabilization (Wang et al., 2022).

Unlike vision tasks, time-series forecasting has unique characteristics that distinguish it from vision tasks: 1) it employs normalization methods different from vision tasks to preserve periodicity and level information, and 2) ground truth becomes sequentially observable after prediction with short delays, enabling the use of direct losses such as Mean Squared Error (MSE).

Time-series forecasting TTA is a recently evolving topic; to the best of our knowledge, only few methods (Kim et al., 2025; Medeiros et al., 2025; Grover & Etemad, 2025) have been proposed. All of them adopted dual-adapter architectures that place calibration modules at both input and output ends of the base model. They map inputs to domains that the base model can handle more easily and restore outputs to the original domain, controlling adaptation intensity through gating. However, these indirect distribution calibration methods involve design complexity and create uncertainty about the impact of input transformations on internal model representations.

To this end, we propose Context-aware Output-Space Adapter (COSA), which offers a direct output-space correction approach that operates with minimal computational overhead. Figure 1 presents the overview of COSA. COSA takes the predictions from a frozen base model and a lightweight context vector, summarizes recent observation statistics as input, computes residuals through linear correction, and controls correction strength using gating. At deployment, we freeze the base forecaster and update only a lightweight output adapter (i.e., linear correction with a learnable gate) under a leakage-free streaming protocol: after each prediction, adaptation uses only previously revealed ground truth, never current or future labels. COSA is architecture-agnostic and demonstrates consistent performance improvements over existing state-of-the-art time-series forecasting TTA methods across various predictors and horizons.

The main contributions of this study are summarized as follows:

1. **Architecture-agnostic output adapter.** Unlike existing time-series TTA methods that adopt dual input-output adapters, COSA consists of a single output adapter. COSA operates independently in the output space, correcting predictions from any base model without changes to training pipelines or internal parameters. COSA also shows compatibility with SOTA normalizers, consistently reducing prediction error.

2. **Context-aware linear residual with gating.** A linear correction uses the base prediction and a lightweight context vector that summarizes statistics of recent observed ground truth, and a learnable gate modulates correction strength.

3. **Consistent accuracy gains.** Across six benchmarks, four horizons, and six baseline architectures, COSA improves test MSE over baselines (13.91~17.03%) and SOTA TTA methods (10.48~13.05%), in particular, with the largest gains at longer horizons.

4. **Fast, efficient TTA.** Adaptive learning rate enables faster convergence of COSA, leading to higher accuracy within a few adaptation steps. Specifically, COSA enables 88.59~90.10% faster inference time against prior SOTA TTA methods.

## 2 RELATED WORK

### 2.1 TIME-SERIES FORECASTING

To handle non-stationarity in time-series forecasting, existing methods typically employ 1) online learning, 2) continual learning, and 3) domain adaptation. Representative online learning, D3A (Zhang et al., 2024) narrows source–target gaps through z-score monitoring of loss distributions and Gaussian noise injection, whereas Adarnn (Du et al., 2021) reduces temporal distribution shifts using temporal distribution characterization and distribution matching. In continual learning, cPNN (Giannini et al., 2023) grows temporal columns and transfers knowledge via lateral connections, and FSNet (Pham et al., 2022) separates per-layer adapters for rapid adaptation from associative memory for long-term retention to balance plasticity and stability. For domain adaptation, CoDATS (Wilson et al., 2020) learns domain-invariant features adversarially, and DAF (Jin et al., 2022) shares attention with domain-invariant queries/keys and domain-specific values. These families generally update the base model during training or online operation, differing from TTA, which adapts lightweight modules on unlabeled test streams while keeping the base model frozen.

### 2.2 TEST-TIME ADAPTATION

Tent (Wang et al., 2020) optimizes only batch-normalization affine parameters under entropy minimization, and SHOT (Liang et al., 2021) combines information maximization with self-supervised objectives to transfer source hypotheses to the target. AdaContrast (Chen et al., 2022) constructs pseudo-labels via contrastive learning with a dynamic memory bank for gradual adaptation, while MEMO (Zhang et al., 2022) applies multi-augmentation to a single test example to minimize marginal output entropy, updating all weights. CoTTA (Wang et al., 2022) limits error accumulation via weight and stochastic restoration, and ACCUP (Gong et al., 2025) integrates adaptive clustering with pseudo-labeling. However, they are proposed for vision tasks. TTA for time-series forecasting requires different approaches from those for vision tasks due to its own characteristics.

### 2.3 TEST-TIME ADAPTATION FOR TIME-SERIES FORECASTING

Time-series forecasting TTA methods typically employ dual adapters that calibrate distributions at both input and output. TAFAS (Kim et al., 2025) couples a calibration module to map inputs to a model-friendly domain and restores outputs to the original domain. It uses gating to modulate the calibration strength and utilizes Periodicity-Aware Adaptive Scheduling (PAAS) to adjust adaptation frequency using frequency patterns based on inputs. PETSA (Medeiros et al., 2025) factorizes the calibration module with a low-rank structure and adopts a combined loss for stable adaptation with fewer parameters. DynaTTA (Grover & Etemad, 2025) adjusts the dynamic learning rate, based on local distribution shift, global distribution shift, loss z-score. Existing time-series forecasting TTA methods employ an indirect approach that bidirectionally calibrates distributions at the input and output sides of the base model. They entailed design complexity due to indirect calibration and difficulty in predicting the impact of input transformations on internal representations. In contrast, we aim to utilize a single output-space adapter that directly corrects predictions without requiring input calibration or bidirectional transformation, resulting in a simpler design and more predictable adaptation behavior.

Table 1: Adapter-specific notation. Basic sizes/indices are defined as $(W, L, K, B; t, i, k)$.

| Symbol | Meaning (shape) |
|---|---|
| $\mathbf{Y}_t^{(0)}$ | Base (frozen) $L$-step prediction at time $t$ ($\mathbb{R}^L$). |
| $\mathbf{Y}_t^{\text{true}}$ | True $L$-step target revealed after $t$ ($\mathbb{R}^L$). |
| $\mathbf{C}_t$ | Context vector from revealed batch statistics $[\mu_t - K, \ldots, \mu_t - 1]^\top$ ($\mathbb{R}^K$). |
| $\mathbf{X}_t$ | Input look-back window ($\mathbb{R}^W$). |
| $\mathbf{X}_t^{(a)}$ | Adapter input $[\mathbf{Y}_t^{(0)} \,\|\, \mathbf{C}_t]$ ($\mathbb{R}^{L+K}$). |
| $\mathbf{H}_t$ | Linear residual $\boldsymbol{W} \mathbf{X}_t^{(a)} + \boldsymbol{b}$ ($\mathbb{R}^L$). |
| $\hat{\mathbf{Y}}_t$ | Corrected output $\mathbf{Y}_t^{(0)} + \alpha \mathbf{H}_t$ with $\alpha = \tanh(g) \in [-1, 1]$ ($\mathbb{R}^L$). |
| $\boldsymbol{W}, \boldsymbol{b}, \boldsymbol{g}$ | Adapter weights ($\mathbb{R}^L \times (L + K)$), bias ($\mathbb{R}^L$), and gate parameter ($\mathbb{R}$). |

*Operators:* concatenation $[\boldsymbol{a} \,\|\, \boldsymbol{b}]$; $\| \cdot \|_2$ vector norm; $\| \cdot \|_F$ Frobenius.

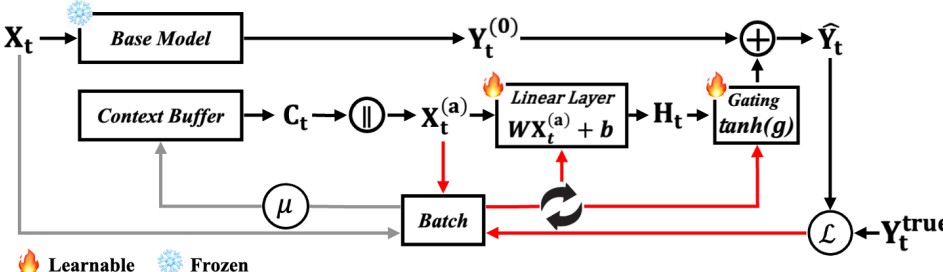

Figure 2: Detailed architecture of COSA illustrating the linear correction layer (weight matrix $\boldsymbol{W}$ and bias $\boldsymbol{b}$), learnable gating parameter ($\boldsymbol{g}$), and context vector ($\mathbf{C}$) integration for output-space correction.

## 3 COSA: CONTEXT-AWARE OUTPUT-SPACE ADAPTER

### 3.1 NOTATION AND PROBLEM FORMULATION

Table 1 shows the symbols necessary for COSA and their meanings.

This study targets univariate time-series forecasting, following the existing SOTA time-series forecasting TTA methods (Kim et al., 2025; Medeiros et al., 2025; Grover & Etemad, 2025). For multivariate time-series inputs, we decompose them into per-variable univariate forecasting tasks and perform the task iteratively for each variable. At time $t$, base model generates $L$-step original predictions $\mathbf{Y}_t^{(0)} \in \mathbb{R}^L$ from input $\mathbf{X}_t \in \mathbb{R}^W$, where $W$ denotes the input look-back window length. COSA generates corrected predictions $\hat{\mathbf{Y}}_t \in \mathbb{R}^L$ from input $\mathbf{X}_t^{(a)} \in \mathbb{R}^{L+K}$, where $K$ denotes the length of the context vector. After making predictions, the ground truth for that interval becomes sequentially observable following a short delay. Like other TTA approaches, we keep the base model completely frozen and perform only adapter adaptation at test time. Adaptation is performed by collecting the most recent $B$ prediction, ground truth pairs (batch index $i \in \{1, \ldots, B\}$ and context index $k \in \{1, \ldots, K\}$).

### 3.2 OVERALL ARCHITECTURE

Figure 2 illustrates the overall operation of COSA. COSA consists of a single output adapter that directly corrects the predictions. The key components are: 1) a linear layer composed of weight matrix $\boldsymbol{W}$ and bias variable $\boldsymbol{b}$ that computes correction values $\mathbf{H}$, 2) learnable gating $g$ that controls correction strength, and 3) a context vector $\mathbf{C}$ that summarizes and stores recent trend information.

We choose a single linear layer for two key reasons: 1) **Efficiency**: Linear operations provide lower latency and higher throughput compared to nonlinear modules, making them suitable for fast adaptation. We confirmed that a single-layer adapter shows 34.95% faster wall-clock time on average than a 2-layer MLP adapter. 2) **Simplicity-Performance balance**: As reported in LTSF-Linear (Zeng

et al., 2023), a linear layer sufficiently performs well in time-series forecasting, despite its simplicity. We also verified that a single linear layer adapter showed 5.71% even better performance on average against a 2-layer MLP adapter. These characteristics make the linear layer beneficial for TTA. Detailed results are provided in Appendix G.3.

The streaming protocol for leakage prevention is as follows (let the last adaptation was performed in $t-1$):

1. **Prediction:** At time $t$, base model generates prediction $\mathbf{Y}_t^{(0)}$ from input $\mathbf{X}_t$.

2. **Correction:** Feed $\mathbf{Y}_t^{(0)}$ and context $C_t$ into COSA to generate the corrected prediction $\hat{\mathbf{Y}}_t$.

3. **Observation:** After delay $\Delta \geq 0$, values of ground truth of the prediction horizon $\mathbf{Y}_t^{true}$ are sequentially observed.

4. **Adaptation:** Collect the most recent $B$ prediction, ground truth pairs $\{\hat{\mathbf{Y}}_{t+i-1}, \mathbf{Y}_{t+i-1}^{true}\}$, and perform adaptation that updates COSA parameters $\{\boldsymbol{W}, \boldsymbol{b}, \boldsymbol{g}\}$.

### 3.3 Output-space Residual Correction

For time $t$, we concatenate the original prediction of base model and context vector to create the adapter input:

$$\mathbf{X}_t^{(a)} = [\mathbf{Y}_t^{(0)} \parallel \mathbf{C}_t].$$

The residual is computed using a linear transformation:

$$\mathbf{H}_t = \boldsymbol{W} \mathbf{X}_t^{(a)} + \boldsymbol{b}.$$

The correction magnitude is controlled through gating to compose the final output:

$$\hat{\mathbf{Y}}_t = \mathbf{Y}_t^{(0)} + \tanh(\boldsymbol{g}) \mathbf{H}_t.$$

The $\tanh$ activation stabilizes the correction magnitude.

### 3.4 Context Construction

To prevent information leakage, the context summarizes previously observed ground truth information. For time $t$, we compute batch-wise aggregation as:

$$\mu_t = \mathsf{agg}\{ y_{t-(kB)+i}^{\text{true}} : 1 \leq i \leq B\}, \quad 1 \leq k \leq K.$$

where the aggregation function $\mathsf{agg}$ can use statistics such as mean, median, etc. We construct the context vector by stacking the most recent $K$ aggregated values:

$$\mathbf{C}_t = [\,\mu_1, \mu_2, \ldots, \mu_K\,]^\top.$$

This context vector summarizes level/scale changes and gradual drift patterns to help interpret the relative magnitude of the base prediction $\mathbf{Y}_t^{(0)}$ (reducing to single time-series values when $B=1$).

### 3.5 Adaptation Objective and Scheduling

Because targets arrive with a delay, we employ a direct objective with weight decay:

$$\mathcal{L} = \sum_{i=1}^{B} \big\|\big(\hat{\mathbf{Y}}_{t-i-1} - \mathbf{Y}_{t-i-1}^{\text{true}}\big)\big\|_2^2 + \lambda\big(\|\boldsymbol{W}\|_F^2 + \|\boldsymbol{b}\|_2^2 + \|\boldsymbol{g}\|_2^2\big). \tag{1}$$

When $B$ forecast–target pairs have been enqueued, we run $S$ gradient steps on the adapter parameters using a cosine–adaptive learning-rate schedule, simply *CALR*. We apply cosine annealing within the $S$ steps,

$$\eta^{(s+1)} = \eta_{\min} + \tfrac{1}{2}\big(\eta^{(s)} - \eta_{\min}\big)\Big(1 + \cos\frac{s\pi}{S}\Big). \tag{2}$$

and then adjust $\eta$ online, based on short-horizon loss trends to encourage fast but stable convergence (decrease $\eta$ on loss upticks; mildly increase on plateaus). When a new batch arrives, it is always

initialized with the same learning rate, and thereafter the learning rate for the next step within the batch is determined through Equation 2 according to the loss. Early stopping and gradient clipping are also implemented. The threshold values for learning rate adjustment are stability-induced by balancing adaptation speed against stability. Conservative thresholds ensure convergence while aggressive values enable faster response to distribution shifts. Full pseudocode and thresholds are given in Algorithm 1 in Appendix A.

COSA targets TSF-TTA under non-stationary environments in which the distribution of time-series data changes over time. In such environments, the classical notion of convergence toward a fixed optimal point is not well-defined. Instead, stable learning within each adaptation window is critical. CALR guarantees uniformly bounded step-wise updates through the following four mechanisms, which structurally prevent error amplification and thus ensure stability during adaptation.

1. **Upper-bounded learning rate**: The learning rate is constrained by $\eta \leq \eta_{\max}$, limiting the maximum magnitude of a single-step update.

2. **Gradient clipping**: At Line 18 of Algorithm 1, the gradient norm is adaptively bounded as $\|\mathbf{g}_\phi\| \leftarrow \min(\|\mathbf{g}_\phi\|, \max(c, \mathcal{L}))$.

3. **L2 regularization**: The weight-decay term in Equation 1, $\lambda(\|\mathbf{W}\|_F^2 + \|\mathbf{b}\|_2^2 + \|g\|_2^2)$, constrains parameter magnitude.

4. **Bounded gating**: Because $\alpha = \tanh(g) \in [-1, 1]$, the correction magnitude is structurally limited.

For every new batch, the learning rate is reinitialized to $\eta_{\max}$ (Line 2 of Algorithm 1), giving each batch an equal opportunity for adaptation. The learning rate is then adapted according to the batch's loss behavior. When the loss spikes, we reduce the learning rate as $\eta \leftarrow \max(0.5\eta, \eta_{\min})$, temporarily lowering update intensity. When the loss decreases stably, we increase it as $\eta \leftarrow \min(1.1\eta, \eta_{\max})$, strengthening adaptation. This enables stable learning even when short-term perturbations or anomalies appear in the input data, allowing rapid recovery.

## 4 EXPERIMENTS

### 4.1 EXPERIMENTAL SETTINGS

We evaluate COSA on six benchmark datasets (ETTh1/2, ETTm1/2, Exchange Rate, and Weather) with a fixed look-back window ($W = 96$) and four prediction horizons ($L \in \{96, 192, 336, 720\}$). We used six representative base models spanning different architectures: Transformer-based (iTransformer (Liu et al., 2023), PatchTST (Nie et al., 2023)), linear-based (DLinear (Zeng et al., 2023), OLS (Toner & Darlow, 2024)), and MLP-based (FreTS (Yi et al., 2023), MICN (Wang et al., 2023)). By default, all input time series are treated as variable-wise univariate forecasting tasks, standard normalization is applied, and MSE serves as the performance comparison metric.

We compare *COSA* (our method) with *Baseline* (without TTA), *TAFAS* (Kim et al., 2025), and *PETSA* (Medeiros et al., 2025). All experiments were conducted according to the official benchmark library (Wang et al., 2024) [1]. The train:valdiation:test ratio is 7:1:2 for all datasets.

Unless otherwise noted, we fix the adapter hyperparameters to $K=10$ and $S=3$, enabled $CALR$. We utilize the average as agg. Ablation studies for the variations of agg are provided in Appendix G.1.

We report two variants for COSA: COSA-F, which uses a fixed $B=48$ (half of the look-back), and COSA-P, which sets $B$ online following the PAAS in TAFAS (Kim et al., 2025). Hyperparameters for comparative methods follow the settings reported in the original papers or official code defaults. In tables, the best score is shown in bold and the second best is underlined.

We utilize Xavier uniform initializer (Glorot & Bengio, 2010) with gain $= 0.1$ for parameters of the weight matrix $\boldsymbol{W}$. The bias $\boldsymbol{b}$ and gating $\boldsymbol{g}$ are initialized to 0, and Adam optimizer is utilized.

---

[1]In the case of *DynaTTA* (Grover & Etemad, 2025), there were reproducibility issues when we used the officially released source code. Therefore, we report the comparison results with them in the Appendix F.5 with the used detailed hyperparameters.

Table 2: Prediction accuracy comparison. Standard deviations less than 0.001 are omitted.

| | | Transformer-based | | | | | Linear-based | | | | | MLP-based | | | | |
|---|---|---|---|---|---|---|---|---|---|---|---|---|---|---|---|---|
| | | iTransformer | | | | | DLinear | | | | | FreTS | | | | |
| | | Baseline | TAFAS | PETSA | COSA-F | COSA-P | Baseline | TAFAS | PETSA | COSA-F | COSA-P | Baseline | TAFAS | PETSA | COSA-F | COSA-P |
| ETTh1 | 96 | .4507 | .4411 | .4393 | .4368 | **.4363** | .4695 | .4618 | .4594 | .4574 | **.4482** | .4462 | .4403 | .4387 | .4384 | **.4371** |
| | 192 | .5078 | .4928 | .4949 | .4961 | **.4919** | .5213 | .5117 | .5118 | .5066 | **.5050** | .5022 | .4954 | .4942 | .4951 | **.4940** |
| | 336 | .5658 | .5629 | .5640 | .5651 | **.5300** | .5659 | .5604 | .5617 | .5528 | **.5456** | .5544 | .5521 | .5527 | **.5467** | .5351 |
| | 720 | .7038 | .6612 | .6596 | .5958 | **.5638** | .7117 | .6820 | .6743 | .6107 | **.5896** | .7182 | .6852 | .6846 | .6259 | **.5959** |
| ETTh2 | 96 | .2577 | .2549 | .2551 | .2504 | **.2493** | .2323 | .2303 | .2306 | .2300 | **.2281** | .2384 | .2367 | .2364 | .2367 | **.2350** |
| | 192 | .3161 | .3010 | .3006 | .2983 | **.2947** | .2862 | .2842 | .2876 | .2827 | **.2819** | .2866 | .2824 | .2832 | **.2816** | .2824 |
| | 336 | .3545 | .3352 | .3348 | .3241 | **.3339** | .3252 | .3185 | .3184 | **.3050** | .3083 | .3317 | .3229 | .3233 | **.3031** | .3153 |
| | 720 | .4276 | .4023 | .4043 | **.3487** | .3591 | .4087 | .3873 | .3853 | **.3062** | .3477 | .4119 | .3857 | .3860 | **.3169** | .3399 |
| ETTm1 | 96 | .3823 | .3558 | .3570 | **.3447** | .3455 | .3715 | .3497 | .3524 | **.3456** | .3475 | .3675 | .3582 | .3583 | **.3520** | *.3525* |
| | 192 | .4423 | .4146 | .4142 | **.4124** | .4140 | .4438 | .4166 | .4178 | **.4113** | .4122 | .4325 | .4212 | .4198 | **.4150** | .4212 |
| | 336 | .5093 | .4754 | .4751 | **.4569** | .4643 | .5183 | .4799 | .4803 | **.4753** | .4858 | .5005 | .4827 | .4789 | **.4661** | .4775 |
| | 720 | .6065 | .5562 | .5553 | **.4773** | .5102 | .5929 | .5488 | .5532 | **.4774** | .4991 | .5704 | .5486 | .5476 | **.4718** | .4982 |
| ETTm2 | 96 | .1647 | .1634 | .1637 | **.1627** | .1632 | .1598 | .1584 | .1584 | **.1583** | .1586 | .1581 | .1572 | .1572 | **.1568** | .1569 |
| | 192 | .2209 | .2183 | .2173 | **.2171** | .2173 | .1930 | .1913 | .1913 | **.1904** | .1905 | .1923 | .1909 | .1908 | **.1905** | .1908 |
| | 336 | .2727 | .2630 | .2592 | **.2435** | .2535 | .2324 | .2289 | .2292 | **.2083** | .2242 | .2320 | .2288 | .2289 | **.2098** | .2211 |
| | 720 | .3451 | .3305 | .3332 | **.2477** | .2606 | .3062 | .2968 | .2963 | **.2215** | .2316 | .3012 | .2916 | .2926 | **.2158** | .2314 |
| Exchange Rate | 96 | .0882 | .0876 | .0885 | **.0818** | .0837 | .0913 | .0885 | .0878 | **.0812** | .0834 | .0828 | .0799 | .0803 | **.0744** | .0766 |
| | 192 | .1811 | .1686 | .1740 | **.1403** | .1479 | .1827 | .1760 | .1730 | **.1459** | .1519 | .1734 | .1665 | .1648 | **.1366** | .1499 |
| | 336 | .3428 | .3079 | .3097 | **.2089** | .2624 | .3277 | .2941 | .2920 | **.2039** | .2480 | .3240 | .2930 | .2923 | **.2053** | .2461 |
| | 720 | .8540 | .8322 | .8004 | **.3421** | .4460 | .8873 | .8762 | .8781 | **.3494** | .4481 | .8368 | .8273 | .8067 | **.3352** | .4458 |
| Weather | 96 | .1755 | .1664 | .1674 | **.1597** | .1617 | .1954 | .1796 | .1823 | **.1773** | .1793 | .1856 | .1759 | .1765 | **.1724** | .1737 |
| | 192 | .2232 | .2101 | .2128 | **.2067** | .2088 | .2403 | .2244 | .2254 | **.2216** | .2217 | .2310 | .2165 | .2192 | **.2135** | .2189 |
| | 336 | .2800 | .2614 | .2665 | **.2503** | .2515 | .2918 | .2709 | .2740 | **.2567** | .2626 | .2843 | .2653 | .2681 | **.2561** | .2587 |
| | 720 | .3571 | .3458 | .3459 | **.2480** | .2730 | .3643 | .3500 | .3497 | **.2581** | .2708 | .3599 | .3490 | .3488 | **.2573** | .2692 |

| | | PatchTST | | | | | OLS | | | | | MICN | | | | |
|---|---|---|---|---|---|---|---|---|---|---|---|---|---|---|---|---|
| | | Baseline | TAFAS | PETSA | COSA-F | COSA-P | Baseline | TAFAS | PETSA | COSA-F | COSA-P | Baseline | TAFAS | PETSA | COSA-F | COSA-P |
| ETTh1 | 96 | .4312 | .4262 | .4269 | .4242 | **.4238** | .4511 | .4409 | .4391 | .4390 | **.4372** | .5103 | .4901 | .4898 | .4693 | **.4684** |
| | 192 | .4955 | .4865 | .4854 | .4830 | **.4805** | .5046 | .4934 | .4937 | .4915 | **.4906** | .5954 | .5617 | .5620 | .5372 | **.5328** |
| | 336 | .5559 | .5478 | .5475 | .5438 | **.5320** | .5510 | .5440 | .5465 | .5385 | **.5320** | .6615 | .6387 | .6420 | .5950 | **.5878** |
| | 720 | .7117 | .6860 | .6822 | .6113 | **.5822** | .6997 | .6630 | .6431 | .5969 | **.5733** | .9233 | .8142 | .8375 | .7001 | **.6504** |
| ETTh2 | 96 | .2362 | .2351 | .2362 | .2349 | **.2343** | .2306 | .2285 | .2288 | **.2232** | .2265 | .2582 | .2551 | .2552 | .2492 | **.2485** |
| | 192 | .2826 | .2758 | .2773 | .2665 | **.2608** | .2839 | .2824 | .2848 | .2796 | **.2791** | .3282 | .3179 | .3258 | .3049 | **.3017** |
| | 336 | .3199 | .3125 | .3132 | **.2971** | .2978 | .3258 | .3182 | .3189 | **.3003** | .3043 | .3732 | .3482 | .3497 | **.3241** | .3310 |
| | 720 | .4264 | .4005 | .4012 | **.3233** | .3428 | .4162 | .3908 | .3884 | **.3177** | .3453 | .4617 | .4474 | .4473 | **.3650** | .3885 |
| ETTm1 | 96 | .4024 | .3894 | .3937 | **.3625** | .3626 | .3710 | .3506 | .3536 | **.3454** | .3475 | .4354 | .3951 | .3951 | .3837 | **.3831** |
| | 192 | .4512 | .4372 | .4413 | **.4250** | .4258 | .4439 | .4160 | .4184 | **.4115** | .4119 | .4855 | .4566 | .4574 | **.4476** | .4514 |
| | 336 | .5081 | .4905 | .4946 | **.4568** | .4697 | .5182 | .4787 | .4792 | **.4748** | .4749 | .5556 | .5108 | .5082 | **.4832** | .5054 |
| | 720 | .5629 | .5427 | .5462 | **.4681** | .4882 | .5922 | .5478 | .5522 | **.4763** | .5007 | .6212 | .5756 | .5778 | **.5029** | .5225 |
| ETTm2 | 96 | .1584 | .1581 | .1583 | **.1558** | .1562 | .1602 | .1590 | .1589 | **.1582** | .1586 | .1710 | .1711 | .1730 | **.1702** | .1704 |
| | 192 | .2059 | .2036 | .2037 | **.2007** | .2022 | .1936 | .1921 | .1919 | **.1906** | .1907 | .2121 | .2102 | .2126 | **.2102** | .2120 |
| | 336 | .2458 | .2451 | .2452 | **.2258** | .2352 | .2331 | .2299 | .2302 | **.2131** | .2226 | .2530 | .2501 | .2520 | **.2337** | .2351 |
| | 720 | .3268 | .3268 | .3256 | **.2446** | .2645 | .3066 | .2986 | .2971 | **.2171** | .2349 | .3327 | .3220 | .3131 | **.2477** | .2643 |
| Exchange Rate | 96 | .0867 | .0843 | .0837 | **.0765** | .0788 | .0814 | .0792 | .0798 | **.0756** | .0773 | .1151 | .1087 | .1146 | **.0955** | .1008 |
| | 192 | .1877 | .1805 | .1832 | **.1464** | .1570 | .1727 | .1658 | .1653 | **.1393** | .1457 | .2150 | .2198 | .1999 | **.1663** | .1722 |
| | 336 | .3389 | .3275 | .3300 | **.1983** | .2445 | .3226 | .2877 | .2898 | **.2020** | .2323 | .3950 | .3047 | .3100 | **.2119** | .2660 |
| | 720 | .8648 | .8659 | .8643 | **.3543** | .4662 | .8366 | .8138 | .8149 | **.3444** | .4541 | 1.0259 | .7191 | .7805 | **.3871** | .4815 |
| Weather | 96 | .1742 | .1724 | .1743 | **.1624** | .1634 | .1957 | .1807 | .1795 | **.1772** | .1803 | .1757 | .1853 | .1970 | **.1636** | .1651 |
| | 192 | .2195 | .2147 | .2167 | **.2006** | .2108 | .2404 | .2244 | .2274 | **.2223** | .2237 | .2237 | .2161 | .2265 | **.2082** | .2120 |
| | 336 | .2766 | .2666 | .2701 | **.2451** | .2488 | .2921 | .2714 | .2748 | **.2551** | .2642 | .2812 | .2746 | .2788 | **.2729** | .2737 |
| | 720 | .3544 | .3383 | .3442 | **.2590** | .2713 | .3644 | .3466 | .3493 | **.2579** | .2708 | .3508 | .3573 | .3681 | **.2582** | .2855 |

All experiments were conducted on a machine with an Intel i7-7800X CPU and NVIDIA GeForce RTX 3080 10GB.

## 4.2 MAIN RESULTS

### 4.2.1 COMPARISON WITH SOTA TIME-SERIES TTA METHODS

The proposed COSA achieves best performance in all scenarios, as shown in Table 2 [2]. The results reveal two key performance patterns:

1. **Architecture-agnostic benefits:** Consistent improvements across all base models demonstrate that effectiveness of COSA is not dependent on specific model architectures. The average improvement ranges from 10.48% to 13.05%.

---

[2]All reported results are averaged over 10 runs with different random seeds to ensure statistical reliability.

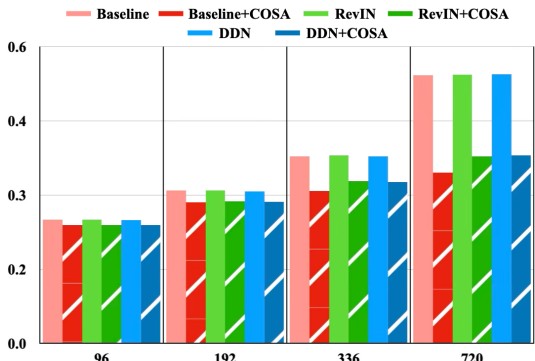

Figure 3: Prediction accuracy comparison with normalization methods.

2. **Effectiveness in long-term forecasting:** The largest performance improvements were observed at the 720 horizon, where COSA-F and COSA-P showed performance improvements of 32.24% and 26.33% compared to baseline, respectively, and 28.21% and 21.96% compared to other methods. This trend suggests that COSA becomes increasingly valuable for longer prediction horizons.

These findings demonstrate that the COSA, which performs residual correction directly in the output space, proves more effective than existing indirect dual-adapter approaches.

### 4.2.2 COMPARISON WITH NORMALIZATION METHODS

In this section, we analyze whether COSA serves as an alternative or complementary role to existing normalizations, and demonstrate that COSA can work independently of normalizations and is compatible with normalization mechanisms. Figure 3 shows performance comparison of COSA against two representative time-series normalizers, RevIN (Kim et al., 2021) and DDN (Dai et al., 2024). The results support two claims:

1. COSA on its own generally outperforms explicit normalization, as demonstrated by the comparative analysis showing that using COSA with basic normalization achieves the lowest mean MSE across all experimental settings.

2. COSA is compatible with normalization and consistently improves accuracy within the same normalizer settings, with the addition of COSA reducing MSE by approximately 16.8∼16.9%.

COSA directly optimizes MSE on revealed targets and uses the context vector to encode recent level/scale, so the linear layer learns scale and level corrections from the error signal itself. Further analysis is provided in Appendix G.7.

### 4.3 SENSITIVITY AND ABLATIONS

We probe the following four key design choices: 1) adaptation steps $S$, 2) context length $K$, 3) batch size $B$, and 4) adaptive learning rate $CALR$.

We evaluated performance by varying each hyperparameter individually while keeping others fixed at the default settings. Figure 4 shows MSE and wall-clock time according to each hyperparameter. Figure 4a shows changes according to the number of iterative learning steps $S$. COSA shows a pattern where test MSE decreases as $S$ increases. However, while wall-clock time also increases with increasing $S$, even at the highest $S = 4$ setting, it showed time levels similar to PETSA.

Figure 4b examines the effect of context length $K$, which controls how much past information the adapter uses. Accuracy improves consistently with larger $K$, while wall-clock time remains unchanged. Since the adapter input concatenates base model's prediction with the context vector (dimensionality $L + K$), and $L$ typically dominates, increasing $K$ has negligible runtime impact. The context provides incremental but reliable gains by supplementing the level/scale information.

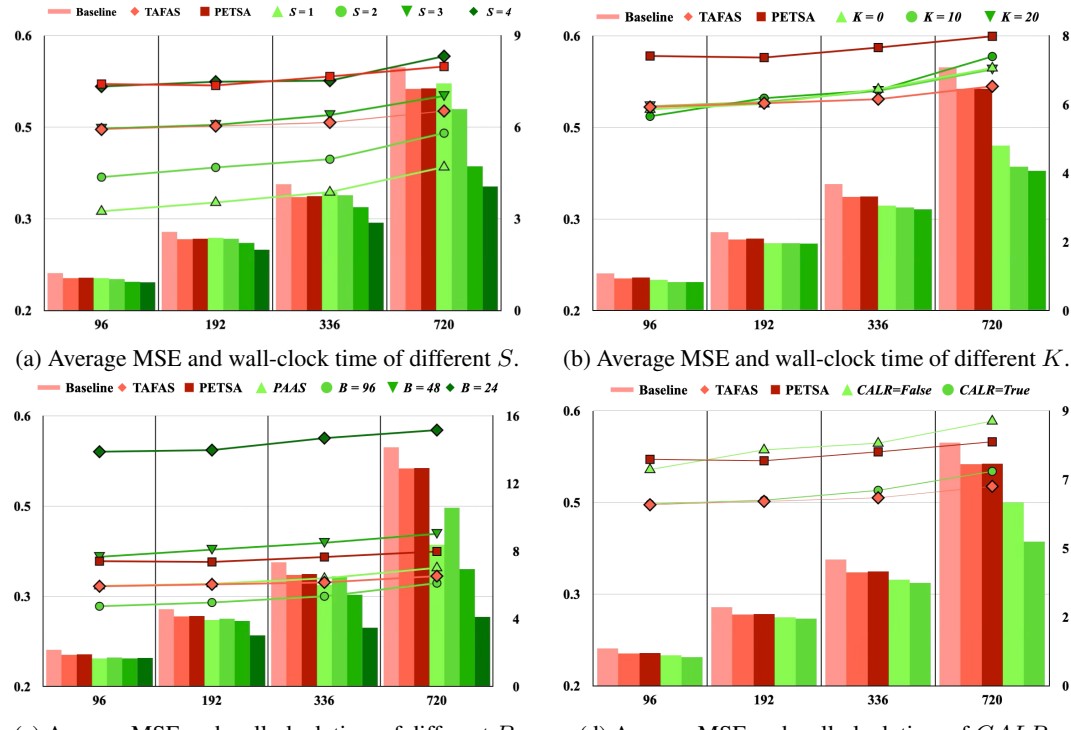

(a) Average MSE and wall-clock time of different $S$.

(b) Average MSE and wall-clock time of different $K$.

(c) Average MSE and wall-clock time of different $B$.

(d) Average MSE and wall-clock time of $CALR$.

Figure 4: Hyperparameter analysis showing the trade-off between performance and efficiency. Charts display average test MSE (bars, left axis) and wall-clock time (lines, right axis, in seconds) across different parameter settings.

Figure 4c shows performance changes with batch size $B$. As described in Section 3.5, $B$ determines the frequency of adaptation, collecting $B$ {prediction, ground truth} pairs before each update. Even with $B = 96$, COSA outperforms the baseline, with accuracy improving as $B$ decreases due to more frequent adaptation. This explains the superior performance of COSA-P over COSA-F on ETTh1: the average $B$ determined by PAAS for ETTh1 is 24.55, while for other datasets the values are over 80. Detailed analysis is provided in Appendix C. However, smaller $B$ increases wall-clock time due to both more adaptation calls and the computational cost of adaptation steps $S$.

Figure 4d shows performance with and without $CALR$ (Section 3.5). $CALR$ achieves up to 12.13% accuracy improvement as the window length increases and a 21.34% reduction in wall-clock time. This confirms that aggressive dynamic learning rate scheduling enhances performance within limited adaptation steps $S$ while enabling early-stopping for computational efficiency.

Results for additional ablations (context aggregation methods and correction layer architecture) are provided in Appendix G.

## 4.4 COMPUTATIONAL OVERHEAD

Table 3: Computational overhead comparison of adapter methods.

| Method | # Params ↓ | Peak mem (MB) ↓ | Samples/sec ↑ | Adaptation time/batch (ms) ↓ | Inference time/batch (ms) ↓ |
|---|---|---|---|---|---|
| TAFAS | 1,252,958 | **17.59** | **1,413.23 ± 92.28** | **73.23 ± 8.74** | 10.96 ± 0.64 |
| PETSA | **58,334** | 36.09 | 987.91 ± 78.67 | 88.46 ± 7.08 | 12.63 ± 0.37 |
| COSA (Ours) | 1,211,287 | 27.07 | 1080.70 ± 80.66 | 80.12 ± 5.93 | **1.25 ± 0.06** |

We report 1) observed additional parameters, 2) peak memory utilization, 3) throughput (samples per second), 4) wall-clock time per batch, and 5) inference time per batch.

All hyperparameters remain at default values. Table 3 summarizes the average overhead across all datasets, base model, and horizons. COSA shows moderate overhead, falling between TAFAS and PETSA, while achieving the significantly fastest inference time. COSA performs adaptation repeat-

edly for $S$ steps, meaning that the throughput and adaptation time are dependent on $S$. However, the single adapter structure and simplicity of COSA alleviate the overhead and improve the inference time, which is not affected by $S$. The computational complexity is $\mathcal{O}(L \cdot (L + K))$ for the linear transformation plus $\mathcal{O}(L)$ for the gating operation, resulting in quadratic scaling with respect to prediction horizon $L$. Further theoretical calculations are included in Appendix B.

## 5 CONCLUSION

We introduced COSA, an architecture-agnostic TTA module that directly corrects the prediction of base model with a single linear layer guided by short-term context and a stabilizing gate. Across six benchmarks and diverse base models, COSA improves accuracy by 13.91% to 17.03% over baselines and by 10.48% to 13.05% over prior state-of-the-art TTA methods. These gains arise from the synergy of trend-aware context, residual correction, and gated modulation.

While COSA shows strong empirical results, several areas offer room for refinement. The current adaptation relies on full ground truth, though extending to partial observations would enable real-time deployment. Performance varies with batch size $B$, and the fixed context length $K$ may not be optimal for all temporal patterns. Additionally, linear corrections, while effective for many cases, could be enhanced for complex nonlinear shifts.

Future work will explore masked updates for real-time adaptation with partial targets, adaptive selection of $K$ and $B$ based on detected periodicity, and hybrid linear/nonlinear adapters for more complex distribution shifts. These extensions will broaden the applicability of the proposed method while maintaining its computational efficiency.

### ACKNOWLEDGMENTS

This work was supported by the National Research Foundation of Korea(NRF) grant funded by the Korea government(MSIT) (RS-2025-00521452), and by the National Research Foundation of Korea(NRF) grant funded by the Korea government(MSIT) (RS-2025-15002982).

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

# A  ADAPTATION ALGORITHM OF COSA

This section details the core adaptation algorithm of COSA. This algorithm extends the optimizer mentioned in Equation 2, combining short-term loss trends, cosine annealing, and adaptive gradient clipping, batch-wise learning rate reset. We apply widely used values for the coefficients of $CALR$ and the threshold value of early stopping. The base model remains frozen, operating only on adapter parameters $\varphi = \{\boldsymbol{W}, \boldsymbol{b}, \boldsymbol{g}\}$.

**Purpose:** Performs rapid adaptation of linear adapters using direct loss, adaptive learning rate scheduling, and gradient clipping to improve prediction accuracy within a few adaptation step $S$.

---

**Algorithm 1** COSA adaptation.

---

**Require:** Stream of batches $(\hat{\mathbf{Y}}, \mathbf{Y}^{\text{true}})$ with length $B$, context vector $\boldsymbol{C}$, steps $S$, $\eta_{\min}, \eta_{\max}$, weight decay $\lambda$, clip base $c$
**Ensure:** adapted predictions with improved accuracy over the base model

1: **for** each batch $(\hat{\mathbf{Y}}, \mathbf{Y}^{\text{true}})$ in Data Stream **do**                    ▷ Loop over new batches
2:    **Initialize:** $\eta \leftarrow \eta_{\max}$, loss history $\mathcal{H} \leftarrow []$                    ▷ Reset LR for new batch
3:    **for** $s = 1$ to $S$ **do**                    ▷ Main adaptation loop
4:       **Forward pass:** form $\boldsymbol{X}^{(a)} = [\mathbf{Y}^{(0)} \| \boldsymbol{C}]$, then $\mathbf{H} = (\boldsymbol{W}\, \boldsymbol{X}^{(a)\top} + \boldsymbol{b})^{\top}$
5:       **Gating:** $\hat{\mathbf{Y}} \leftarrow \mathbf{Y}^{(0)} + \tanh(\boldsymbol{g}) \odot \mathbf{H}$
6:       **MSE loss:** $\mathcal{L} \leftarrow \left\|(\hat{\mathbf{Y}} - \mathbf{Y}^{\text{true}})\right\|_F^2 + \lambda\|\varphi\|_2^2$
7:       **Learning rate adaptation:**
8:       **if** $|\mathcal{H}| \geq 2$ **then**
9:          $\Delta \leftarrow \mathcal{L} - \mathcal{H}[-1]$                    ▷ Recent loss change
10:         **if** $\Delta > 0$ **then**                    ▷ Loss increased - reduce LR
11:           $\eta \leftarrow \max(0.5\,\eta,\, \eta_{\min})$
12:         **else if** $|\Delta| < 10^{-6}$ **then**                    ▷ Converged - increase LR for next batch
13:           $\eta \leftarrow \min(1.1\,\eta,\, \eta_{\max})$
14:         **end if**
15:       **end if**
16:       **Cosine annealing:** $\eta \leftarrow \eta_{\min} + \frac{1}{2}(\eta - \eta_{\min})\big(1 + \cos(\frac{s\pi}{S})\big)$
17:       **Gradient computation:** $g_\varphi \leftarrow \nabla_\varphi \mathcal{L}$
18:       **Adaptive clipping:** $\|g_\varphi\| \leftarrow \min\big(\|g_\varphi\|,\, \max(c,\, \mathcal{L})\big)$
19:       **Parameter update:** $\varphi \leftarrow \varphi - \eta\, g_\varphi$
20:       **Early stopping:**
21:       **if** $s > 2$ and $|\mathcal{H}[-1] - \mathcal{H}[-2]| < 10^{-6}$ **then**
22:         **break**
23:       **end if**
24:    **end for**
25: **end for**

---

# B  COMPUTATIONAL COST

In this section, we report theoretical calculations of parameters, FLOPs, and memory footprint.

For univariate time series, the number of parameters is as follows:

**Parameter count.**

$$\underbrace{L(L+K)}_{\boldsymbol{W}} + \underbrace{L}_{\boldsymbol{b}} + \underbrace{1}_{\boldsymbol{g}} \quad \Rightarrow \quad \#\text{params} = \big(L(L+K) + L + 1\big).$$

**FLOPs per adaptation step (batch of size $B$).** The dominant cost is the linear transform and residual composition:

$$\mathcal{O}\big(B\,L(L+K)\big) \quad \text{for } \boldsymbol{W}\mathbf{X}^{(a)}, \qquad \text{plus } \mathcal{O}(BL) \text{ for gating \& residual add.}$$

**Memory footprint.** Additional activations are modest: the linear residual $\mathbf{H} \in \mathbb{R}^{B \times L}$ and the context vector $\boldsymbol{C} \in \mathbb{R}^{K}$. The total adaptation cost scales linearly with the number of adaptation steps $S$ and variables $V$.

Table 4: Average $B$ of each dataset determined by PAAS.

| Dataset | ETTh1 | ETTh2 | ETTm1 | ETTm2 | Exchange Rate | Weather |
|---|---|---|---|---|---|---|
| Average $B$ | 24.55 | 38.41 | 92.73 | 83.41 | 92.80 | 80.38 |

## C  BATCH SIZE ANALYSIS OF PAAS

**Adaptive vs. Fixed Batch Strategy:** While COSA-F shows the best performance in most cases, COSA-P generally performs better on the ETTh1 and ETTh2 datasets, revealing important insights about temporal adaptation strategies. The reason for this trend is that the size of $B$ determined by PAAS is often smaller than 48 for these datasets, as shown in Table 4, enabling more frequent adaptation that better captures the higher-frequency patterns characteristic of these hourly datasets.

This differential performance validates the importance of dataset-specific adaptation scheduling: datasets with more complex temporal dynamics benefit from more frequent adaptation (smaller $B$), while datasets with smoother patterns can effectively use larger batch sizes for computational efficiency.

## D  BEHAVIOR ANALYSIS OF COSA

### D.1  ANALYSIS BETWEEN GATING AND LINEAR RESIDUAL LAYER

In COSA, the gating is defined as $gating = \tanh(g) \in [-1, 1]$, where $g$ is a learnable parameter, and this bounded scalar modulates the correction strength by multiplying the output of the linear residual layer. If we were to use $g$ directly instead of $\tanh(g)$, small variations in $g$ could induce disproportionately large and unstable changes in the correction, making the adapter overly sensitive to noisy points. The $\tanh$ transform keeps the gating bounded, ensuring that changes in $g$ are reflected smoothly and gradually. When the residual magnitude spikes, the gate moves toward 0, as shown in Figure 5, thereby attenuating the residual correction and stabilizing the adaptation process.

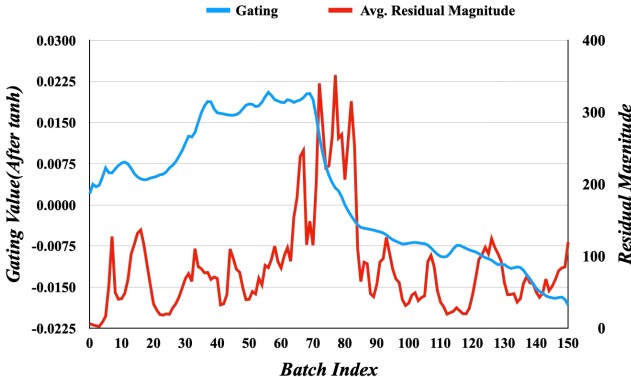

Figure 5: Trajectory of average gating and average residual magnitude of batches over time.

### D.2  ANALYSIS BETWEEN LEARNING RATE AND MSE

Figure 6 visualizes the trajectory of pre-adaptation loss and learning rate for the iTransformer–ETTm1, $L = 96$ case. As shown in Figure 6, when a short-term loss spike occurs, CALR immediately decreases the learning rate to minimize the impact of the perturbation, and once the loss enters a stable decreasing phase, CALR increases the learning rate again to promote rapid re-adaptation. This control mechanism suppresses excessive parameter drift without requiring roll-back, enabling COSA to recover its correction performance instantly after a perturbation. The interaction between the learning rate and loss shows that, in non-stationary environments with short-term perturbations, COSA can respond and recover performance stably.

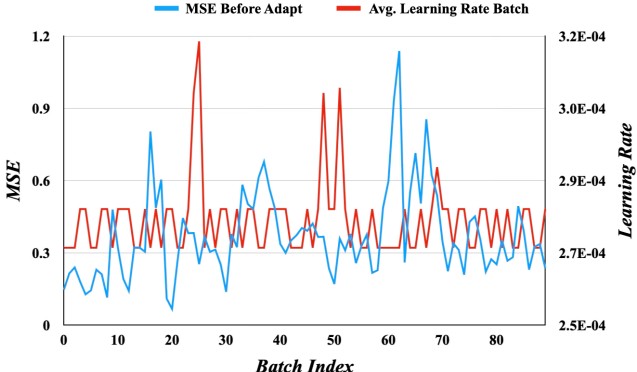

Figure 6: Trajectory of initial MSE and average learning rate of batches over time.

Table 5: Average ERV and NAR across all benchmark datasets and base models.

|  | TAFAS | PETSA | COSA |
|---|---|---|---|
| ERV ↑ | .0100 | .0160 | **.0768** |
| NAR ↓ | 34.94% | 36.34% | **20.25%** |

## E    QUALITY OF TTA

To evaluate TTA quality, we measured Explained Residual Variance (ERV) and Negative Adaptation Rate (NAR) metrics across all datasets, base models, and forecasting horizons. ERV is defined as in Equation 3, where $\hat{R}$ represents residuals for TTA-applied predictions and $R^{(0)}$ represents residuals for base model predictions. Specifically, ERV quantifies the extent to which TTA reduces the residual variance of the base model's predictions. Higher ERV values indicate greater residual variance reduction and correspondingly improved prediction performance through TTA.

$$\text{ERV} \;=\; 1 \;-\; \frac{\text{Var}(\hat{R})}{\text{Var}(R^{(0)})}. \tag{3}$$

NAR is the ratio of prediction windows where the MSE worsened when TTA was applied, with smaller values indicating better performance.

$$NAR = \frac{1}{N}\sum_{t=1}^{N} I[MSE(\hat{\mathbf{Y}}_t) > MSE(\mathbf{Y}_t^{(0)})], \quad where \begin{cases} I[\text{con}] = 1 & \text{if con = True} \\ I[\text{con}] = 0 & \text{otherwise} \end{cases} \tag{4}$$

As shown in Table 5, COSA shows the highest ERV and the lowest NAR. This indicates that COSA provides effective TTA while decreasing the residual and improving accuracy on average.

## F    FURTHER EXPERIMENTS

### F.1    COMPARISON ON A LARGER DATASET

To demonstrate that COSA can achieve stable and substantial performance gains even in larger-scale environments, we additionally conducted experiments on the Electricity dataset Lai et al. (2018); Godahewa et al. (2021). We followed the same experimental setup as in Section 4.1, and used three representative base models, iTransformer, DLinear, and FreTS. Table 6 is consistent with the findings in Section 4.2, COSA achieves either the best or second-best performance across all forecasting horizons $L$, and unlike other methods, which exhibit performance degradation compared to baselines in some cases, COSA improves their performances in every case. These results indicate that COSA remains robust and effective even in large-scale environments.

Table 6: Prediction accuracy comparison on Electricity dataset.

|  | iTransformer | | | | | DLinear | | | | | FreTS | | | | |
|---|---|---|---|---|---|---|---|---|---|---|---|---|---|---|---|
|  | No TTA | TAFAS | PETSA | COSA-F | COSA-P | No TTA | TAFAS | PETSA | COSA-F | COSA-P | No TTA | TAFAS | PETSA | COSA-F | COSA-P |
| 96 | .1663 | **.1568** | .1596 | .1571 | .1570 | .2235 | **.2224** | .2242 | .2232 | .2228 | .1824 | **.1781** | .1802 | .1801 | .1799 |
| 192 | .1794 | .1635 | .1644 | **.1631** | **.1631** | .2242 | .2153 | .2152 | .2146 | **.2143** | .1794 | .1894 | .1781 | .1780 | **.1779** |
| 336 | .1952 | .1743 | .1741 | .1730 | **.1727** | .2383 | .2247 | .2233 | **.2226** | .2223 | .1905 | .1885 | .1885 | .1881 | **.1879** |
| 720 | .2567 | .2316 | .2301 | .2251 | **.2239** | .2792 | .2629 | .2615 | .2557 | **.2541** | .2304 | .2299 | .2287 | .2236 | **.2221** |

## F.2 COMPARISON ON A REAL DATASET

To validate that COSA performs well not only on benchmark datasets but also in real-world deployment environments exhibiting more extreme distribution shifts, we conducted a performance evaluation on a real-world dataset. The dataset consists of power generation data collected from an actual solar power plant spanning from Jan. 24, 2024, 00:00 to Dec. 31, 2024, 23:59. It comprises hourly active power sensor readings from 30 devices with no missing values. The data was split into train, test, and validation sets at a ratio of 7:1:2. The dataset contains 8,232 rows, approximately 47.3% the size of ETTh1. We conducted experiments solely under the setting $L = 96$, $W = 96$, with all other configurations kept identical.

The experimental results demonstrate that, consistent with the benchmark findings, COSA achieves improved performance over existing methods regardless of $B$. This indicates that COSA can operate effectively even under more extreme distributional changes and can reliably calibrate forecasting performance on very small-scale datasets.

Table 7: Prediction accuracy comparison.

|  | iTransformer | | | | DLinear | | | | FreTS | | | |
|---|---|---|---|---|---|---|---|---|---|---|---|---|
| **Metric** | No TTA | TAFAS | COSA-F | COSA-P | No TTA | TAFAS | COSA-F | COSA-P | No TTA | TAFAS | COSA-F | COSA-P |
| Avg. MSE | .2901 | .2879 | **.2730** | .2897 | .2832 | .2813 | **.2801** | .2803 | .3013 | .2983 | **.2944** | .2980 |

## F.3 COMPARISON WITH VARYING INPUT/PREDICTION SEQUENCE LENGTH

To further verify the performance of COSA across diverse scenarios, we conducted experiments by varying both the input window $W$ and the prediction horizon $L$. To examine short-term forecasting rather than long-term forecasting, we added the setting $W = 96$, $L \in \{24, 48\}$. To evaluate performance under longer input windows, we additionally tested $W = 192$ with $L \in \{192, 336, 720\}$ and $W = 336$ with $L \in \{336, 720\}$. Table 8 summarizes the results across these different combinations of $W$ and $L$. Consistent with Section 4.2, COSA achieves the best or second-best performance in most cases, demonstrating its ability to maintain high predictive accuracy across a wide range of settings. Unlike other methods, which show performance degradation relative to baselines in several cases, COSA improves prediction accuracy over baselines in every case. In contrast, TAFAS and PETSA, which adopt dual-adapter architectures that modify both the input and output of base model incur substantial additional complexity as $W$ increases, which likely contributes to their degraded performance. Since COSA operates solely in the output space of base model, it delivers consistent performance gains regardless of $W$.

## F.4 COMPARISON WITH SOLID

SOLID (Chen et al., 2024b) is a fine-tuning method of the prediction layer of the base model by detecting context drift. Its reconditioner estimates context drift based on the mutual information between the model's residual and the input context. When drift is detected, SOLID selectively chooses samples and fine-tunes the model's prediction layer at the sample level. Like COSA, SOLID operates in the output space of base model. However, unlike COSA, which keeps base model frozen and performs corrections through a lightweight adapter, SOLID directly updates the prediction layer of base model.

Table 8: Prediction accuracy comparison with different input/prediction sequence length.

| | Input | Pred | iTransformer | | | | | DLinear | | | | | FreTS | | | | |
|---|---|---|---|---|---|---|---|---|---|---|---|---|---|---|---|---|---|
| | | | No TTA | TAFAS | PETSA | COSA-F | COSA-P | No TTA | TAFAS | PETSA | COSA-F | COSA-P | No TTA | TAFAS | PETSA | COSA-F | COSA-P |
| ETTh1 | 96 | 24 | .3269 | .3254 | .3299 | .3105 | **.3098** | .3437 | .3751 | .3766 | .3431 | **.3429** | .2953 | .3243 | .3239 | .2946 | **.2943** |
| | 96 | 48 | .3732 | .3768 | .3776 | .3539 | **.3527** | .3838 | .4177 | .4166 | .3826 | **.3820** | .3424 | .3770 | .3754 | .3416 | **.3413** |
| | 192 | 192 | .4459 | .5015 | .4893 | .4175 | **.4120** | .4172 | .4967 | .4834 | .4157 | **.4141** | .4235 | .4991 | .4835 | .4207 | **.4186** |
| | 192 | 336 | .4729 | .5709 | .5422 | .4486 | **.4456** | .4537 | .5558 | .5362 | .4528 | **.4489** | .4596 | .5677 | .5398 | .4577 | **.4535** |
| | 192 | 720 | .5885 | .6529 | .6687 | .5619 | **.5477** | .5828 | .6504 | .6526 | .5734 | **.5660** | .6484 | .6627 | .6728 | .6351 | **.6190** |
| | 336 | 336 | .5354 | .6139 | .5686 | .5209 | **.5059** | .4392 | .5652 | .5174 | .4368 | **.4289** | .4837 | .6114 | .5444 | .4783 | **.4674** |
| | 336 | 720 | .6512 | .6873 | .6635 | .6115 | **.5780** | .5748 | .6344 | .6443 | .5589 | **.5198** | .7265 | .6905 | .6738 | .6960 | **.6295** |
| ETTh2 | 96 | 24 | .0837 | .1277 | .1266 | **.0812** | .0812 | .0862 | .1217 | .1218 | **.0861** | .0862 | .0812 | .1180 | .1178 | **.0806** | .0807 |
| | 96 | 48 | .1047 | .1518 | .1520 | **.1007** | .1008 | .1031 | .1444 | .1452 | **.1028** | .1030 | .1004 | .1452 | .1447 | **.0996** | .0996 |
| | 192 | 192 | .1633 | .2272 | .2280 | **.1546** | .1563 | .1344 | .2155 | .2096 | **.1334** | .1338 | .1437 | .2218 | .2184 | **.1412** | .1420 |
| | 192 | 336 | .1658 | .2541 | .2487 | **.1551** | .1573 | .1474 | .2445 | .2394 | **.1460** | .1468 | .1556 | .2494 | .2460 | **.1519** | .1536 |
| | 192 | 720 | .2197 | .3453 | .3192 | **.2110** | .2158 | .1782 | .3152 | .2962 | **.1762** | .1778 | .1943 | .3261 | .3086 | **.1892** | .1925 |
| | 336 | 336 | .2070 | .2986 | .3005 | **.2021** | .2108 | .1469 | .2366 | .2375 | **.1447** | .1468 | .1722 | .2579 | .2604 | **.1645** | .1695 |
| | 336 | 720 | .3334 | .4253 | .4355 | **.3255** | .3334 | .1807 | .3101 | .3004 | **.1776** | .1798 | .1969 | .3130 | .3124 | **.1920** | .1949 |
| ETTm1 | 96 | 24 | .2543 | **.2359** | .2429 | .2405 | .2450 | .2578 | .2616 | .2599 | **.2523** | .2569 | .2494 | .2503 | .2471 | **.2449** | .2492 |
| | 96 | 48 | .3305 | **.3099** | .3135 | .3115 | .3133 | .3105 | .3167 | .3151 | **.3035** | .3064 | .3191 | .3269 | .3222 | **.3171** | .3177 |
| | 192 | 192 | .3800 | .4006 | .3961 | **.3591** | .3603 | .3694 | .4029 | .3942 | **.3643** | .3648 | .3551 | .3863 | .3815 | **.3524** | .3528 |
| | 192 | 336 | .4336 | .4499 | .4401 | **.4149** | .4150 | .4239 | .4620 | .4436 | **.4162** | .4169 | .4036 | .4371 | .4290 | **.3985** | .3987 |
| | 192 | 720 | .4992 | .5385 | .5090 | **.4716** | .4718 | .4797 | .5376 | .5101 | **.4671** | .4681 | .4631 | .5152 | .4941 | **.4550** | .4560 |
| | 336 | 336 | .4820 | .4913 | .4441 | **.4490** | .4592 | .3981 | .4444 | .4244 | **.3968** | .3977 | .4011 | .4380 | .4275 | **.3975** | .4006 |
| | 336 | 720 | .5202 | .5432 | .5169 | **.4678** | .4904 | .4437 | .5053 | .4852 | **.4256** | .4423 | .4407 | .5046 | .4877 | **.4261** | .4397 |
| ETTm2 | 96 | 24 | .0542 | .0800 | .0744 | **.0530** | .0531 | .0601 | .0782 | .0777 | **.0598** | .0599 | .0562 | .0748 | .0741 | **.0561** | .0561 |
| | 96 | 48 | .0735 | .1044 | .0988 | **.0731** | .0732 | .0767 | .1027 | .1015 | **.0761** | .0762 | .0736 | .0992 | .0987 | .0734 | **.0733** |
| | 192 | 192 | .1035 | .1647 | .1601 | .1023 | **.1022** | .0990 | .1451 | .1429 | **.0984** | .0985 | .0998 | .1480 | .1456 | **.0982** | .0984 |
| | 192 | 336 | .1343 | .2191 | .1970 | .1339 | **.1335** | .1201 | .1791 | .1741 | **.1190** | .1195 | .1210 | .1863 | .1781 | .1200 | **.1192** |
| | 192 | 720 | .1628 | .2511 | .2377 | **.1564** | .1589 | .1518 | .2291 | .2247 | **.1473** | .1498 | .1476 | .2337 | .2232 | **.1436** | .1446 |
| | 336 | 336 | .1354 | .2280 | .1987 | **.1328** | .1347 | .1176 | .1782 | .1704 | **.1167** | .1174 | .1198 | .1815 | .1760 | **.1172** | .1188 |
| | 336 | 720 | .1645 | .2785 | .2517 | **.1455** | .1604 | .1456 | .2338 | .2185 | **.1317** | .1446 | .1490 | .2309 | .2255 | **.1289** | .1453 |
| Exchange Rate | 96 | 24 | .0318 | .0292 | **.0273** | .0306 | .0306 | .0434 | .0397 | **.0393** | .0434 | .0434 | .0283 | .0254 | **.0241** | .0283 | .0283 |
| | 96 | 48 | .0526 | .0542 | **.0479** | .0516 | .0516 | .0606 | .0590 | **.0579** | .0606 | .0606 | .0481 | .0438 | **.0418** | .0480 | .0481 |
| | 192 | 192 | .2045 | **.1999** | .2189 | .2040 | .2040 | .1715 | .1819 | .2024 | .1714 | .1715 | .1655 | .1728 | .1809 | **.1655** | **.1655** |
| | 192 | 336 | .2963 | .3221 | .3511 | **.2919** | .2933 | .2878 | .3316 | .3484 | **.2850** | .2875 | .2804 | .3129 | .3318 | **.2780** | .2802 |
| | 192 | 720 | .8546 | **.7713** | .7828 | .8316 | .8320 | .9037 | .9393 | .9488 | **.9016** | .9016 | .4506 | .8200 | .8315 | **.4501** | .4501 |
| | 336 | 336 | .4088 | **.3790** | .4204 | .4036 | .4036 | .2753 | .3036 | .3323 | **.2700** | .2751 | .3246 | .3219 | .3589 | **.3182** | .3245 |
| | 336 | 720 | 1.7882 | **1.0041** | 1.0775 | 1.5829 | 1.5828 | .5274 | .8620 | .8816 | **.5269** | .5269 | .4266 | .9353 | .9175 | **.4263** | .4263 |
| Weather | 96 | 24 | .1095 | .1026 | **.1016** | .1077 | .1077 | .1207 | **.1151** | **.1151** | .1200 | .1199 | .1193 | **.1085** | .1111 | .1183 | .1183 |
| | 96 | 48 | .1380 | **.1292** | .1310 | .1362 | .1363 | .1581 | .1499 | **.1475** | .1559 | .1560 | .1530 | **.1435** | .1452 | .1508 | .1510 |
| | 192 | 192 | .2165 | .2102 | **.2051** | .2075 | .2099 | .2393 | .2225 | **.2214** | .2332 | .2354 | .2131 | .2058 | **.2021** | .2086 | .2108 |
| | 192 | 336 | .2747 | .2588 | **.2549** | .2611 | .2653 | .2904 | .2709 | **.2691** | .2791 | .2829 | .2694 | .2547 | **.2531** | .2604 | .2637 |
| | 192 | 720 | .3321 | .3316 | .3390 | **.3151** | .3211 | .3426 | .3363 | .3446 | **.3276** | .3339 | .3290 | .3286 | .3360 | **.3158** | .3205 |
| | 336 | 336 | .2988 | **.2705** | .2708 | .2778 | .2978 | .2722 | .2633 | .2619 | **.2627** | .2707 | .2526 | .2520 | .2474 | **.2444** | .2504 |
| | 336 | 720 | .3381 | .3423 | .3557 | **.2885** | .3340 | .3245 | .3247 | .3316 | **.2742** | .3207 | .3129 | .3203 | .3229 | **.2648** | .3088 |

Using the publicly available source codes, we evaluated SOLID on three base forecasting models: DLinear, FreTS, and iTransformer. The results are summarized in Table 9. Across all settings, COSA achieves higher predictive accuracy compared to SOLID.

Moreover, COSA is significantly more efficient: it requires only $7.44 \pm 0.0488$ seconds of wall-clock time on average, whereas SOLID incurs a much larger overhead with $306.55 \pm 2.6183$ seconds. The substantial cost of SOLID arises from computing context drift and repeatedly updating the prediction layer of the base model, processes that are far heavier than COSA's lightweight updates. Because COSA freezes base model and updates only a compact output-space adapter, it offers a dramatically faster execution while maintaining higher accuracy.

## F.5 COMPARISON WITH DYNATTA

This section presents complete baseline results of prediction accuracy comparison with various existing methods. DynaTTA requires setting a total of 15 hyperparameters, and due to this complexity, we report its performance using the best settings from our trials. The used hyperparameters of DynaTTA are as follows:

- HIDDEN_DIM=64, GATING_INIT=0.01, BASE_LR= 0.005, MSE_BUFFER_SIZE=100, RTAB_SIZE=50, RDB_SIZE=30, METRIC_HISTORY_SIZE=20, ALPHA_MIN=0.0001, ALPHA_MAX=0.01, KAPPA=2.0, ETA=0.1, EPS=1e-8, WARMUP_FACTOR=2, UPDATE_BUFFERS_INTERVAL=5, UPDATE_METRICS_INTERVAL=3

Table 9: Comparison with SOLID

| | | iTransformer | | DLinear | | FreTS | |
|---|---|---|---|---|---|---|---|
| | | SOLID | COSA-P | SOLID | COSA-P | SOLID | COSA-P |
| ETTh1 | 96 | .4404 | **.4363** | .4595 | **.4574** | **.4093** | .4371 |
| | 192 | .4935 | **.4919** | .5063 | **.5066** | **.4701** | .4940 |
| | 336 | .5420 | **.5300** | .5602 | **.5528** | **.5254** | .5467 |
| | 720 | **.6523** | .5638 | .7107 | **.6107** | .6980 | **.6259** |
| ETTh2 | 96 | **.2480** | .2493 | .2315 | **.2281** | .2354 | **.2350** |
| | 192 | .3061 | **.2947** | .2824 | **.2819** | .2830 | **.2972** |
| | 336 | **.3339** | **.3339** | .3103 | **.3083** | .3248 | **.3153** |
| | 720 | .3701 | **.3591** | **.3136** | .3477 | **.3167** | .3399 |
| ETTm1 | 96 | **.3353** | .3455 | .3634 | **.3456** | **.3121** | .3525 |
| | 192 | .4266 | **.4140** | .4336 | **.4222** | .4299 | **.4212** |
| | 336 | .5019 | **.4643** | .5071 | **.4858** | .4915 | **.4775** |
| | 720 | .5986 | **.5102** | .5821 | **.4991** | .5612 | **.4982** |
| ETTm2 | 96 | .1641 | **.1632** | .1590 | **.1583** | .1574 | **.1569** |
| | 192 | .2291 | **.2173** | .1910 | **.1943** | .1933 | **.1934** |
| | 336 | **.2525** | .2535 | .2138 | **.2242** | .2328 | **.2211** |
| | 720 | .2704 | **.2606** | .2419 | **.2316** | .2377 | **.2314** |
| Exchange Rate | 96 | .0873 | **.0837** | .0913 | **.0834** | .0822 | **.0766** |
| | 192 | .1783 | **.1479** | .1826 | **.1519** | .1723 | **.1499** |
| | 336 | .3294 | **.2624** | .3276 | **.2480** | .3218 | **.2461** |
| | 720 | .7531 | **.4460** | .8872 | **.4481** | .8325 | **.4458** |
| Weather | 96 | .1753 | **.1617** | .1954 | **.1793** | .1849 | **.1737** |
| | 192 | .2231 | **.2088** | .2403 | **.2217** | .2309 | **.2189** |
| | 336 | .2801 | **.2515** | .2918 | **.2626** | .2843 | **.2587** |
| | 720 | .3450 | **.2730** | .3643 | **.2708** | .3561 | **.2692** |

Table 10: Prediction accuracy comparison with DynaTTA Grover & Etemad (2025).

| | | Transformer-based | | | | | | Linear-based | | | | | | MLP-based | | | | | |
|---|---|---|---|---|---|---|---|---|---|---|---|---|---|---|---|---|---|---|---|
| | | iTransformer | | | PatchTST | | | DLinear | | | OLS | | | FreTS | | | MICN | | |
| | | DynaTTA | COSA-F | COSA-P | DynaTTA | COSA-F | COSA-P | DynaTTA | COSA-F | COSA-P | DynaTTA | COSA-F | COSA-P | DynaTTA | COSA-F | COSA-P | DynaTTA | COSA-F | COSA-P |
| ETTh1 | 96 | .4523 | .4368 | .4363 | .8371 | .4242 | .4238 | .4708 | .4574 | .4482 | .4486 | .4390 | .4372 | .4508 | .4384 | .4371 | .5063 | .4693 | .4684 |
| | 192 | .5175 | .4961 | .4919 | .8006 | .4830 | .4805 | .5321 | .5066 | .5050 | .5096 | .4915 | .4906 | .5139 | .4951 | .4940 | .5745 | .5372 | .5328 |
| | 336 | .5874 | .5651 | .5300 | .8097 | .5438 | .5320 | .5792 | .5528 | .5456 | .5626 | .5385 | .5320 | .5840 | .5467 | .5351 | .6594 | .5950 | .5878 |
| | 720 | .7123 | .5958 | .5638 | 1.0887 | .6113 | .5822 | .7112 | .6107 | .5896 | .6933 | .5969 | .5733 | .7095 | .6259 | .5959 | .8262 | .7001 | .6504 |
| ETTh2 | 96 | .2630 | .2504 | .2493 | .4154 | .2349 | .2343 | .2338 | .2300 | .2281 | .2326 | .2232 | .2265 | .2395 | .2367 | .2350 | .2661 | .2492 | .2485 |
| | 192 | .3210 | .2983 | .2947 | .4315 | .2665 | .2608 | .2888 | .2827 | .2819 | .2911 | .2796 | .2791 | .2916 | .2816 | .2824 | .3417 | .3049 | .3017 |
| | 336 | .3677 | .3241 | .3339 | .4386 | .2971 | .2978 | .3380 | .3050 | .3083 | .3430 | .3003 | .3043 | .3474 | .3031 | .3153 | .3777 | .3241 | .3310 |
| | 720 | .4646 | .3487 | .3591 | .4991 | .3233 | .3428 | .4325 | .3062 | .3177 | .4270 | .3177 | .3453 | .4297 | .3169 | .3650 | .5229 | .3650 | .3885 |
| ETTm1 | 96 | .3753 | .3447 | .3455 | .7205 | .3625 | .3626 | .3814 | .3456 | .3475 | .3830 | .3454 | .3475 | .3723 | .3520 | .3525 | .4095 | .3837 | .3831 |
| | 192 | .4835 | .4124 | .4140 | .6898 | .4250 | .4258 | .4809 | .4113 | .4122 | .4827 | .4115 | .4119 | .4811 | .4150 | .4212 | .5331 | .4476 | .4514 |
| | 336 | .5843 | .4569 | .4643 | .8073 | .4568 | .4697 | .5711 | .4753 | .4858 | .5719 | .4748 | .4749 | .5739 | .4661 | .4775 | .6418 | .4832 | .5054 |
| | 720 | .7243 | .4773 | .5102 | .9292 | .4681 | .4882 | .6569 | .4774 | .4991 | .6884 | .4763 | .5007 | .7152 | .4718 | .4982 | .6932 | .5029 | .5225 |
| ETTm2 | 96 | .1832 | .1627 | .1632 | .2508 | .1558 | .1562 | .1640 | .1583 | .1586 | .1586 | .1582 | .1586 | .1661 | .1568 | .1569 | .2033 | .1702 | .1704 |
| | 192 | .2897 | .2171 | .2173 | .2744 | .2007 | .2022 | .2040 | .1904 | .1905 | .2115 | .1906 | .1907 | .2048 | .1905 | .1908 | .2349 | .2102 | .2120 |
| | 336 | .3246 | .2435 | .2535 | .3434 | .2258 | .2352 | .2908 | .2083 | .2242 | .2917 | .2131 | .2226 | .2778 | .2098 | .2211 | .2988 | .2337 | .2351 |
| | 720 | .5344 | .2477 | .2606 | .4447 | .2446 | .2645 | .3765 | .2215 | .2316 | .4571 | .2171 | .2349 | .4020 | .2158 | .2314 | .5362 | .2477 | .2643 |
| Exchange Rate | 96 | .0983 | .0818 | .0837 | .1217 | .0765 | .0788 | .0948 | .0812 | .0834 | .0918 | .0756 | .0773 | .0906 | .0744 | .0766 | .1210 | .0955 | .1008 |
| | 192 | .1970 | .1403 | .1479 | .2517 | .1464 | .1570 | .1975 | .1459 | .1519 | .1749 | .1393 | .1457 | .1871 | .1366 | .1499 | .2227 | .1663 | .1722 |
| | 336 | .3251 | .2089 | .2624 | .3728 | .1983 | .2445 | .3001 | .2039 | .2480 | .3024 | .2020 | .2323 | .3111 | .2053 | .2461 | .3536 | .2119 | .2660 |
| | 720 | .8790 | .3421 | .4460 | .9999 | .3543 | .4662 | .8812 | .3494 | .4481 | .8300 | .3444 | .4541 | .8348 | .3352 | .4458 | .8570 | .3871 | .4815 |
| Weather | 96 | .1823 | .1597 | .1617 | .2317 | .1624 | .1634 | .1950 | .1773 | .1793 | .1803 | .1772 | .1772 | .1937 | .1724 | .1737 | .2303 | .1636 | .1651 |
| | 192 | .2678 | .2067 | .2088 | .2611 | .2006 | .2108 | .3311 | .2216 | .2217 | .3074 | .2223 | .2237 | .2850 | .2135 | .2189 | .3839 | .2082 | .2120 |
| | 336 | .3894 | .2503 | .2515 | .3329 | .2451 | .2488 | .4103 | .2567 | .2626 | .9593 | .2551 | .2642 | .3970 | .2561 | .2587 | .5355 | .2729 | .2737 |
| | 720 | .4996 | .2480 | .2730 | .4067 | .2590 | .2713 | .4915 | .2581 | .2708 | .4974 | .2579 | .2708 | .5251 | .2573 | .2692 | .5773 | .2582 | .2855 |

## F.6 Comparison with Various Batch Sizes

Table 11a and Table 11b summarize the prediction accuracy and efficiency overhead under different batch sizes $B$. Thanks to CALR's stability-induced design, COSA adapts reliably even with very small batches, and the resulting increase in update frequency often leads to improved forecasting accuracy. Notably, even in extremely small settings such as $B = 8$, COSA maintains higher accuracy than existing TTA methods, demonstrating resilience against over-correction and short-term perturbations. On the other hand, smaller $B$ inevitably increases the number of adaptation steps, leading

Table 11: Performance comparison with different batch sizes $B$.

(a) Prediction accuracy.

|   | No TTA | TAFAS | PETSA | PAAS | 8 | 16 | 24 | 48 | 96 |
|---|---|---|---|---|---|---|---|---|---|
| **96** | 0.2545 | 0.2471 | 0.2480 | 0.2409 | **0.1908** | 0.2202 | 0.2330 | 0.2398 | 0.2411 |
| **192** | 0.3144 | 0.3038 | 0.3046 | 0.2887 | **0.1900** | 0.2295 | 0.2524 | 0.2848 | 0.2946 |
| **336** | 0.3839 | 0.3653 | 0.3664 | 0.3280 | **0.1837** | 0.2292 | 0.2526 | 0.3013 | 0.3483 |
| **720** | 0.5539 | 0.5226 | 0.5232 | 0.3804 | **0.1811** | 0.2384 | 0.2672 | 0.3286 | 0.4158 |

(b) Wall-clock time (Seconds).

|   | TAFAS | PETSA | PAAS | 8 | 16 | 24 | 48 | 96 |
|---|---|---|---|---|---|---|---|---|
| **96** | 5.9271 | 7.4129 | 7.0642 | 47.3922 | 25.6158 | 17.2297 | 9.6333 | **5.1258** |
| **192** | 6.0383 | 7.3675 | 7.1419 | 49.4422 | 25.1222 | 18.0969 | 9.4344 | **5.5133** |
| **336** | 6.1554 | 7.6588 | 7.4131 | 49.2211 | 25.1119 | 17.2906 | 9.5644 | **5.8081** |
| **720** | **6.5300** | 7.9867 | 8.1578 | 46.3711 | 24.4683 | 17.6253 | 10.3036 | 6.5397 |

to higher adaptation time and a clear computation–accuracy trade-off. Considering this trade-off, we adopt $B = 48$ for COSA-F in our main experiments, which provides a balanced choice between accuracy gains and computational efficiency.

# G   ADDITIONAL ABLATIONS

This section provides additional ablation studies on various design choices of COSA. Each experiment was conducted to understand the impact of specific components and determine optimal hyperparameters. The default setting uses mean-based context aggregation with $K = 10$ and $S = 3$.

## G.1   CONTEXT BUILDING METHODS COMPARISON

### G.1.1   STATISTICAL METHODS

This experiment was conducted to evaluate the impact of different context construction methods on adapter performance. We compared three methods, i.e., Mean, Median, and Weighted Average (WA), to find the optimal context construction strategy. The weight of WA was designed to assign greater weight to recent values using exponential decay weighting. Table 12 presents a comprehensive performance comparison of the three context construction methods.

Mean-based context construction demonstrated superior performance compared to median and weighted average approaches across most experimental configurations. While median-based aggregation provided robustness against outliers, it resulted in lower overall accuracy. The weighted average approach showed marginal improvements relative to its implementation complexity. These findings support the adoption of simple statistical aggregation for effective context summarization in COSA.

### G.1.2   CONTEXT SELECTION STRATEGY

COSA aims not to model long-term time-series structure, but to perform fast and stable local residual correction for output bias observed in the current window. In non-stationary environments, the input distribution shifts continuously over time; thus, information from distant past windows may become misaligned with the current drift direction and deteriorate correction quality. For this reason, the default COSA employs a lightweight context vector constructed solely from the most recently observed batches.

To examine the effects of longer-range temporal patterns, we additionally implemented a Selective Context mechanism. This approach stores all past context values in a buffer and computes importance scores via attention between the current window and past contexts, selecting the top-$K$ values

Table 12: Prediction accuracy comparison of diverse aggregation functions.

| | | Transformer-based | | | | | | Linear-based | | | | | | MLP-based | | | | | |
|---|---|---|---|---|---|---|---|---|---|---|---|---|---|---|---|---|---|---|---|
| | | iTransformer | | | PatchTST | | | DLinear | | | OLS | | | FreTS | | | MICN | | |
| | | Mean | Median | WA | Mean | Median | WA | Mean | Median | WA | Mean | Median | WA | Mean | Median | WA | Mean | Median | WA |
| **ETTh1** | 96 | **.4327** | .4472 | .4472 | **.4092** | .4274 | .4275 | **.4615** | .4660 | .4657 | **.4359** | .4463 | .4462 | **.4362** | .4439 | .4440 | **.4704** | .4934 | .4926 |
| | 192 | **.4491** | .4850 | .4870 | **.4422** | .4778 | .4770 | **.4869** | .5090 | .5111 | **.4932** | .4897 | .4894 | **.4671** | .4893 | .4954 | **.4769** | .5567 | .5568 |
| | 336 | **.4476** | .5301 | .5317 | **.4550** | .5199 | .5285 | **.4632** | .5379 | .5378 | **.5188** | .5208 | .5209 | **.4411** | .5273 | .5245 | **.4814** | .5889 | .5904 |
| | 720 | **.4592** | .6237 | .6251 | **.4788** | .6329 | .6335 | **.4777** | .6358 | .6376 | **.5616** | .6243 | .6242 | **.4755** | .6439 | .6428 | **.5230** | .7196 | .7181 |
| **ETTh2** | 96 | **.2489** | .2532 | .2536 | **.2336** | .2346 | .2358 | **.2303** | .2352 | .2324 | **.2283** | .2300 | .2299 | **.2359** | .2379 | .2382 | **.2468** | .2566 | .2568 |
| | 192 | **.3003** | .3012 | .3013 | **.2708** | .2886 | .2894 | **.2682** | .2846 | .2863 | **.2762** | .2954 | .2893 | **.2766** | .2846 | .2917 | **.2906** | .3190 | .3145 |
| | 336 | **.3277** | .3671 | .3685 | **.2688** | .3190 | .3202 | **.2860** | .3283 | .3285 | **.2937** | .3030 | .3025 | **.2889** | .3214 | .3205 | **.3088** | .3429 | .3426 |
| | 720 | **.3311** | .4013 | .4023 | **.3091** | .3737 | .3737 | **.3006** | .3358 | .3358 | **.2992** | .3415 | .3415 | **.2979** | .3588 | .3564 | **.3454** | .4172 | .4185 |
| **ETTm1** | 96 | **.3428** | .3687 | .3685 | **.3604** | .3918 | .3921 | **.3453** | .3575 | .3574 | **.3440** | .3573 | .3572 | **.3522** | .3633 | .3633 | **.3804** | .4252 | .4248 |
| | 192 | **.4076** | .4307 | .4304 | **.4171** | .4398 | .4398 | **.4158** | .4266 | .4265 | **.4125** | .4265 | .4263 | **.4173** | .4257 | .4266 | **.4402** | .4742 | .4740 |
| | 336 | **.4663** | .4930 | .4949 | **.4662** | .4942 | .4944 | **.4784** | .5009 | .5005 | **.4700** | .4995 | .5006 | **.4771** | .4898 | .4886 | **.4937** | .5381 | .5381 |
| | 720 | **.4940** | .5614 | .5615 | **.4776** | .5359 | .5316 | **.4928** | .5640 | .5603 | **.4693** | .5638 | .5653 | **.4863** | .5414 | .5419 | **.5083** | .5830 | .5838 |
| **ETTm2** | 96 | **.1616** | .1672 | .1653 | **.1552** | .1591 | .1595 | **.1578** | .1594 | .1596 | **.1583** | .1596 | .1600 | **.1556** | .1592 | .1594 | **.1710** | .1712 | .1723 |
| | 192 | **.2172** | .2194 | .2240 | **.1989** | .2087 | .2121 | **.1900** | .1959 | .1957 | **.1919** | .1970 | .1995 | **.1908** | .1935 | .1933 | **.2102** | .2131 | .2132 |
| | 336 | **.2402** | .2700 | .2762 | **.2279** | .2821 | .2543 | **.2135** | .2448 | .2561 | **.2066** | .2499 | .2477 | **.2158** | .2353 | .2420 | **.2354** | .2523 | .2546 |
| | 720 | **.2550** | .3461 | .3463 | **.2397** | .3011 | .3222 | **.2357** | .2985 | .2833 | **.2104** | .3179 | .2821 | **.2360** | .2732 | .2753 | **.2510** | .3020 | .3084 |
| **Exchange Rate** | 96 | **.0843** | .0852 | .0852 | **.0803** | .0814 | .0815 | **.0843** | .0884 | .0885 | **.0728** | .0787 | .0787 | **.0790** | .0785 | .0785 | **.0980** | .1065 | .1068 |
| | 192 | **.1540** | .1606 | .1607 | **.1480** | .1544 | .1546 | **.1599** | .1653 | .1654 | **.1335** | .1507 | .1509 | **.1390** | .1500 | .1502 | **.1827** | .1870 | .1872 |
| | 336 | **.2588** | .2754 | .2756 | **.2611** | .2787 | .2789 | **.2565** | .2918 | .2920 | **.1833** | .2736 | .2738 | **.2511** | .2683 | .2684 | **.2660** | .3040 | .3043 |
| | 720 | **.5039** | .5113 | .5115 | **.4937** | .5076 | .5078 | **.5001** | .5267 | .5270 | **.3349** | .4983 | .4986 | **.4789** | .4977 | .4980 | **.4815** | .5806 | .5809 |
| **Weather** | 96 | **.1636** | .1726 | .1726 | **.1655** | .1731 | .1735 | **.1738** | .1931 | .1934 | **.1748** | .1934 | .1934 | **.1758** | .1831 | .1835 | **.1666** | .1739 | .1738 |
| | 192 | **.2073** | .2268 | .2265 | **.2052** | .2162 | .2162 | **.2217** | .2500 | .2520 | **.2144** | .2496 | .2490 | **.2151** | .2346 | .2347 | **.2060** | .2208 | .2205 |
| | 336 | **.2474** | .2707 | .2707 | **.2443** | .2668 | .2660 | **.2622** | .2891 | .2885 | **.2428** | .2911 | .2897 | **.2630** | .2736 | .2735 | **.2735** | .2650 | .2648 |
| | 720 | **.2907** | .3118 | .3119 | **.2682** | .3088 | .3090 | **.2713** | .3310 | .3325 | **.2487** | .3327 | .3331 | **.2583** | .3268 | .3247 | **.2670** | .3140 | .3094 |
| **Average** | | **.3121** | .3450 | .3458 | **.3032** | .3364 | .3366 | **.3097** | .3423 | .3422 | **.2990** | .3371 | .3354 | **.3046** | .3334 | .3340 | **.3284** | .3669 | .3670 |

to form the context vector. Such a mechanism can leverage repeated phases or cycles in datasets with strong periodicity.

Table 13a compares COSA (the standard recent-context construction) with the dynamic context selection method. While the dynamic context selection method achieves clear improvements on datasets such as ETT, where the periodic structure is strong and easily detectable, the overall performance of the original COSA remains superior. Selective Context also introduces non-trivial overhead, since computing importance scores increases both adaptation time and inference time. Moreover, in fully non-stationary settings where distributional characteristics change continuously, older contexts may become outdated and destabilize the correction process. Nonetheless, the observed gains on datasets with pronounced periodicity show the potential of Selective Context. We provide a more detailed discussion of these observations in Section H.

### G.1.3 COMPARISON WITH ENCODER-BASED CONTEXT

To compare with encoder-based context construction, we implemented an alternative approach that replaces the original statistics-based method in COSA with a temporal encoder that directly consumes the previously observed ground-truth sequence from the past 720 steps (the longest $L$). We added RNN-, LSTM-, and Attention-based encoders, each taking the past sequence in the form of $[720, 1]$ as input and producing a $[K, 1]$ context vector. The resulting context vector is concatenated with the base model's prediction output, just as in the original design, and then fed into the linear correction layer. The encoder is seamlessly integrated at the front of COSA, modifying only the context-generation stage while keeping the remaining components unchanged.

Table 14 shows that, except for a few isolated cases, the original statistics-based context (0.3240) performs better than encoder-based alternatives (0.3254, 0.3260, 0.3278). Furthermore, the added architectural complexity increases both adaptation and inference overhead. In non-stationary TTA settings, where the input distribution shifts rapidly and adaptation steps are short, it is difficult for an encoder to learn stable temporal representations. Consequently, the generated embeddings may become misaligned with the current drift direction or overfit to outdated historical patterns, ultimately degrading correction performance.

Table 13: Prediction accuracy and overhead of selective context.

(a) Prediction accuracy.

| | | Transformer-based | | | | Linear-based | | | | MLP-based | | | |
|---|---|---|---|---|---|---|---|---|---|---|---|---|---|
| | | iTransformer | | PatchTST | | DLinear | | OLS | | FreTS | | MICN | |
| | | Recent | Selective | Recent | Selective | Recent | Selective | Recent | Selective | Recent | Selective | Recent | Selective |
| ETTh1 | 96 | .4363 | **.4362** | .4238 | **.4234** | **.4482** | .4562 | .4372 | **.4359** | .4371 | **.4361** | .4684 | **.4673** |
| | 192 | **.4919** | .4927 | **.4805** | .4848 | **.5050** | .5088 | **.4906** | .4933 | **.4940** | .4965 | **.5328** | .5368 |
| | 336 | **.5300** | .5367 | .5320 | **.5310** | **.5456** | .5541 | .5320 | **.5368** | **.5351** | .5505 | **.5878** | .5966 |
| | 720 | **.5638** | .5671 | **.5822** | .5881 | **.5896** | .6180 | **.5733** | .6093 | .5959 | **.6430** | **.6504** | .7137 |
| ETTh2 | 96 | **.2493** | .2494 | **.2343** | .2349 | .2281 | **.2258** | .2265 | **.2262** | .2350 | **.2346** | **.2485** | .2486 |
| | 192 | .2947 | **.2942** | **.2608** | .2661 | .2819 | **.2813** | .2791 | .2825 | .2824 | **.2945** | **.3017** | .3027 |
| | 336 | **.3339** | .3367 | .2978 | **.2949** | .3083 | **.3064** | .3043 | **.3027** | **.3153** | .3175 | **.3310** | .3328 |
| | 720 | **.3591** | .3603 | **.3428** | .3453 | .3477 | **.3462** | **.3453** | .3619 | **.3399** | .3509 | **.3885** | .3906 |
| ETTm1 | 96 | .3455 | **.3440** | **.3626** | .3627 | .3475 | **.3456** | .3475 | **.3454** | .3525 | **.3522** | .3831 | **.3815** |
| | 192 | .4140 | **.4128** | **.4258** | .4278 | .4122 | **.4221** | **.4119** | .4219 | **.4212** | .4212 | **.4514** | .4533 |
| | 336 | **.4643** | .4718 | .4697 | **.4693** | .4858 | **.4857** | **.4749** | .4854 | **.4775** | .4805 | **.5054** | .5060 |
| | 720 | **.5102** | .5246 | .4882 | **.4868** | **.4991** | .5065 | .5007 | **.5005** | .4982 | **.4970** | **.5225** | .5256 |
| ETTm2 | 96 | .1632 | **.1631** | .1562 | **.1560** | .1586 | **.1582** | .1586 | **.1581** | **.1569** | .1570 | **.1704** | .1705 |
| | 192 | .2173 | **.2165** | .2022 | **.2011** | **.1905** | .1930 | **.1907** | .1952 | .1908 | **.1929** | **.2120** | .2125 |
| | 336 | **.2535** | .2551 | .2352 | **.2331** | **.2242** | .2245 | .2226 | **.2212** | .2211 | **.2185** | **.2351** | .2555 |
| | 720 | .2606 | **.2578** | .2645 | **.2633** | **.2316** | .2427 | **.2349** | .2368 | **.2314** | .2373 | **.2643** | .2764 |
| Exchange Rate | 96 | .0837 | **.0835** | **.0788** | .0789 | .0834 | **.0834** | .0773 | **.0773** | .0766 | **.0763** | .1008 | **.1007** |
| | 192 | **.1479** | .1516 | .1570 | **.1554** | **.1519** | .1543 | **.1457** | .1462 | **.1499** | .1514 | **.1722** | .1770 |
| | 336 | **.2624** | .2633 | **.2445** | .2478 | **.2480** | .2481 | **.2323** | .2361 | **.2461** | .2541 | **.2660** | .2732 |
| | 720 | **.4460** | .4749 | **.4662** | .4983 | **.4481** | .4984 | **.4541** | .4835 | **.4458** | .4914 | **.4815** | .5255 |
| Weather | 96 | .1617 | **.1616** | .1634 | **.1631** | .1793 | **.1790** | .1803 | **.1799** | .1737 | **.1731** | **.1651** | .1654 |
| | 192 | .2088 | **.2056** | **.2108** | .2109 | .2217 | **.2181** | .2237 | **.2222** | .2189 | **.2187** | **.2120** | .2134 |
| | 336 | **.2515** | .2524 | **.2488** | .2554 | **.2626** | .2665 | **.2642** | .2659 | **.2587** | .2603 | .2737 | **.2813** |
| | 720 | **.2730** | .2798 | **.2713** | .2798 | **.2708** | .2729 | **.2708** | .2721 | **.2692** | .2732 | **.2855** | .2970 |

(b) Overhead analysis.

| Method | # Params ↓ | Adaptation time/batch (ms) ↓ | Inference time/batch (ms) ↓ | Average MSE ↓ |
|---|---|---|---|---|
| Recent | **1,211,287** | **80.12 ± 13.58** | **1.25 ± .0984** | **.3240** |
| Selective | 1,212,217 | 83.64 ± 15.71 | 1.26 ± .1039 | .3287 |

These findings confirm that the statistics-based context remains the most robust and stable choice for non-stationary adaptation, while encoder-based context generation still demonstrates potential. We discuss these observations in greater detail in Section H.

## G.2 INPUT CALIBRATION EFFECTS

COSA performs residual correction directly in the output space, and we demonstrated that output-only correction is often sufficient. However, in certain time series, we observe that input-level spikes or local noise degrade the base model's predictions first, and this degradation subsequently propagates to the residual correction stage. To examine how such input perturbations influence the overall correction process, we conducted experiments combining COSA with the input-side GCM module from TAFAS. Table 15 reports the results.

In most cases, output-only correction achieves higher predictive accuracy. However, for datasets such as ETTh1, ETTh2, and ETTm2, where significant input noise is present, the combination with GCM produces improved results. In these cases, input GCM smooths the noisy input patterns, allowing the base model to generate more stable predictions, which in turn enhances the effectiveness of COSA.

Nevertheless, because GCM operates via distribution-shift normalization, it risks oversmoothing or removing meaningful drift signals when the input exhibits rapid or irregular changes. This behavior explains why, on average (in terms of MSE), output-only correction remains more stable across diverse non-stationary scenarios. Overall, when input disturbances are not the primary source of prediction error, output-only correction is the most robust and reliable option.

Table 14: Prediction accuracy and overhead comparison with encoder-based context.

(a) Prediction accuracy.

| | | Transformer-based | | | | Linear-based | | | | MLP-based | | | |
|---|---|---|---|---|---|---|---|---|---|---|---|---|---|
| | | iTransformer | | | | DLinear | | | | FreTS | | | |
| | | COSA | RNN | LSTM | Attn | COSA | RNN | LSTM | Attn | COSA | RNN | LSTM | Attn |
| ETTh1 | 96 | .4363 | .4363 | .4238 | .4234 | .4482 | .4562 | .4372 | .4359 | .4371 | .4361 | .4684 | .4674 |
| | 192 | .4919 | .4927 | .4805 | .4849 | .5050 | .5088 | .4906 | .4933 | .4940 | .4965 | .5328 | .5368 |
| | 336 | .5300 | .5367 | .5320 | .5310 | .5456 | .5541 | .5320 | .5368 | .5351 | .5506 | .5878 | .5967 |
| | 720 | .5638 | .5672 | .5822 | .5881 | .5896 | .6180 | .5733 | .6093 | .5959 | .6430 | .6504 | .7137 |
| ETTh2 | 96 | .2493 | .2494 | .2343 | .2349 | .2281 | .2258 | .2265 | .2263 | .2350 | .2346 | .2485 | .2486 |
| | 192 | .2947 | .2942 | .2608 | .2661 | .2819 | .2813 | .2791 | .2825 | .2824 | .2946 | .3017 | .3027 |
| | 336 | .3339 | .3367 | .2978 | .2950 | .3083 | .3064 | .3043 | .3028 | .3153 | .3175 | .3310 | .3328 |
| | 720 | .3591 | .3603 | .3428 | .3453 | .3477 | .3462 | .3453 | .3619 | .3399 | .3509 | .3885 | .3906 |
| ETTm1 | 96 | .3455 | .3440 | .3626 | .3627 | .3475 | .3456 | .3475 | .3454 | .3525 | .3522 | .3831 | .3815 |
| | 192 | .4140 | .4128 | .4258 | .4278 | .4122 | .4221 | .4119 | .4219 | .4212 | .4212 | .4514 | .4533 |
| | 336 | .4643 | .4718 | .4697 | .4693 | .4858 | .4857 | .4749 | .4854 | .4775 | .4806 | .5054 | .5060 |
| | 720 | .5102 | .5246 | .4882 | .4868 | .4991 | .5066 | .5007 | .5005 | .4982 | .4970 | .5225 | .5256 |
| ETTm2 | 96 | .1632 | .1631 | .1562 | .1561 | .1586 | .1582 | .1586 | .1581 | .1569 | .1570 | .1704 | .1705 |
| | 192 | .2173 | .2165 | .2022 | .2011 | .1905 | .1930 | .1907 | .1952 | .1908 | .1929 | .2120 | .2125 |
| | 336 | .2535 | .2551 | .2352 | .2331 | .2242 | .2245 | .2226 | .2212 | .2211 | .2185 | .2351 | .2555 |
| | 720 | .2606 | .2578 | .2645 | .2633 | .2316 | .2427 | .2349 | .2368 | .2314 | .2373 | .2643 | .2764 |
| Exchange Rate | 96 | .0837 | .0835 | .0788 | .0789 | .0834 | .0834 | .0773 | .0773 | .0766 | .0763 | .1008 | .1007 |
| | 192 | .1479 | .1517 | .1570 | .1554 | .1519 | .1543 | .1457 | .1462 | .1499 | .1515 | .1722 | .1771 |
| | 336 | .2624 | .2633 | .2445 | .2478 | .2480 | .2481 | .2323 | .2361 | .2461 | .2541 | .2660 | .2732 |
| | 720 | .4460 | .4749 | .4662 | .4983 | .4481 | .4984 | .4541 | .4835 | .4458 | .4914 | .4815 | .5255 |
| Weather | 96 | .1617 | .1617 | .1634 | .1631 | .1793 | .1790 | .1803 | .1799 | .1737 | .1731 | .1651 | .1654 |
| | 192 | .2088 | .2056 | .2108 | .2109 | .2217 | .2181 | .2237 | .2222 | .2189 | .2187 | .2120 | .2134 |
| | 336 | .2515 | .2524 | .2488 | .2554 | .2626 | .2665 | .2642 | .2659 | .2587 | .2614 | .2737 | .2813 |
| | 720 | .2730 | .2798 | .2713 | .2798 | .2708 | .2729 | .2708 | .2721 | .2692 | .2660 | .2855 | .2970 |
| | | PatchTST | | | | OLS | | | | MICN | | | |
| | | COSA | RNN | LSTM | Attn | COSA | RNN | LSTM | Attn | COSA | RNN | LSTM | Attn |
| ETTh1 | 96 | .4363 | .4363 | .4363 | .4357 | .4482 | .4582 | .4582 | .4570 | .4371 | .4375 | .4378 | .4362 |
| | 192 | .4919 | .4813 | .4825 | .4768 | .5050 | .5077 | .5069 | .5029 | .4940 | .4951 | .4953 | .4928 |
| | 336 | .5300 | .5319 | .5318 | .5305 | .5456 | .5532 | .5576 | .5620 | .5351 | .5445 | .5487 | .5426 |
| | 720 | .5638 | .5625 | .5626 | .5621 | .5896 | .6142 | .6154 | .6120 | .5959 | .6257 | .6257 | .6284 |
| ETTh2 | 96 | .2493 | .2507 | .2503 | .2511 | .2281 | .2304 | .2280 | .2278 | .2350 | .2349 | .2350 | .2355 |
| | 192 | .2947 | .2939 | .2938 | .2938 | .2819 | .2813 | .2808 | .2791 | .2824 | .2981 | .2984 | .2981 |
| | 336 | .3339 | .3334 | .3348 | .3337 | .3083 | .3040 | .3029 | .3004 | .3153 | .3138 | .3152 | .3186 |
| | 720 | .3591 | .3551 | .3534 | .3555 | .3477 | .3495 | .3536 | .3423 | .3399 | .3314 | .3484 | .3386 |
| ETTm1 | 96 | .3455 | .3459 | .3459 | .3453 | .3475 | .3457 | .3457 | .3455 | .3525 | .3529 | .3529 | .3519 |
| | 192 | .4140 | .4185 | .4183 | .4131 | .4122 | .4224 | .4223 | .4237 | .4212 | .4215 | .4215 | .4216 |
| | 336 | .4643 | .4588 | .4593 | .4550 | .4858 | .4867 | .4869 | .4874 | .4775 | .4778 | .4761 | .4811 |
| | 720 | .5102 | .5132 | .5138 | .5142 | .4991 | .4989 | .4962 | .4913 | .4982 | .5043 | .5033 | .5021 |
| ETTm2 | 96 | .1632 | .1642 | .1641 | .1636 | .1586 | .1585 | .1583 | .1583 | .1569 | .1568 | .1570 | .1569 |
| | 192 | .2173 | .2174 | .2175 | .2158 | .1905 | .1951 | .1939 | .1932 | .1908 | .1938 | .1938 | .1922 |
| | 336 | .2535 | .2536 | .2521 | .2544 | .2242 | .2217 | .2208 | .2230 | .2211 | .2214 | .2214 | .2183 |
| | 720 | .2606 | .2633 | .2641 | .2653 | .2316 | .2284 | .2359 | .2383 | .2314 | .2296 | .2280 | .2305 |
| Exchange Rate | 96 | .0837 | .0851 | .0853 | .0862 | .0834 | .0833 | .0837 | .0849 | .0766 | .0774 | .0779 | .0789 |
| | 192 | .1479 | .1499 | .1544 | .1605 | .1519 | .1550 | .1562 | .1597 | .1499 | .1548 | .1564 | .1605 |
| | 336 | .2624 | .2694 | .2700 | .2797 | .2480 | .2512 | .2541 | .2545 | .2461 | .2556 | .2556 | .2587 |
| | 720 | .4460 | .4494 | .4491 | .4728 | .4481 | .4486 | .4470 | .4735 | .4458 | .4478 | .4440 | .4735 |
| Weather | 96 | .1617 | .1614 | .1612 | .1616 | .1793 | .1793 | .1793 | .1807 | .1737 | .1739 | .1739 | .1733 |
| | 192 | .2088 | .2106 | .2080 | .2126 | .2217 | .2196 | .2158 | .2180 | .2189 | .2161 | .2157 | .2169 |
| | 336 | .2515 | .2543 | .2592 | .2652 | .2626 | .2613 | .2611 | .2690 | .2587 | .2614 | .2608 | .2742 |
| | 720 | .2730 | .2709 | .2754 | .2799 | .2708 | .2700 | .2698 | .2772 | .2692 | .2660 | .2665 | .2723 |

(b) Overhead analysis.

| Method | # Params ↓ | Adaptation time/batch (ms) ↓ | Inference time/batch (ms) ↓ | Average MSE ↓ |
|---|---|---|---|---|
| COSA | **1,211,287** | **80.12 ± 13.58** | **1.25 ± 0.0984** | **.3240** |
| RNN | 1,317,239 | 126.52 ± 22.88 | 1.66 ± 0.2315 | .3254 |
| LSTM | 1,466,615 | 134.29 ± 25.67 | 1.74 ± 0.2778 | .3260 |
| Attention | 1,647,735 | 175.72 ± 16.88 | 1.96 ± 0.3183 | 3278 |

## G.3 ADAPTER ARCHITECTURE COMPARISON

This experiment was conducted to compare the performance and computational efficiency of single linear adapters versus 2-layer MLP adapters with 64 hidden dimensions. We aimed to determine whether more complex architectures necessarily guarantee better performance. Table 16 provides a comprehensive performance and efficiency comparison between linear and MLP adapters.

Table 15: COSA with input GCM.

| | | Transformer-based | | | | Linear-based | | | | MLP-based | | | |
| | | iTransformer | | PatchTST | | DLinear | | OLS | | FreTS | | MICN | |
| | | COSA | w. Input | COSA | w. Input | COSA | w. Input | COSA | w. Input | COSA | w. Input | COSA | w. Input |
|---|---|---|---|---|---|---|---|---|---|---|---|---|---|
| ETTh1 | 96 | .4363 | **.4362** | .4238 | **.4209** | .4482 | **.4460** | .4372 | **.4370** | .4371 | **.4343** | **.4684** | .4852 |
| | 192 | .4919 | **.4821** | .4805 | **.4790** | .5050 | **.4995** | **.4906** | .4907 | .4940 | **.4919** | **.5328** | .5530 |
| | 336 | **.5300** | .5386 | .5320 | **.5216** | .5456 | **.5359** | .5320 | **.5193** | .5351 | **.5286** | **.5878** | .5953 |
| | 720 | **.5638** | .6287 | **.5822** | .6386 | **.5896** | .6513 | **.5733** | .6500 | **.5959** | .6726 | **.6504** | .7509 |
| ETTh2 | 96 | .2493 | **.2001** | .2343 | **.1845** | .2281 | **.1819** | .2265 | **.1803** | .2350 | **.1849** | .2485 | **.1995** |
| | 192 | .2947 | **.2384** | .2608 | **.2287** | .2819 | **.2217** | .2791 | **.2292** | .2824 | **.2241** | .3017 | **.2457** |
| | 336 | .3339 | **.2528** | .2978 | **.2405** | .3083 | **.2506** | .3043 | **.2474** | .3153 | **.2539** | .3310 | **.2755** |
| | 720 | .3591 | **.3048** | .3428 | **.2913** | .3477 | **.2932** | .3453 | **.2940** | .3399 | **.2954** | .3885 | **.3411** |
| ETTm1 | 96 | **.3455** | .3676 | **.3626** | .3903 | **.3475** | .3518 | **.3475** | .3516 | **.3525** | .3596 | **.3831** | .4216 |
| | 192 | **.4140** | .4273 | **.4258** | .4396 | **.4122** | .4194 | **.4119** | .4192 | .4212 | **.4200** | **.4514** | .4742 |
| | 336 | **.4643** | .4963 | **.4697** | .4881 | **.4858** | .4924 | **.4749** | .4926 | **.4775** | .4885 | **.5054** | .5365 |
| | 720 | **.5102** | .5869 | **.4882** | .5317 | **.4991** | .5678 | **.5007** | .5598 | **.4982** | .5452 | **.5225** | .5920 |
| ETTm2 | 96 | .1632 | **.1242** | .1562 | **.1200** | .1586 | **.1217** | .1586 | **.1218** | .1569 | **.1203** | .1704 | **.1288** |
| | 192 | .2173 | **.1673** | .2022 | **.1551** | .1905 | **.1502** | .1907 | **.1501** | .1908 | **.1491** | .2120 | **.1618** |
| | 336 | .2535 | **.2017** | .2352 | **.1900** | .2242 | **.1813** | .2226 | **.1831** | .2211 | **.1823** | .2351 | **.1975** |
| | 720 | .2606 | **.2554** | .2645 | **.2468** | **.2316** | .2465 | **.2349** | .2531 | **.2314** | .2364 | .2643 | **.2635** |
| Exchange Rate | 96 | **.0837** | .0859 | **.0788** | .0835 | **.0834** | .0883 | **.0773** | .0789 | **.0766** | .0798 | **.1008** | .1105 |
| | 192 | **.1479** | .1706 | **.1570** | .1754 | **.1519** | .1807 | **.1457** | .1644 | **.1499** | .1660 | **.1722** | .2083 |
| | 336 | **.2624** | .2974 | **.2445** | .3017 | **.2480** | .2986 | **.2323** | .2937 | **.2461** | .2951 | **.2660** | .3358 |
| | 720 | **.4460** | .5884 | **.4662** | .5969 | **.4481** | .6107 | **.4541** | .5837 | **.4458** | .5823 | **.4815** | .6885 |
| Weather | 96 | **.1617** | .1723 | **.1634** | .1728 | **.1793** | .1924 | **.1803** | .1927 | **.1737** | .1822 | **.1651** | .1741 |
| | 192 | **.2088** | .2214 | **.2108** | .2181 | **.2217** | .2363 | **.2237** | .2367 | **.2189** | .2269 | **.2120** | .2289 |
| | 336 | **.2515** | .2752 | **.2488** | .2809 | **.2626** | .2951 | **.2642** | .2946 | **.2587** | .2845 | **.2737** | .2728 |
| | 720 | **.2730** | .3274 | **.2713** | .3316 | **.2708** | .3356 | **.2708** | .3361 | **.2692** | .3298 | **.2855** | .3364 |

Table 16: Prediction accuracy comparison of single linear adapter and 2-layer MLP adapter.

| | | Transformer-based | | | | Linear-based | | | | MLP-based | | | | Transformer-based | | | | Linear-based | | | | MLP-based | | | |
| | | iTransformer | | PatchTST | | DLinear | | OLS | | FreTS | | MICN | | iTransformer | | PatchTST | | DLinear | | OLS | | FreTS | | MICN | |
| | | Linear | MLP | Linear | MLP | Linear | MLP | Linear | MLP | Linear | MLP | Linear | MLP | Linear | MLP | Linear | MLP | Linear | MLP | Linear | MLP | Linear | MLP | Linear | MLP |
|---|---|---|---|---|---|---|---|---|---|---|---|---|---|---|---|---|---|---|---|---|---|---|---|---|---|---|---|
| ETTh1 | 96 | **.4363** | .4376 | **.4238** | .4267 | **.4482** | .4644 | **.4372** | .4460 | **.4371** | .4412 | **.4684** | .4841 | **2.35** | 3.79 | **2.35** | 3.83 | **2.24** | 3.45 | **2.15** | 3.10 | **2.14** | 3.49 | **2.38** | 3.84 |
| | 192 | **.4919** | .4968 | **.4805** | .4857 | **.5050** | .5135 | **.4906** | .4980 | **.4940** | .4967 | **.5328** | .6067 | **2.39** | 3.73 | **2.44** | 3.83 | **2.23** | 3.49 | **2.21** | 3.42 | **2.16** | 3.47 | **2.59** | 3.83 |
| | 336 | **.5300** | .5471 | .5320 | **.5294** | **.5456** | .5501 | .5320 | **.5282** | .5351 | **.5346** | **.5878** | .6067 | **2.40** | 3.68 | **2.43** | 3.77 | **2.22** | 3.53 | **2.18** | 3.45 | **2.32** | 3.40 | **2.51** | 3.91 |
| | 720 | **.5638** | .6167 | **.5822** | .6350 | **.5896** | .6498 | **.5733** | .6351 | **.5959** | .6558 | **.6504** | .7325 | **2.39** | 3.46 | **2.25** | 3.49 | **2.17** | 3.10 | **2.23** | 3.22 | **2.24** | 3.11 | **3.69** | 4.57 |
| ETTh2 | 96 | **.2493** | .2517 | .2343 | **.2339** | **.2281** | .2299 | **.2265** | .2269 | **.2350** | .2350 | **.2485** | .2517 | **2.38** | 3.76 | **2.36** | 3.63 | **2.19** | 3.49 | **2.19** | 3.43 | **1.54** | 3.47 | **2.37** | 3.81 |
| | 192 | **.2947** | .2981 | **.2608** | .2825 | **.2819** | .2828 | .2791 | **.2767** | **.2824** | .3094 | **.3017** | .3485 | **2.40** | 3.74 | **2.44** | 3.80 | **2.28** | 3.51 | **2.20** | 3.41 | **2.24** | 3.43 | **2.66** | 3.85 |
| | 336 | .3339 | **.3306** | **.2978** | .3127 | **.3083** | .3106 | **.3043** | .3113 | **.3153** | .3168 | **.3310** | .3484 | **2.48** | 3.71 | **2.45** | 3.77 | **2.25** | 3.49 | **2.17** | 3.38 | **2.30** | 3.47 | **2.87** | 3.95 |
| | 720 | **.3591** | .3931 | **.3428** | .3750 | **.3477** | .3689 | **.3453** | .3750 | **.3399** | .3660 | **.3885** | .4170 | **2.34** | 3.40 | **2.35** | 3.48 | **2.21** | 3.33 | **2.18** | 3.24 | **2.19** | 3.19 | **3.72** | 4.62 |
| ETTm1 | 96 | **.3455** | .3522 | **.3626** | .3680 | **.3475** | .3505 | **.3475** | .3489 | **.3525** | .3577 | **.3831** | .3914 | **8.49** | 14.08 | **8.56** | 14.24 | **8.14** | 11.50 | **8.17** | 13.16 | **8.22** | 13.22 | **8.33** | 14.38 |
| | 192 | .4140 | **.4103** | .4258 | **.4246** | **.4122** | .4147 | **.4119** | .4137 | .4212 | **.4168** | .4514 | **.4470** | **8.72** | 14.41 | **8.67** | 14.31 | **8.35** | 13.25 | **5.68** | 13.38 | **8.57** | 13.68 | **9.27** | 14.39 |
| | 336 | .4643 | **.4625** | **.4697** | .4709 | .4858 | **.4678** | .4749 | **.4680** | .4775 | **.4708** | .5054 | **.4784** | **9.07** | 14.62 | **9.03** | 12.04 | **8.72** | 13.59 | **8.86** | 13.31 | **8.87** | 14.17 | **10.86** | 15.26 |
| | 720 | .5102 | **.4927** | .4882 | **.4651** | .4991 | **.4763** | .5007 | **.4774** | .4982 | **.4888** | **.5225** | .5483 | **9.64** | 15.32 | **8.72** | 12.74 | **9.32** | 14.00 | **9.36** | 14.31 | **9.60** | 14.62 | **16.05** | 20.23 |
| ETTm2 | 96 | **.1632** | .1632 | **.1562** | .1569 | .1586 | **.1582** | **.1586** | .1587 | **.1569** | .1571 | .1704 | **.1703** | **8.51** | 11.73 | **8.72** | 14.41 | **8.34** | 13.23 | **8.06** | 13.04 | **8.45** | 9.13 | **8.25** | 14.35 |
| | 192 | .2173 | **.2140** | .2022 | **.1978** | .1905 | **.1880** | .1907 | **.1884** | .1908 | **.1850** | .2120 | **.2039** | **8.75** | 14.35 | **8.77** | 14.38 | **8.32** | 13.31 | **5.27** | 13.28 | **8.51** | 13.65 | **9.32** | 14.42 |
| | 336 | .2535 | **.2412** | .2352 | **.2283** | .2242 | **.2103** | .2226 | **.2142** | .2211 | **.2092** | .2351 | **.2331** | **8.10** | 14.64 | **9.10** | 14.56 | **8.84** | 13.50 | **8.89** | 13.57 | **8.86** | 13.64 | **10.80** | 15.24 |
| | 720 | **.2606** | .2759 | **.2645** | .2711 | **.2316** | .2532 | **.2349** | .2551 | **.2314** | .2477 | **.2643** | .2709 | **7.74** | 15.14 | **9.51** | 15.04 | **9.29** | 13.86 | **9.32** | 14.26 | **9.50** | 14.48 | **16.04** | 20.21 |
| Exchange Rate | 96 | **.0837** | .0869 | **.0788** | .0838 | **.0834** | .0890 | **.0773** | .0789 | **.0766** | .0803 | **.1008** | .1106 | **1.38** | 2.01 | **1.38** | 1.99 | **1.22** | 1.79 | **1.20** | 1.77 | **1.26** | 1.81 | **1.39** | 2.00 |
| | 192 | **.1479** | .1730 | **.1570** | .1696 | **.1519** | .1570 | **.1457** | .1570 | **.1499** | .1603 | **.1722** | .1956 | **1.31** | 1.89 | **1.36** | 1.88 | **1.17** | 1.68 | **1.15** | 1.69 | **1.17** | 1.70 | **1.42** | 1.96 |
| | 336 | **.2624** | .3103 | **.2445** | .2941 | **.2480** | .2872 | **.2323** | .2856 | **.2461** | .2910 | **.2660** | .3399 | **1.26** | 1.70 | **1.32** | 1.81 | **1.09** | 1.57 | **1.10** | 1.58 | **1.10** | 1.62 | **1.48** | 1.60 |
| | 720 | **.4460** | .8099 | **.4662** | .8268 | **.4481** | .8373 | **.4541** | .7980 | **.4458** | .7983 | **.4815** | .9665 | **1.09** | 1.47 | **1.09** | 1.43 | **.91** | 1.24 | **.90** | 1.26 | **.93** | 1.27 | **1.46** | 1.76 |
| Weather | 96 | **.1617** | .1691 | **.1634** | .1691 | **.1793** | .1904 | **.1803** | .1904 | **.1737** | .1767 | **.1651** | .1703 | **13.05** | 20.86 | **15.19** | 23.35 | **12.81** | 20.34 | **12.76** | 18.41 | **12.84** | 20.90 | **12.88** | 20.56 |
| | 192 | .2088 | **.2045** | .2108 | **.2026** | **.2217** | .2285 | .2237 | **.2224** | .2189 | **.2120** | .2120 | **.2070** | **13.33** | 20.79 | **15.33** | 23.72 | **13.10** | 20.43 | **12.89** | 20.55 | **13.12** | 20.72 | **13.92** | 20.44 |
| | 336 | .2515 | **.2390** | .2488 | **.2360** | .2626 | **.2469** | .2642 | **.2499** | .2587 | **.2440** | .2737 | **.2386** | **13.69** | 21.12 | **15.93** | 24.21 | **13.49** | 20.83 | **13.43** | 20.98 | **13.53** | 20.96 | **14.31** | 22.14 |
| | 720 | **.2730** | .2752 | **.2713** | .2764 | **.2708** | .2785 | **.2708** | .2785 | **.2692** | .2737 | .2855 | **.2582** | **14.68** | 17.06 | **16.71** | 25.44 | **14.05** | 21.26 | **14.36** | 21.61 | **14.41** | 21.81 | **18.09** | 26.57 |
| Average | | **.3218** | .3438 | **.3166** | .3383 | **.3196** | .3425 | **.3158** | .3368 | **.3176** | .3369 | **.3421** | .3696 | **6.16** | 9.60 | **6.60** | 10.21 | **6.04** | 9.28 | **5.79** | 9.28 | **6.09** | 9.35 | **7.36** | 10.75 |

The comparative analysis reveals that single linear adapters achieve performance comparable to or superior to 2-layer MLP adapters while requiring significantly reduced computational resources. Although MLP adapters occasionally demonstrated slight performance improvements, the 1.5∼2× computational overhead renders them impractical for real-time adaptation scenarios. These results validate the architectural design principle of COSA that emphasizes simplicity without compromising effectiveness.

### G.4 COMPARISON WITH VARIOUS LEARNING-RATE SCHEDULERS

To validate the effectiveness of CALR, we compared it against several official PyTorch learning-rate schedulers: CosineAnnealingLR, ExponentialLR, ReduceLROnPlateau, StepLR, and a fixed learning rate. All schedulers were configured with the same base learning rate of 0.005 for a fair comparison. One-Cycle, although widely used, was excluded because it requires a predefined learning-rate schedule; in a streaming TTA scenario where samples arrive continuously and the batch size changes dynamically under PAAS, such predefinition is not feasible.

As summarized in Table 17, each scheduler achieves improvements in some individual cases. However, when results are averaged across all datasets and prediction lengths, the proposed CALR achieves the best or second-best accuracy in the vast majority of settings. These findings support CALR's stability-induced design and its suitability for non-stationary TTA environments.

### G.5 EXTENTION TO MULTIVARIATE TIME-SERIES FORECASTING

COSA is originally introduced as an output-space residual correction module, treating each variable as an independent univariate forecasting task, for fair comparison with existing SOTA methods that assume univariate forecasting. However, in real multivariate time-series settings, correlations among variables may influence drift patterns, suggesting that modeling cross-variable interactions could potentially benefit COSA. Basically, COSA can be extended to multivariate forecasting; to examine this possibility, we implemented it.

We first incorporate *Cross-Variable Context Attention*, allowing the context of each variable to reference information from other variables and thereby capture correlation-driven contextual interactions. Additionally, we introduce a mixed structure composed of a low-rank shared component and variable-specific components: the shared component captures drift patterns common across variables in an efficient low-dimensional form, while the specific components model idiosyncratic behavior unique to each variable. These two components are combined through a learnable mixing coefficient that automatically balances global and variable-specific contributions.

Table 18a presents the comparison results. For datasets where meaningful cross-variable dependencies exist, the multivariate structure achieves higher predictive accuracy than the univariate version of COSA. However, for datasets with weak inter-variable correlations, such as the Exchange Rate, the univariate structure remains more stable. In such cases, the shared component struggles to learn useful common patterns, which can lead to performance degradation. Moreover, as shown in Table 18b, the additional complexity leads to increased adaptation time and inference latency.

These results indicate that multivariate-based correlation modeling can indeed provide accuracy gains, but further design improvements are required for effective deployment under TTA constraints. We discuss these limitations and potential future directions in Section H.

### G.6 EXTENSION TO VECTOR GATING

To evaluate whether finer-grained control over correction strength could provide additional benefits, we implemented an element-wise gating vector as an extension of the scalar gating mechanism in COSA. This vector shares the same dimensionality as the prediction length, allowing each time step within the prediction window to modulate its correction intensity independently. Such a design is intended to handle scenarios where drift occurs in only a specific portion of the horizon.

However, as shown in Table 19a, vector gating yields degraded overall accuracy compared to the original scalar gating, and also exhibits reduced stability. We attribute this performance degradation to the propagation of local noise: a noise spike at a particular horizon position can influence the entire gating vector over successive adaptation steps, causing cumulative negative impact throughout the correction process. In contrast, scalar gating provides consistent batch-level modulation that effectively bounds the influence of noise and maintains stable behavior across adaptation windows.

### G.7 VISUALIZATION OF NORMALIZATION

Figure 7 visualizes the learned weights of the linear layer in COSA with and without representative time-series normalization modules (RevIN (Kim et al., 2021) and DDN (Dai et al., 2024)).

Table 17: Prediction accuracy comparison with various learning rate schedulers.

| | | Transformer-based iTransformer | | | | | | Linear-based DLinear | | | | | | MLP-based FreTS | | | | | |
|---|---|---|---|---|---|---|---|---|---|---|---|---|---|---|---|---|---|---|---|
| | | CALR | Cosine | Exp | Fixed | Plateau | Step | CALR | Cosine | Exp | Fixed | Plateau | Step | CALR | Cosine | Exp | Fixed | Plateau | Step |
| ETTh1 | 96 | **.4363** | .4858 | .4961 | .4848 | .4848 | .4885 | **.4482** | .4498 | .4556 | .4513 | .4513 | .4537 | **.4371** | .4400 | .4441 | .4398 | .4398 | .4405 |
| | 192 | **.4919** | .5405 | .5613 | .5413 | .5413 | .5426 | **.5050** | .5069 | .5183 | .5090 | .5090 | .5110 | .4940 | **.4914** | .5002 | .4915 | .4915 | .4931 |
| | 336 | **.5300** | .5741 | .5952 | .5751 | .5751 | .5826 | **.5456** | .5560 | .5714 | .5586 | .5586 | .5642 | **.5351** | .5429 | .5570 | .5442 | .5442 | .5494 |
| | 720 | **.5638** | .6216 | .6835 | .6163 | .6163 | .6336 | **.5896** | .6033 | .6511 | .5953 | .5953 | .6166 | **.5959** | .6116 | .6593 | .6041 | .6041 | .6200 |
| ETTh2 | 96 | **.2493** | .2984 | .3021 | .2984 | .2984 | .2987 | **.2281** | .2308 | .2287 | .2321 | .2321 | .2333 | **.2350** | .2378 | .2359 | .2376 | .2376 | .2372 |
| | 192 | **.2947** | .3386 | .3520 | .3399 | .3400 | .3394 | .2819 | .2823 | **.2811** | .2836 | .2836 | .2845 | **.2824** | .2837 | .2827 | .2834 | .2834 | .2831 |
| | 336 | **.3339** | .3823 | .3992 | .3856 | .3856 | .3802 | .3083 | .3078 | .3093 | .3104 | .3105 | **.3081** | .3153 | .3115 | .3151 | .3131 | .3131 | **.3087** |
| | 720 | **.3591** | .4094 | .4278 | .4113 | .4113 | .4070 | .3477 | .3490 | .3562 | **.3462** | .3463 | .3488 | .3399 | .3421 | .3524 | .3391 | .3389 | **.3387** |
| ETTm1 | 96 | **.3455** | .4006 | .4087 | .3978 | .3978 | .4048 | **.3475** | .3501 | .3610 | .3497 | .3497 | .3569 | **.3525** | .3548 | .3567 | .3539 | .3539 | .3555 |
| | 192 | **.4140** | .4650 | .4772 | .4630 | .4630 | .4683 | **.4122** | .4140 | .4283 | .4146 | .4146 | .4191 | **.4212** | .4236 | .4279 | .4231 | .4230 | .4246 |
| | 336 | **.4643** | .5110 | .5286 | .5100 | .5100 | .5160 | .4858 | **.4828** | .5028 | .4838 | .4838 | .4875 | .4775 | **.4740** | .4821 | .4740 | .4740 | .4750 |
| | 720 | **.5102** | .5321 | .5732 | .5350 | .5349 | .5392 | .4991 | **.4861** | .5153 | .4889 | .4889 | .4915 | .4982 | **.4923** | .5073 | .4950 | .4950 | .4951 |
| ETTm2 | 96 | **.1632** | .2138 | .2140 | .2139 | .2139 | .2136 | **.1586** | .1621 | .1591 | .1629 | .1629 | .1646 | **.1569** | .1611 | .1572 | .1604 | .1604 | .1606 |
| | 192 | **.2173** | .2697 | .2705 | .2692 | .2693 | .2704 | **.1905** | .1948 | .1916 | .1955 | .1956 | .1973 | **.1908** | .1958 | .1921 | .1951 | .1951 | .1955 |
| | 336 | **.2535** | .3028 | .3125 | .3047 | .3045 | .3065 | **.2242** | .2305 | .2268 | .2310 | .2310 | .2324 | **.2211** | .2285 | .2247 | .2276 | .2276 | .2278 |
| | 720 | **.2606** | .3000 | .3159 | .2987 | .2986 | .2966 | **.2316** | .2369 | .2383 | .2405 | .2405 | .2348 | **.2314** | .2388 | .2397 | .2405 | .2403 | .2334 |
| Exchange Rate | 96 | **.0837** | .1335 | .1344 | .1335 | .1335 | .1334 | **.0834** | .0865 | .0848 | .0873 | .0873 | .0886 | **.0766** | .0800 | .0775 | .0794 | .0794 | .0792 |
| | 192 | **.1479** | .1961 | .2025 | .1960 | .1960 | .1952 | **.1519** | .1523 | .1552 | .1533 | .1533 | .1542 | **.1499** | .1518 | .1545 | .1513 | .1513 | .1505 |
| | 336 | **.2624** | .3120 | .3495 | .3116 | .3117 | .3102 | **.2480** | .2521 | .2710 | .2528 | .2527 | .2535 | **.2461** | .2501 | .2735 | .2491 | .2492 | .2484 |
| | 720 | **.4460** | .4481 | .6222 | .4498 | .4460 | .4448 | .4481 | .4185 | .5919 | .4164 | **.4162** | .4179 | .4458 | .4128 | .5679 | .4072 | **.4063** | .4068 |
| Weather | 96 | **.1617** | .2118 | .2151 | .2115 | .2115 | .2123 | **.1793** | .1824 | .1827 | .1827 | .1827 | .1858 | **.1737** | .1770 | .1767 | .1761 | .1761 | .1776 |
| | 192 | **.2088** | .2514 | .2592 | .2517 | .2517 | .2517 | **.2217** | .2226 | .2270 | .2232 | .2232 | .2259 | **.2189** | .2201 | .2235 | .2191 | .2192 | .2204 |
| | 336 | **.2515** | .2916 | .3063 | .2949 | .2948 | .2920 | .2626 | **.2510** | .2622 | .2532 | .2532 | .2541 | .2587 | .2497 | .2606 | .2505 | .2505 | **.2496** |
| | 720 | **.2730** | .3230 | .3460 | .3225 | .3225 | .3267 | **.2708** | .2774 | .2965 | .2802 | .2802 | .2810 | **.2692** | .2726 | .2921 | .2755 | .2755 | .2725 |

| | | PatchTST | | | | | | OLS | | | | | | MICN | | | | | |
|---|---|---|---|---|---|---|---|---|---|---|---|---|---|---|---|---|---|---|---|
| | | CALR | Cosine | Exp | Fixed | Plateau | Step | CALR | Cosine | Exp | Fixed | Plateau | Step | CALR | Cosine | Exp | Fixed | Plateau | Step |
| ETTh1 | 96 | .4238 | .4723 | .4294 | .5723 | .4923 | **.4232** | **.4372** | .4375 | .4448 | .4401 | .4401 | .4413 | **.4684** | .4739 | .4904 | .4708 | .4708 | .4769 |
| | 192 | **.4805** | .5307 | .4958 | .6301 | .5501 | .4838 | **.4906** | .4912 | .5041 | .4944 | .4944 | .4951 | .5328 | .5320 | .5709 | **.5297** | .5298 | .5390 |
| | 336 | **.5320** | .5851 | .5535 | .6826 | .6026 | .5396 | **.5320** | .5342 | .5515 | .5384 | .5384 | .5424 | .5878 | .5864 | .6417 | **.5861** | .5861 | .5974 |
| | 720 | **.5822** | .6412 | .6553 | .7330 | .6533 | .6031 | **.5733** | .5872 | .6371 | .5808 | .5808 | .5987 | **.6504** | .6726 | .8268 | .6506 | .6506 | .6977 |
| ETTh2 | 96 | **.2343** | .2840 | .2352 | .3839 | .3039 | .2345 | **.2265** | .2275 | .2268 | .2301 | .2300 | .2296 | **.2485** | .2499 | .2499 | .2521 | .2521 | .2466 |
| | 192 | **.2608** | .3114 | .2641 | .4092 | .3292 | .2617 | **.2791** | .2804 | .2804 | .2828 | .2828 | .2823 | .3017 | .2990 | .3077 | .3025 | .3026 | **.2960** |
| | 336 | **.2978** | .3514 | .3086 | .4543 | .3743 | .2993 | .3043 | .3030 | .3064 | .3063 | .3063 | **.3029** | .3310 | .3303 | .3399 | .3388 | .3388 | **.3243** |
| | 720 | **.3428** | .3913 | .3663 | .4951 | .4151 | .3485 | **.3453** | .3487 | .3589 | .3473 | .3472 | .3476 | **.3885** | .3965 | .4019 | .3929 | .3929 | .3884 |
| ETTm1 | 96 | **.3626** | .4104 | .3741 | .5183 | .4383 | .3655 | **.3475** | .3487 | .3608 | .3495 | .3495 | .3552 | **.3831** | .3912 | .3982 | .3900 | .3900 | .3913 |
| | 192 | **.4258** | .4747 | .4427 | .5764 | .4964 | .4315 | **.4119** | .4125 | .4283 | .4142 | .4142 | .4174 | **.4514** | .4533 | .4641 | .4524 | .4524 | .4541 |
| | 336 | **.4697** | .5149 | .4825 | .6203 | .5403 | .4700 | **.4706** | .4728 | .5136 | .4873 | .4873 | .4882 | **.5054** | .5053 | .5249 | .5048 | .5048 | .5079 |
| | 720 | **.4882** | .5349 | .5077 | .6391 | .5550 | .4877 | .5007 | **.4831** | .5136 | .4873 | .4873 | .4882 | .5225 | **.5152** | .5443 | .5188 | .5188 | .5185 |
| ETTm2 | 96 | **.1562** | .2065 | .1566 | .3064 | .2265 | .1566 | **.1586** | .1608 | .1590 | .1627 | .1627 | .1630 | **.1704** | .1742 | .1709 | .1762 | .1763 | .1709 |
| | 192 | **.2022** | .2527 | .2056 | .3532 | .2732 | .2034 | **.1907** | .1933 | .1914 | .1954 | .1953 | .1956 | **.2120** | .2154 | .2138 | .2180 | .2180 | .2122 |
| | 336 | **.2352** | .2863 | .2367 | .3863 | .3063 | .2363 | **.2226** | .2278 | .2256 | .2298 | .2298 | .2298 | **.2351** | .2383 | .2365 | .2408 | .2408 | .2354 |
| | 720 | **.2645** | .3145 | .2692 | .4114 | .3314 | .2619 | **.2349** | .2390 | .2417 | .2445 | .2445 | .2367 | .2643 | .2628 | .2672 | .2647 | .2647 | **.2565** |
| Exchange Rate | 96 | .0788 | .1278 | .0802 | .2278 | .1478 | **.0776** | **.0773** | .0789 | .0780 | .0809 | .0809 | .0808 | .1008 | .1031 | .1027 | .1051 | .1051 | **.0994** |
| | 192 | .1570 | .2051 | .1642 | .3052 | .2251 | **.1544** | .1457 | **.1452** | .1473 | .1466 | .1466 | .1473 | .1715 | .1779 | .1735 | .1735 | .1735 | **.1670** |
| | 336 | **.2445** | .2966 | .2780 | .3962 | .3164 | .2448 | **.2323** | .2331 | .2578 | .2349 | .2348 | .2334 | **.2660** | .2730 | .3114 | .2745 | .2745 | .2675 |
| | 720 | **.4662** | .4743 | .5963 | .5724 | .4923 | .4218 | .4541 | .4224 | .5806 | .4190 | .4190 | **.4189** | .4815 | .4515 | .6597 | .4488 | .4511 | **.4460** |
| Weather | 96 | .1634 | .2124 | .1651 | .3124 | .2324 | **.1629** | **.1803** | .1822 | .1840 | .1838 | .1838 | .1855 | .1651 | .1678 | .1669 | .1695 | .1695 | **.1644** |
| | 192 | .2108 | .2564 | .2122 | .3569 | .2769 | **.2065** | .2237 | **.2232** | .2290 | .2250 | .2250 | .2263 | .2120 | .2096 | .2105 | .2109 | .2109 | **.2050** |
| | 336 | .2488 | .2924 | .2540 | .3950 | .3150 | **.2419** | .2642 | **.2618** | .2743 | .2651 | .2651 | .2647 | .2737 | .2545 | .2599 | .2556 | .2556 | **.2499** |
| | 720 | **.2713** | .3230 | .2913 | .4275 | .3475 | .2737 | **.2708** | .2783 | .2987 | .2824 | .2824 | .2817 | **.2855** | .2864 | .3035 | .2959 | .2959 | .2906 |

RevIN performs standard normalization on each input time series and then applies a corresponding denormalization step on the output. This procedure mitigates train–test distribution mismatch while restoring the information removed during normalization at the prediction stage, preventing degradation in forecasting performance.

DDN, in contrast, operates jointly in the time and frequency domains. It decomposes the input into low-frequency and high-frequency components and computes local statistics from each domain to remove non-stationarity. DDN then reconstructs non-stationary patterns in the predicted outputs using distribution statistics estimated from the model's predictions, enabling dynamic tracking of distribution drift.

Table 18: Prediction accuracy and overhead of multivariate consideration.

(a) Prediction accuracy.

| | | Transformer-based | | | | Linear-based | | | | MLP-based | | | |
|---|---|---|---|---|---|---|---|---|---|---|---|---|---|
| | | iTransformer | | PatchTST | | DLinear | | OLS | | FreTS | | MICN | |
| | | Indiv. | Corr. | Indiv. | Corr. | Indiv. | Corr. | Indiv. | Corr. | Indiv. | Corr. | Indiv. | Corr. |
| ETTh1 | 96 | .4363 | **.4351** | .4238 | **.4157** | .4482 | **.4202** | **.4372** | .4388 | **.4371** | .4408 | .4684 | **.4533** |
| | 192 | .4919 | **.4762** | .4805 | **.4589** | .5050 | **.4646** | .4906 | **.4806** | .4940 | **.4769** | .5328 | **.5039** |
| | 336 | .5300 | **.4759** | .5320 | **.4798** | .5456 | **.4823** | .5320 | **.4884** | .5351 | **.4987** | .5878 | **.5105** |
| | 720 | .5638 | **.4371** | .5822 | **.5244** | .5896 | **.4695** | .5733 | **.4660** | .5959 | **.4954** | .6504 | **.5314** |
| ETTh2 | 96 | .2493 | **.2453** | .2343 | **.1836** | **.2281** | .2342 | .2265 | **.2249** | .2350 | **.2311** | .2485 | **.2411** |
| | 192 | .2947 | **.2871** | .2608 | **.2172** | .2819 | **.2578** | .2791 | **.2770** | .2824 | **.2922** | .3017 | **.2923** |
| | 336 | **.3339** | .3341 | .2978 | **.2361** | .3083 | **.2866** | .3043 | **.2911** | .3153 | **.2971** | .3310 | **.3122** |
| | 720 | .3591 | **.3306** | .3428 | **.2638** | .3477 | **.3094** | .3453 | **.3083** | .3399 | **.2979** | .3885 | **.3441** |
| ETTm1 | 96 | .3455 | **.3186** | .3626 | **.3595** | .3475 | **.3335** | .3475 | **.3330** | .3525 | **.3381** | .3831 | **.3385** |
| | 192 | .4140 | **.3988** | .4258 | **.4159** | .4122 | **.4058** | .4119 | **.4125** | .4212 | **.4153** | .4514 | **.4188** |
| | 336 | .4643 | **.4491** | **.4697** | .4739 | .4858 | **.4561** | .4749 | **.4648** | .4775 | **.4618** | .5054 | **.4628** |
| | 720 | .5102 | **.4372** | **.4882** | .4892 | .4991 | **.4406** | .5007 | **.4323** | .4982 | **.4533** | .5225 | **.4496** |
| ETTm2 | 96 | **.1632** | .1633 | .1562 | **.1195** | .1586 | **.1557** | .1586 | **.1574** | .1569 | **.1560** | .1704 | **.1697** |
| | 192 | .2173 | **.2141** | .2022 | **.1526** | **.1905** | .2005 | .1907 | **.1935** | .1908 | **.1921** | .2120 | **.2097** |
| | 336 | .2535 | **.2347** | .2352 | **.1783** | .2242 | **.2235** | .2226 | **.2145** | .2211 | **.2149** | .2351 | **.2429** |
| | 720 | .2606 | **.2110** | .2645 | **.2162** | .2316 | **.2295** | .2349 | **.2006** | .2314 | **.2022** | .2643 | **.2265** |
| Exchange Rate | 96 | **.0837** | .0840 | **.0788** | .0851 | .0834 | **.0791** | **.0773** | .0773 | .0766 | **.0765** | .1008 | **.0995** |
| | 192 | **.1479** | .1493 | **.1570** | .1828 | **.1519** | .1609 | **.1457** | .1457 | **.1499** | .1516 | **.1722** | .1726 |
| | 336 | **.2624** | .2838 | **.2445** | .3162 | **.2480** | .2599 | **.2323** | .2456 | **.2461** | .2627 | **.2660** | .2955 |
| | 720 | **.4460** | .5221 | **.4662** | .7731 | **.4481** | .5280 | **.4541** | .5184 | **.4458** | .5079 | **.4815** | .5711 |
| Weather | 96 | .1617 | **.1547** | **.1634** | .1656 | .1793 | **.1566** | .1803 | **.1731** | .1737 | **.1673** | .1651 | **.1591** |
| | 192 | .2088 | **.1938** | .2108 | **.2073** | .2217 | **.2003** | .2237 | **.2158** | .2189 | **.2095** | .2120 | **.1979** |
| | 336 | .2515 | **.2300** | **.2488** | .2539 | .2626 | **.2321** | .2642 | **.2461** | .2587 | **.2331** | .2737 | **.2565** |
| | 720 | .2730 | **.2236** | **.2713** | .2868 | .2708 | **.2254** | .2708 | **.2172** | .2692 | **.2165** | .2855 | **.2420** |

(b) Overhead analysis.

| Method | # Params ↓ | Adaptation time/batch (ms) ↓ | Inference time/batch (ms) ↓ | Average MSE ↓ |
|---|---|---|---|---|
| Univariate | **1,211,287** | **80.12 ± 13.58** | **1.25 ± .0984** | .3240 |
| Multivariate | 1,214,851 | 186.28 ± 15.36 | 6.35 ± .2452 | **.3071** |

Each heatmap entry $(i, j)$ shows the weight connecting the $j$-th input to the $i$-th output; columns 1:$L$ correspond to the original prediction of base model $\mathbf{Y}^{(0)}$ and columns $L+1$:$L+K$ to the context vector $\mathbf{C}$. The example is taken from a single variable of ETTh1 with look-back $W=96$ and horizon $L=96$. Notably, the rightmost block (context columns) is strongly attenuated when RevIN or DDN is applied, whereas without a normalizer, the same block exhibits structured, non-negligible weights. This pattern indicates that explicit normalization reduces the marginal utility of the context (level/scale cues are already standardized), while in the non-normalized COSA leverages $\mathbf{C}$ to perform level-shift correction directly in the output space—supporting our claim that the adapter can subsume normalization effects when needed and remain compatible with them when present.

# H  DISCUSSION

Although COSA is built around a simple output-space linear adapter, the extended experiments in the Appendix examined multiple alternative design choices. While low-rank adaptation, input-side calibration, vector gating, selective/encoder-based context, and multivariate extensions each provide potential benefits in specific scenarios, our overall findings show that, considering average accuracy, stability, and latency, the architecture adopted in this paper remains the most consistent and robust choice for TSF-TTA.

The key observations are summarized below:

Table 19: Prediction accuracy and overhead of vector gating.

(a) Performance comparison.

| | | iTransformer | | DLinear | | FreTS | |
|---|---|---|---|---|---|---|---|
| | | Scalar | Vector | Scalar | Vector | Scalar | Vector |
| ETTh1 | 96 | .4363 | **.4336** | .4574 | **.4436** | .4371 | **.4336** |
| | 192 | .4919 | **.4875** | .5066 | **.4985** | .4940 | **.4905** |
| | 336 | **.5300** | .5430 | .5528 | **.5466** | .5467 | **.5353** |
| | 720 | **.5638** | .6013 | **.6107** | .6178 | .6259 | **.6236** |
| ETTh2 | 96 | .2493 | **.1990** | .2281 | **.1798** | .2350 | **.1842** |
| | 192 | .2947 | **.2318** | .2819 | **.2198** | .2972 | **.2202** |
| | 336 | .3339 | **.2604** | .3083 | **.2472** | .3153 | **.2501** |
| | 720 | .3591 | **.3035** | .3477 | **.2893** | .3399 | **.2898** |
| ETTm1 | 96 | **.3455** | .3611 | .3456 | **.3425** | **.3525** | .3551 |
| | 192 | .4140 | **.4119** | .4222 | **.4068** | .4212 | **.4137** |
| | 336 | **.4643** | .4720 | .4858 | **.4706** | .4775 | **.4767** |
| | 720 | **.5102** | .5484 | **.4991** | .5370 | **.4982** | .5405 |
| ETTm2 | 96 | .1632 | **.1245** | .1583 | **.1215** | .1569 | **.1202** |
| | 192 | .2173 | **.1683** | .1943 | **.1487** | .1934 | **.1484** |
| | 336 | .2535 | **.2010** | .2242 | **.1798** | .2211 | **.1795** |
| | 720 | .2606 | **.2488** | **.2316** | .2319 | .2314 | **.2280** |
| Exchange Rate | 96 | **.0837** | .0875 | **.0834** | .0903 | **.0766** | .0818 |
| | 192 | **.1479** | .1774 | **.1519** | .1790 | **.1499** | .1698 |
| | 336 | **.2624** | .3258 | **.2480** | .3134 | **.2461** | .3094 |
| | 720 | **.4460** | .7649 | **.4481** | .7904 | **.4458** | .7500 |
| Weather | 96 | **.1617** | .1718 | **.1793** | .1908 | **.1737** | .1825 |
| | 192 | **.2088** | .2176 | **.2217** | .2326 | **.2189** | .2250 |
| | 336 | **.2515** | .2706 | **.2626** | .2808 | **.2587** | .2745 |
| | 720 | **.2730** | .3359 | **.2708** | .3418 | **.2692** | .3378 |

(b) Overhead analysis.

| Method | # Params ↓ | Adaptation time/batch (ms) ↓ | Inference time/batch (ms) ↓ | Average MSE ↓ |
|---|---|---|---|---|
| Scalar | **1,211,287** | **80.12 ± 13.58** | **1.25 ± .0984** | **.3240** |
| Vector | 1,212,446 | 96.34 ± 12.48 | 1.89 ± .0745 | .3287 |

- **Low-rank factorization.** Despite its parameter-efficiency appeal, reducing representational capacity can lead to unstable adaptation. A joint adapter that integrates low-rank structure without compromising stability is a meaningful direction for future work.

- **Input-side calibration (GCM).** Combining COSA with input GCM improves performance on datasets with strong input noise (e.g., ETTh1/h2/m2) by smoothing perturbations before prediction. However, for fast-drifting or irregular series, GCM may oversmooth important variations and degrade performance, reaffirming that output-only correction is a reasonable and stable default.

- **Gating and its variants.** Although vector gating was expected to modulate drift at a finer temporal resolution, local noise propagated across the gating vector and reduced stability compared to scalar gating. Scalar gating's batch-level modulation limits the influence of noise and achieves more reliable behavior.

- **Context construction.** Selective context (phase-aligned retrieval) and encoder-based context (RNN/LSTM/Attention) showed improvements in certain periodic datasets, but both suffered from outdated information, overfitting, or latency overhead in non-stationary settings. These results highlight that additional complexity does not guarantee better TTA performance unless paired with drift-aware representations and online update strategies. Future directions include sub-sequence vector gating, structure-aware context that explicitly

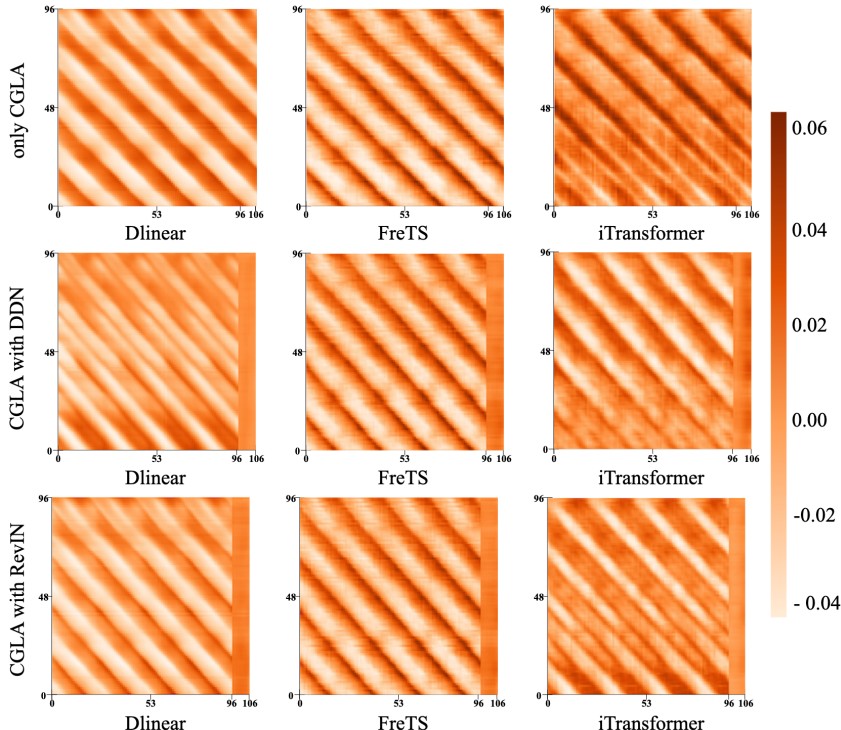

Figure 7: Weight heatmaps of the COSA linear layer for one ETTH1 variable ($W$=96, $L$=96). Columns 1:$L$ are base-prediction inputs $\mathbf{Y}^{(0)}$; columns $L$+1:$L$+$K$ are context inputs $C$. Each cell shows the weight from input $j$ to output $i$. Color scales are kept identical across panels to allow magnitude comparison.

      encodes trend and seasonality, and lightweight encoders capable of tracking drift without incurring high overhead.

- **Multivariate residual correction.** While multivariate modeling with cross-variable attention and shared components improved performance in most datasets, it degraded both accuracy and efficiency in datasets with weak inter-variable correlations (e.g., Exchange Rate). This suggests the need for selective correlation modeling or structural sparsity to suppress unnecessary cross-variable interactions.

Taken together, the extended Appendix experiments reinforce that the proposed simple architecture is particularly well-suited for TSF-TTA. They also indicate substantial room for generalizing COSA through carefully integrated low-rank structures, input calibration modules, selective or encoder-based context modeling, and vector gating, while preserving the efficiency and stability crucial for non-stationary test-time adaptation.

## I   CONFIDENCE INTERVAL OF MAIN RESULTS

Table 20 reports the 95% confidence intervals of the main accuracy comparison results over 10 independent runs for each method–dataset–horizon combination. Overall, the intervals are narrow, indicating that the run-to-run variability of all methods is small, and COSA-F/P consistently retain their advantage over baselines even when accounting for this uncertainty.

## THE USE OF LARGE LANGUAGE MODELS

**Tool & Version**: Claude Sonnet 4 (Anthropic, 2025-09)
**Research Stage**: Not used.

Table 20: 95% confidence interval of main accuracy comparision results

| | | Transformer-based | | | | | | | | Linear-based | | | | | | | | MLP-based | | | | | | | |
| --- | --- | --- | --- | --- | --- | --- | --- | --- | --- | --- | --- | --- | --- | --- | --- | --- | --- | --- | --- | --- | --- | --- | --- | --- | --- |
| | | iTransformer | | | | PatchTST | | | | DLinear | | | | OLS | | | | FreTS | | | | MICN | | | |
| | | TAFAS | PETSA | COSA-F | COSA-P | TAFAS | PETSA | COSA-F | COSA-P | TAFAS | PETSA | COSA-F | COSA-P | TAFAS | PETSA | COSA-F | COSA-P | TAFAS | PETSA | COSA-F | COSA-P | TAFAS | PETSA | COSA-F | COSA-P |
| ETTh1 | 96 | .4409~.4413 | .4390~.4396 | .4366~.4370 | .4361~.4365 | .4243~.4281 | .4247~.4291 | .4226~.4258 | .4222~.4254 | .4616~.4620 | .4592~.4596 | .4571~.4577 | .4571~.4577 | .4405~.4413 | .4385~.4397 | .4385~.4395 | .4367~.4377 | .4397~.4409 | .4381~.4393 | .4377~.4391 | .4365~.4377 | .4890~.4912 | .4886~.4910 | .4683~.4703 | .4682~.4704 |
| | 192 | .4916~.4940 | .4937~.4961 | .4948~.4974 | .4908~.4930 | .4830~.4900 | .4825~.4883 | .4805~.4855 | .4775~.4835 | .5088~.5146 | .5079~.5157 | .5029~.5103 | .5032~.5100 | .4920~.4948 | .4921~.4953 | .4901~.4929 | .4892~.4920 | .4918~.4990 | .4905~.4979 | .4922~.4980 | .4906~.4974 | .5615~.5619 | .5619~.5621 | .5370~.5374 | .5370~.5374 |
| | 336 | .5580~.5678 | .5586~.5694 | .5604~.5698 | .5248~.5352 | .5444~.5512 | .5441~.5509 | .5399~.5477 | .5288~.5352 | .5587~.5621 | .5601~.5633 | .5508~.5548 | .5511~.5545 | .5420~.5460 | .5444~.5486 | .5368~.5402 | .5367~.5403 | .5505~.5537 | .5511~.5543 | .5448~.5486 | .5451~.5483 | .6332~.6442 | .6371~.6469 | .5890~.6010 | .5897~.6003 |
| | 720 | .6581~.6643 | .6570~.6622 | .5936~.5980 | .5612~.5664 | .6805~.6915 | .6762~.6882 | .6062~.6164 | .5760~.5884 | .6712~.6928 | .6654~.6832 | .6003~.6211 | .6000~.6214 | .6513~.6747 | .6322~.6540 | .5865~.6073 | .5859~.6079 | .6678~.7026 | .6665~.7027 | .6128~.6390 | .6099~.6419 | .8006~.8278 | .8223~.8527 | .6822~.7180 | .6836~.7166 |
| ETTh2 | 96 | .2546~.2552 | .2548~.2554 | .2502~.2506 | .2490~.2496 | .2347~.2355 | .2358~.2366 | .2345~.2353 | .2339~.2347 | .2298~.2308 | .2300~.2312 | .2295~.2305 | .2275~.2287 | .2270~.2300 | .2273~.2303 | .2217~.2247 | .2248~.2282 | .2363~.2371 | .2360~.2368 | .2361~.2373 | .2344~.2356 | .2549~.2553 | .2550~.2554 | .2490~.2494 | .2484~.2486 |
| | 192 | .2997~.3023 | .2990~.3022 | .2970~.2996 | .2934~.2960 | .2734~.2782 | .2747~.2799 | .2634~.2696 | .2579~.2637 | .2828~.2856 | .2864~.2888 | .2813~.2841 | .2807~.2831 | .2801~.2847 | .2827~.2869 | .2775~.2817 | .2769~.2813 | .2809~.2839 | .2812~.2852 | .2798~.2834 | .2956~.2988 | .3167~.3191 | .3249~.3267 | .3039~.3059 | .3006~.3028 |
| | 336 | .3320~.3384 | .3311~.3385 | .3215~.3267 | .3308~.3370 | .3091~.3159 | .3098~.3166 | .2940~.3002 | .2949~.3007 | .3116~.3254 | .3124~.3244 | .2985~.3115 | .3021~.3145 | .3170~.3194 | .3179~.3199 | .2990~.3016 | .3032~.3054 | .3201~.3257 | .3207~.3259 | .3008~.3054 | .3129~.3177 | .3448~.3516 | .3457~.3537 | .3202~.3280 | .3274~.3346 |
| | 720 | .3995~.4051 | .4024~.4062 | .3460~.3514 | .3567~.3615 | .3956~.4054 | .3962~.4062 | .3177~.3289 | .3380~.3476 | .3842~.3904 | .3824~.3882 | .3030~.3094 | .3448~.3506 | .3841~.3975 | .3830~.3938 | .3114~.3240 | .3390~.3516 | .3811~.3903 | .3817~.3903 | .3124~.3214 | .3356~.3442 | .4435~.4513 | .4437~.4509 | .3616~.3684 | .3844~.3926 |
| ETTm1 | 96 | .3556~.3560 | .3568~.3572 | .3445~.3449 | .3453~.3457 | .3883~.3905 | .3927~.3947 | .3616~.3634 | .3616~.3636 | .3496~.3498 | .3523~.3525 | .3455~.3457 | .3455~.3457 | .3505~.3507 | .3535~.3537 | .3453~.3455 | .3453~.3455 | .3581~.3583 | .3582~.3584 | .3519~.3521 | .3524~.3526 | .3945~.3957 | .3946~.3956 | .3832~.3842 | .3826~.3836 |
| | 192 | .4139~.4153 | .4136~.4148 | .4118~.4130 | .4134~.4146 | .4351~.4393 | .4393~.4433 | .4234~.4266 | .4241~.4275 | .4153~.4179 | .4164~.4192 | .4103~.4123 | .4211~.4233 | .4159~.4161 | .4183~.4185 | .4114~.4116 | .4218~.4220 | .4210~.4214 | .4196~.4200 | .4147~.4153 | .4210~.4214 | .4559~.4573 | .4567~.4581 | .4470~.4482 | .4508~.4520 |
| | 336 | .4715~.4793 | .4717~.4785 | .4522~.4616 | .4603~.4683 | .4901~.4909 | .4942~.4950 | .4564~.4572 | .4693~.4701 | .4792~.4806 | .4794~.4812 | .4745~.4761 | .4830~.4866 | .4773~.4801 | .4780~.4804 | .4735~.4761 | .4838~.4860 | .4791~.4863 | .4754~.4824 | .4631~.4691 | .4744~.4806 | .5099~.5117 | .5073~.5091 | .4823~.4841 | .5045~.5063 |
| | 720 | .5460~.5664 | .5426~.5680 | .4633~.4913 | .4977~.5227 | .5416~.5438 | .5452~.5472 | .4672~.4690 | .4871~.4893 | .5415~.5561 | .5459~.5605 | .4692~.4856 | .4923~.5059 | .5424~.5532 | .5464~.5580 | .4699~.4827 | .4944~.5070 | .5442~.5530 | .5442~.5510 | .4669~.4767 | .4941~.5023 | .5686~.5826 | .5709~.5847 | .4945~.5113 | .5152~.5298 |
| ETTm2 | 96 | .1630~.1638 | .1633~.1641 | .1623~.1631 | .1628~.1636 | .1577~.1585 | .1579~.1587 | .1555~.1561 | .1559~.1565 | .1578~.1590 | .1578~.1590 | .1577~.1589 | .1577~.1589 | .1588~.1592 | .1587~.1591 | .1580~.1584 | .1580~.1584 | .1571~.1573 | .1571~.1573 | .1567~.1569 | .1568~.1570 | .1710~.1712 | .1729~.1731 | .1701~.1703 | .1703~.1705 |
| | 192 | .2177~.2189 | .2167~.2179 | .2165~.2177 | .2167~.2179 | .2026~.2046 | .2027~.2047 | .1998~.2016 | .2013~.2031 | .1909~.1917 | .1909~.1917 | .1901~.1907 | .1939~.1947 | .1908~.1934 | .1905~.1933 | .1895~.1917 | .1948~.1974 | .1905~.1913 | .1904~.1912 | .1901~.1909 | .1930~.1938 | .2101~.2103 | .2125~.2127 | .2100~.2104 | .2123~.2125 |
| | 336 | .2616~.2644 | .2576~.2608 | .2420~.2450 | .2521~.2549 | .2447~.2455 | .2448~.2456 | .2255~.2261 | .2348~.2356 | .2277~.2301 | .2281~.2303 | .2070~.2096 | .2230~.2254 | .2286~.2312 | .2287~.2317 | .2118~.2144 | .2213~.2239 | .2276~.2300 | .2279~.2299 | .2087~.2109 | .2200~.2222 | .2510~.2510 | .2511~.2529 | .2328~.2346 | .2543~.2559 |
| | 720 | .3285~.3325 | .3312~.3352 | .2453~.2501 | .2585~.2627 | .3261~.3275 | .3250~.3262 | .2439~.2453 | .2639~.2651 | .2914~.3022 | .2919~.3007 | .2161~.2269 | .2260~.2372 | .2897~.3075 | .2844~.3098 | .2069~.2273 | .2239~.2459 | .2882~.2950 | .2900~.2952 | .2129~.2187 | .2283~.2345 | .3166~.3274 | .3071~.3191 | .2417~.2537 | .2590~.2696 |
| Exchange Rate | 96 | .0873~.0879 | .0883~.0887 | .0816~.0820 | .0835~.0839 | .0835~.0851 | .0828~.0846 | .0756~.0774 | .0779~.0797 | .0881~.0889 | .0874~.0882 | .0809~.0815 | .0831~.0837 | .0789~.0795 | .0795~.0801 | .0753~.0759 | .0770~.0776 | .0793~.0805 | .0797~.0809 | .0738~.0750 | .0761~.0771 | .1071~.1103 | .1130~.1162 | .0936~.0974 | .0991~.1025 |
| | 192 | .1645~.1727 | .1706~.1774 | .1363~.1443 | .1440~.1518 | .1711~.1899 | .1742~.1922 | .1394~.1534 | .1487~.1653 | .1708~.1812 | .1677~.1783 | .1408~.1510 | .1464~.1574 | .1604~.1712 | .1601~.1705 | .1344~.1442 | .1403~.1511 | .1602~.1702 | .1594~.1702 | .1301~.1431 | .1436~.1562 | .2094~.2302 | .1925~.2073 | .1586~.1740 | .1633~.1811 |
| | 336 | .2995~.3163 | .3029~.3165 | .2008~.2170 | .2546~.2702 | .3186~.3364 | .3225~.3375 | .1886~.2080 | .2355~.2535 | .2904~.2978 | .2891~.2949 | .2011~.2067 | .2448~.2512 | .2787~.2967 | .2821~.2975 | .1935~.2105 | .2244~.2402 | .2813~.3047 | .2833~.3013 | .1936~.2170 | .2351~.2571 | .2880~.3214 | .2959~.3241 | .1980~.2258 | .2502~.2818 |
| | 720 | .8077~.8567 | .7758~.8250 | .3085~.3757 | .4155~.4765 | .8348~.8970 | .8317~.8969 | .3238~.3848 | .4312~.5012 | .8427~.9097 | .8468~.9094 | .3123~.3865 | .4145~.4817 | .7882~.8394 | .7876~.8422 | .3131~.3757 | .4242~.4840 | .7895~.8651 | .7719~.8415 | .3004~.3700 | .4120~.4796 | .6813~.7569 | .7434~.8176 | .3417~.4325 | .4415~.5215 |
| Weather | 96 | .1663~.1665 | .1673~.1675 | .1596~.1598 | .1616~.1618 | .1721~.1727 | .1740~.1746 | .1621~.1627 | .1631~.1637 | .1792~.1800 | .1821~.1825 | .1771~.1775 | .1790~.1796 | .1803~.1811 | .1792~.1798 | .1768~.1776 | .1800~.1806 | .1755~.1763 | .1761~.1769 | .1719~.1729 | .1733~.1741 | .1852~.1854 | .1969~.1971 | .1635~.1637 | .1650~.1652 |
| | 192 | .2078~.2124 | .2107~.2149 | .2045~.2089 | .2067~.2109 | .2140~.2154 | .2159~.2175 | .1998~.2014 | .2101~.2115 | .2190~.2298 | .2194~.2314 | .2157~.2275 | .2162~.2272 | .2199~.2289 | .2223~.2325 | .2177~.2269 | .2190~.2284 | .2132~.2198 | .2160~.2224 | .2106~.2164 | .2156~.2222 | .2126~.2196 | .2231~.2299 | .2050~.2114 | .2089~.2151 |
| | 336 | .2595~.2633 | .2641~.2689 | .2482~.2524 | .2494~.2536 | .2594~.2738 | .2647~.2755 | .2382~.2520 | .2428~.2626 | .2682~.2736 | .2719~.2761 | .2544~.2590 | .2601~.2651 | .2703~.2725 | .2737~.2759 | .2540~.2562 | .2630~.2654 | .2618~.2688 | .2632~.2730 | .2526~.2596 | .2545~.2629 | .2717~.2775 | .2757~.2819 | .2703~.2755 | .2839~.2897 |
| | 720 | .3404~.3512 | .3405~.3513 | .2415~.2545 | .2673~.2787 | .3307~.3459 | .3369~.3515 | .2496~.2684 | .2626~.2800 | .3477~.3523 | .3471~.3523 | .2560~.2602 | .2684~.2732 | .3437~.3495 | .3462~.3524 | .2540~.2618 | .2674~.2742 | .3479~.3501 | .3478~.3498 | .2563~.2583 | .2680~.2704 | .3508~.3638 | .3602~.3760 | .2492~.2672 | .2777~.2933 |

**Writing Stage**: Language editing of author-drafted text for clarity and conciseness.
**Human Oversight**: All outputs reviewed/edited by the authors; authors accept full responsibility for the content.

