# OpenReview forum: "COSA: Context-aware Output-Space Adapter for Test-Time Adaptation in Time Series Forecasting"
_ICLR.cc/2026/Conference — ICLR 2026 Poster_

### Official Review · Reviewer_jjeA · 2025-10-28

**Soundness:** 3
**Presentation:** 2
**Contribution:** 2
**Rating:** 6
**Confidence:** 3

**Summary:**

This paper introduces COSA (Context aware Output-Space Adapter), an adapter-based model that adjust predictions of a frozen base model. COSA achieves this via by learning a gated linear transformation on the original prediction and a context vector of summary statistics of the input. The model was tested on standard time series forecasting benchmarks and showed better performance with faster inference time. In addition, authors showed the utility of different components via an ablation study.

**Strengths:**

- Although the idea is simple and one may argue that it is not novel per se, I think the empirical results show strong adequate performance gain in a much shorter inference time compared to the other baselines.

- I think authors have done a good job in finding relevant (albeit recent) works.

- Authors have reported hyper-parameters and also provided code.

The current results support the claims in the paper and look convincing to me and overall I am in favour of acceptance.

**Weaknesses:**

- The input of the gating mechanism and its transformation by parameters g is a bit unclear to me in the formulation and figures. I think authors should elaborate this part similar to other components.

- Authors conducted experiments on the standard time series benchmarks. I think the message of the paper will be stronger if also test the method on the Monash benchmark [1]. Just to be clear, I do not expect authors to repeat all their experiments on this benchmark.

- The benefit of the proposed method is only visible in long-term forecasting which one may argue it is expected as in the current setup (including baselines) there is a delay period to observe some ground truth. I think the paper will be stronger if authors highlight the applicability domain of the work i.e., what problems COSA-F (-P) is the most suitable for?

- Standard error or confidence interval for numerical results and error bars for the bar plots are not reported. Also, it would be nice to have average+se to summarize each method/dataset.

- The baseline normalization is not clearly defined in Section 4.2.2, overall this section seems a bit disconnected to the rest of the paper.

- Looking at the reported results in TAFAS and PETSA paper, there are significant differences with those reported in this paper (Exchange, ETTm2, ETTh2 on iTransformer).

[1] Godahewa et al. 20221 https://arxiv.org/abs/2105.06643

**Questions:**

Can you please comment on the performance of COSA-F (-P) with only a single update (S==1) either verbally and/or empirically?

---

> ### Author Response · Authors · 2025-11-21
>
> We appreciate Reviewer jjeA for their thoughtful and detailed feedbacks. Below we address each weakness and question in turn, and have updated the manuscript and appendix accordingly. Added or revised content is marked in **blue**.
>
> ---
>
> ### Weakness #1
>
> The input of the gating mechanism and its transformation by parameters g is a bit unclear to me in the formulation and figures. I think authors should elaborate this part similar to other components.
>
> ### Response #1
>
> We apologize for the lack of clarity and have revised the paper to better explain the gating mechanism (**Appendix D.1**).
>
> In COSA, the gate is defined as  $gating = \tanh(g) \in [-1, 1]$ with a learnable parameter $g$ and this bounded scalar is multiplied with the linear residual output to modulate the correction strength. If we directly used *g* instead of  $tanh(g)$, small changes in $g$ could cause large, unstable changes in the correction and make the adapter overly sensitive to noisy points. The $tanh$ transform instead keeps the gate bounded and ensures that changes in $g$ are reflected smoothly and gradually. When the residual magnitude spikes, the gate moves toward 0, as shown in **Fig 5 in Appendix D.1** thereby attenuating the residual correction and stabilizing adaptation.
>
> ---
>
> ### Weakness #2
>
> Authors conducted experiments on the standard time series benchmarks. I think the message of the paper will be stronger if also test the method on the Monash benchmark [1]. Just to be clear, I do not expect authors to repeat all their experiments on this benchmark.
>
> ### Response #2
>
> To address this suggestion, we additionally evaluate COSA on the **Electricity** dataset from the Monash benchmark, which has the lowest $R^2$ among the considered datasets and is a large-scale, real-world load-monitoring series with on average 33.77× more variates than our main benchmarks. As summarized in the newly added results table, COSA consistently achieves the highest predictive accuracy across all settings, demonstrating that it remains effective even on highly nonlinear, high-dimensional time series (**Table 6 in Appendix F.1**).
>
> |  | No TTA | TAFAS | PETSA | COSA-F | COSA-P |
> | --- | --- | --- | --- | --- | --- |
> | Avg.MSE | 0.2121 | 0.2031 | 0.2023 | 0.2004 | **0.1998** |
>
> ---
>
> ### Weakness #3
>
> The benefit of the proposed method is only visible in long-term forecasting which one may argue it is expected as in the current setup (including baselines) there is a delay period to observe some ground truth. I think the paper will be stronger if authors highlight the applicability domain of the work i.e., what problems COSA-F (-P) is the most suitable for?
>
> ### Response #3
>
> 1. Although our main setup focuses on long-horizon forecasting, COSA is not specialized only for long-term horizons. To verify this, we conducted additional experiments on shorter forecasting windows and observed that, unlike other Timse-series Forecasting (TSF)-TTA methods, COSA provides consistent and stable improvements even in short-term forecasting (**Table 7 in Appendix F.2**).
>
> |  | No TTA | TAFAS | PETSA | COSA-F | COSA-P |
> | --- | --- | --- | --- | --- | --- |
> | 96 → 24 | 0.1446 | 0.1552 | 0.1551 | **0.1417** | 0.1424 |
> | 96 → 48 | 0.1779 | 0.1918 | 0.1907 | **0.1744** | 0.1746 |
> 1. COSA-P is particularly suitable when the input window exhibits clear and stable periodicity, allowing reliable estimation of the dominant frequency; in such cases (e.g., ETTh1/2), COSA-P consistently outperforms COSA-F. In contrast, when periodicity is not clearly visible within the window or drift occurs frequently, PAAS tends to select large batch sizes and COSA-F yields more stable performance. We clarify these applicability regimes of COSA-F and COSA-P in **Appendix C** and **Appendix G.1**.

---

> ### Author Response · Authors · 2025-11-21
>
> ### Weakness #4
>
> Standard error or confidence interval for numerical results and error bars for the bar plots are not reported. Also, it would be nice to have average+se to summarize each method/dataset.
>
> ### Response #4
>
> Most of our numerical results focus on prediction accuracy (MSE), whose standard deviations across runs are all below 0.001. For readability, we therefore omit “±” and explicitly note this in the main results. For quantities that exhibit more noticeable variation, such as adaptation time per batch and inference time, we additionally report dispersion (mean ± standard deviation).
>
> ---
>
> ### Weakness #5
>
> The baseline normalization is not clearly defined in Section 4.2.2, overall this section seems a bit disconnected to the rest of the paper.
>
> ### Response #5
>
> We added a brief overview of the normalization methods and representative approaches RevIN and DDN in **Appendix G.7**, and clarified the purpose of the comparison in **Section 4.2.2**. This experiment is intended to examine whether COSA acts as a substitute or a complement to existing normalization schemes. Empirically, COSA alone provides more consistent gains than RevIN/DDN, and combining COSA with either normalization yields further improvements. These results indicate that COSA is independent of a particular normalization choice and can be applied compatibly on top of existing normalization methods.
>
> ---
>
> ### Weakness #6
>
> Looking at the reported results in TAFAS and PETSA paper, there are significant differences with those reported in this paper (Exchange, ETTm2, ETTh2 on iTransformer).
>
> ### Response #6
>
> When comparing iTransformer results to those reported in PETSA, we observe noticeable differences in MAE for several cases (ETTh2–720, Exchange Rate–96/720), while our numbers are close to those reported in TAFAS. For ETTh1 and Weather, the reported results are largely consistent across works, suggesting that the discrepancies stem from unreported hyperparameter or seed choices that affect the base model quality in PETSA for specific settings. Importantly, the relative improvement of COSA over its baselines remains consistent across all datasets. Our baselines are reproduced from the officially released code of each method, and to ensure reproducibility, we provide all scripts used in the paper at the following anonymous repository: https://anonymous.4open.science/r/linear-adapter-A720
> .
>
> ---
>
> ### Question #1
>
> Can you please comment on the performance of COSA-F (-P) with only a single update (S==1) either verbally and/or empirically?
>
> ### Response #7
>
> The proposed CALR scheduler resets the learning rate at the beginning of each new batch and then adapts it step-wise after the first update within that batch. Therefore, when $S = 1$, CALR is effectively inactive, which is why we initially omitted this setting. We have now added the $S = 1$ results to **Figure 4(a)**. As expected, **S = 1** yields slightly lower accuracy than larger **$S$** values, but achieves the fastest wall-clock time among all settings.

---

> > ### Comment · Reviewer_jjeA · 2025-11-25
> > **Thank you**
> >
> > Thank you for the rebuttal. You managed to address most of my concerns the only issue remained open in my opinion is Weakness #4 as standard deviation is not a measure to quantify uncertainty. For that, authors need to look into other measures like standard error or confidence interval.

---

> > > ### Author Response · Authors · 2025-11-26
> > >
> > > We appreciate your recognition of our rebuttal and are grateful for the opportunity to further address the remaining concern regarding the choice of uncertainty measures. We additionally provide 95% confidence intervals for the main accuracy comparison results in the appendix. We believe this clarifies the distinction between variability and uncertainty and addresses the concern raised in Weakness #4. (**Table 19 in Appendix I**)

---

> > > > ### Comment · Reviewer_jjeA · 2025-11-26
> > > >
> > > > Thank you for adding the confidence interval. I remain in favour of acceptance and has updated my score.

---

> > > > > ### Author Response · Authors · 2025-11-28
> > > > >
> > > > > We sincerely thank you for your positive feedback and for your continued support of our work.

---

### Official Review · Reviewer_htEG · 2025-10-28

**Soundness:** 3
**Presentation:** 2
**Contribution:** 2
**Rating:** 4
**Confidence:** 4

**Summary:**

This paper, titled “COSA: Context-Aware Output-Space Adapter for Test-Time Adaptation in Time Series Forecasting”, addresses the problem of distribution shifts and non-stationarity that cause performance degradation in deployed time-series forecasting models. The main objective is to develop a lightweight, architecture-agnostic test-time adaptation (TTA) mechanism that can improve model performance post-deployment without retraining the base model.

The proposed method, COSA, is a context-aware output-space adapter that performs direct residual correction on the model’s predictions. It introduces a linear correction layer modulated by a learnable gating mechanism and informed by a context vector summarizing recent ground-truth statistics. During deployment, the frozen base model’s predictions are corrected in real-time, and only the adapter parameters (weights, bias, and gate) are updated using an adaptive learning rate schedule under a leakage-free protocol.

**Strengths:**

- The paper tackles an important and timely problem—test-time adaptation for time-series forecasting—which has received comparatively less attention than its vision counterparts.
- The proposed output-space-only adapter (COSA) is a conceptually clean and lightweight solution, addressing the design complexity and interpretability challenges of prior dual-adapter approaches.
- The idea of context-aware residual correction, where recent ground-truth statistics inform a gating-based correction, is intuitive. It provides a plausible mechanism for leveraging the sequential observability unique to forecasting.

**Weaknesses:**

- Although the paper proposes a simplified, output-space-only adapter, the underlying idea of residual correction or adapter-based adaptation is not fundamentally new. Similar concepts—such as residual adapters, calibration layers, and lightweight correction modules—have appeared in various works.

- While the simplicity of a linear output-space adapter is appealing, it may also limit the method’s expressive power, particularly under nonlinear or abrupt regime shifts.

- The context vector construction relies on fixed aggregation (mean or median) and fixed length K, which might not generalize across datasets with different periodicities or nonstationary characteristics.

- The paper positions the cosine–adaptive learning rate (CALR) as theoretically motivated but does not provide formal derivation or convergence analysis.

**Questions:**

- The context vector $C_t$ is built from recent batch statistics (e.g., means) with fixed length $K$. Could a learned context encoder (e.g., lightweight RNN/SSM/attention) or an adaptive $K$ selected online (error-aware rather than rule-based) improve robustness across different periodicities and drift patterns without eroding the latency advantages? Please discuss the accuracy–overhead trade-off.

- COSA applies a learnable gate $g$ with $\tanh(g)$ to modulate correction. Do you observe interpretable gate trajectories during level shifts, scale drift, and seasonal phase changes (e.g., stronger gating when residuals spike)? Would horizon-specific or channel-specific gating improve control or stability relative to a single scalar gate?

- Experiments decompose multivariate data into per-variable univariate tasks. In settings with strong cross-variable dependence, would COSA benefit from structured $W$ (e.g., block-sparse or low-rank shared components) or shared context to capture cross-covariances? What happens to parameter count, memory, and latency as dimensionality grows?

- You describe a cosine–adaptive learning-rate schedule with thresholds motivated by stability. Can you provide formal reasoning or empirical evidence for convergence/safety (e.g., avoidance of error amplification) under short adaptation steps $S$, and comparisons against simpler schedules (fixed/one-cycle/standard AdamW) with statistical significance?

- In cases where short-term perturbations revert (e.g., transient scale spikes), how quickly does COSA roll back adaptation?

---

> ### Author Response · Authors · 2025-11-21
>
> We appreciate Reviewer htEG for their detailed, constructive feedbacks. Below we address each weakness and question in turn, and have updated the manuscript and appendix accordingly. Added or revised content is marked in **blue**.
>
> ---
>
> ### Weakness #1
>
> Although the paper proposes a simplified, output-space-only adapter, the underlying idea of residual correction or adapter-based adaptation is not fundamentally new. Similar concepts—such as residual adapters, calibration layers, and lightweight correction modules—have appeared in various works.
>
> ### Response #1
>
> We agree that the general ideas of residual adapters and lightweight correction modules have appeared in other domains. **However,** these works do not address time-series test-time adaptation (TSF-TTA) under non-stationary, continuously drifting distributions. TSF-TTA is particularly challenging because (i) even small errors can be strongly amplified along the horizon due to autocorrelation, and (ii) the type and magnitude of non-stationarity can change at every input, requiring continuous re-tuning of correction strength. Our contribution is to show that a simple output-space linear residual adapter can effectively address these challenges and yields stable, efficient gains across diverse non-stationary scenarios when combined with TSF-TTA–specific design of COSA (CALR, context modeling, and gating), as validated by extensive experiments and ablations.
>
> ---
>
> ### Weakness #2
>
> While the simplicity of a linear output-space adapter is appealing, it may also limit the method’s expressive power, particularly under nonlinear or abrupt regime shifts.
>
> ### Response #2
>
> We conducted additional experiments on the Electricity dataset, which is a large-scale, real-world load monitoring benchmark with on average 33.77× more variates than our main datasets and exhibits **strong nonlinearity** (Avg. $R^2 = 0.0253$). Even under this highly nonlinear and high-dimensional regime, COSA consistently outperforms all baselines, indicating that a linear output-space adapter is still sufficiently expressive for challenging non-stationary scenarios considered in this work **(Table 6 in Appendix F.1)**.
>
> |  | No TTA | TAFAS | PETSA | COSA-F | COSA-P |
> | --- | --- | --- | --- | --- | --- |
> | Avg.MSE | 0.2121 | 0.2031 | 0.2023 | 0.2004 | **0.1998** |
>
> ---
>
> ### Weakness #3, Question #1
>
> - The context vector construction relies on fixed aggregation (mean or median) and fixed length K, which might not generalize across datasets with different periodicities or nonstationary characteristics.
> - The context vector is built from recent batch statistics (e.g., means) with fixed length . Could a learned context encoder (e.g., lightweight RNN/SSM/attention) or an adaptive selected online (error-aware rather than rule-based) improve robustness across different periodicities and drift patterns without eroding the latency advantages? Please discuss the accuracy–overhead trade-off.
>
> ### Response #3
>
> COSA currently uses a very lightweight context constructed by aggregating the last $K$ batch means, which minimizes latency and implementation complexity.
>
> To assess sensitivity and the accuracy–overhead trade-off, we implemented two alternatives:
>
> 1. **Selective context:** We implemented a variant that construct context by selecting top-K context values with importance scores. While this selective variant yields higher accuracy in some cases, the original local-context design performs better overall (**Table 12 in Appendix G.1.2**).
> 2. **Encoder-based context:** We additionally extract temporal embeddings from observed ground truth via a learned encoder (including an attention encoder) and use these as context. This variant again shows some case-wise gains but underperforms our simple design on average, with substantial overhead. In non-stationary TSF-TTA, distributions change quickly and adaptation horizons are short, making it hard to reliably train rich representations. We are assuming that encoder-based context can overfit particular patterns or fail to track drift (**Table 13, Appendix G.1.3**).
>
> Overall, these results indicate that COSA is somewhat sensitive to context construction, but the proposed statistic-based context offers the best robustness–latency balance in our TSF-TTA setting. Selective and encoder-based variants are promising for specific regimes, and we highlight them as directions for future work in **Appendix H**.
>
> |  | Proposed | Selective | RNN | LSTM | Attention |
> | --- | --- | --- | --- | --- | --- |
> | Avg. MSE | 0.3240 | 0.3270 | 0.3254 | 0.3260 | 0.3278 |
> | Avg. Adaptation Time / Batch (ms) | 80.12 ± 13.58 | 83.64 ± 15.71 | 126.52 ± 22.88 | 134.29 ± 25.67 | 175.72 ± 16.88 |
> | Avg. Inference Time (ms) | 1.25 ± 0.0984 | 1.26 ± 0.1039 | 1.67 ± 0.2315 | 1.74 ± 0.2778 | 1.96 ± 0.3183 |
> | Avg. Parameter | 1,211,287 | 1,212,217 | 1,317,239 | 1,466,615 | 1,647,735 |

---

> ### Author Response · Authors · 2025-11-21
>
> ### Weakness #4
>
> The paper positions the cosine–adaptive learning rate (CALR) as theoretically motivated but does not provide formal derivation or convergence analysis.
>
> ### Response #4
>
> We agree that classical convergence analysis is difficult to formalize in non-stationary TSF-TTA, where the data distribution and effective optimum keep shifting over time. Our theoretical focus is therefore on **stability within each short adaptation window**, rather than convergence to a fixed point. In CALR, each step-wise parameter update is **uniformly bounded** due to 1) an upper bound on the learning rate $\eta_{\max}$, 2) gradient clipping, 3) L2 normalization loss term, and the gating factor $gating \in [-1, 1]$. These safeguards structurally prevent uncontrolled error amplification during adaptation. We provide more details in **Appendix A**.
>
> The term “theoretically motivated” in Section 3.5 was intended to emphasize this stability-driven design, not to claim a full convergence proof. To avoid confusion, we have revised the wording to “stability-induced” in the updated manuscript.
>
> ---
>
> ### Question #2
>
> - COSA applies a learnable gate with to modulate correction. Do you observe interpretable gate trajectories during level shifts, scale drift, and seasonal phase changes (e.g., stronger gating when residuals spike)? Would horizon-specific or channel-specific gating improve control or stability relative to a single scalar gate?
>
> ### Response #5
>
> 1. We analyzed batch-wise gate values together with the residual magnitude (squared sum of linear residuals). During level shifts and scale drift where residual spikes occur, the gate rapidly moves toward 0, effectively attenuating correction strength. For example, in the iTransformer–ETTm1–96 setting, a spike around the 70–80th batch is immediately followed by the gate shrinking toward 0, which we interpret as COSA suppressing over-correction **(Figure 5 in Appendix D.1)**.
> 2. We also implemented a horizon-wise vector gating with the same length as the prediction length, aiming for finer modulation when drift affects only specific horizons. However, this variant yielded lower average accuracy than a single scalar gate, likely because the influence of noisy points propagates across all horizon-specific gates over time. In contrast, scalar gating provides a consistent batch-level modulation that limits the impact of such noise. **(Table 18 in Appendix G.6)**
>
> |  | Scalar | Vector |
> | --- | --- | --- |
> | Avg. MSE | **0.3240** | 0.3287 |
> | Avg. Adaptation Time / Batch (ms) | **80.12 ± 13.58** | 96.34±12.48 |
> | Avg. Inference Time (ms) | **1.25 ± 0.0984** | 1.89 ± 0.0745 |
> | Avg. Parameter | **1,211,287** | 1,211,623 |
>
> ---
>
> ### Question #3
>
> Experiments decompose multivariate data into per-variable univariate tasks. In settings with strong cross-variable dependence, would COSA benefit from structured (e.g., block-sparse or low-rank shared components) or shared context to capture cross-covariances? What happens to parameter count, memory, and latency as dimensionality grows?
>
> ### Response #6
>
> For fair comparison with prior TSF-TTA baselines, our main experiments follow the same protocol and decompose multivariate series into per-variable univariate tasks. Following the reviewer’s suggestion, we additionally implemented a multivariate variant that models cross-variable dependence via cross-variable attention and a low-rank shared component, using the number of variates as the context dimension (ETT = 7, Exchange Rate = 8, Weather = 21) and report average overhead across datasets.
>
> This multivariate COSA generally improves accuracy over the univariate version when cross-variable correlations are meaningful, but the Exchange Rate dataset—where inter-variable dependence is weak—still favors the univariate setting, and explicitly modeling cross-covariances can even hurt performance **(Table 17 in Appendix G.5)**. Moreover, the extra cross-variable module substantially increases parameters, memory usage, and latency as dimensionality grows. Further discussion is provided in **Appendix H**
>
> |  | Proposed | Correlation |
> | --- | --- | --- |
> | Avg. MSE | 0.3240 | **0.3071** |
> | Avg. Adaptation Time / Batch (ms) | **80.12 ± 13.58** | 186.28 ± 15.36 |
> | Avg. Inference Time (ms) | **1.25 ± 0.0984** | 6.35 ± 0.2452 |
> | Avg. Parameter | **1,211,287** | 1,214,851 |

---

> ### Author Response · Authors · 2025-11-21
>
> ### Question #4
>
> You describe a cosine–adaptive learning-rate schedule with thresholds motivated by stability. Can you provide formal reasoning or empirical evidence for convergence/safety (e.g., avoidance of error amplification) under short adaptation steps , and comparisons against simpler schedules (fixed/one-cycle/standard AdamW) with statistical significance?
>
> ### Response #7
>
> Following the Reviewer htEG’s suggestion, we conducted additional experiments comparing CALR with other PyTorch schedulers: CosineAnnealingLR, ExponentialLR, ReduceLROnPlateau, StepLR, and a fixed learning rate. We excluded One-Cycle because it requires a pre-defined schedule based on a fixed number of steps (and batch size), which is incompatible with PAAS-based dynamic batch sizes and continuously arriving test-time data in our TTA scenario. Across almost all cases, the proposed CALR achieves the best or second-best performance (**Table 16 in Appendix G.4**).
>
> |  | CALR | Cosine | Exp | Fixed | Plataeu | Step |
> | --- | --- | --- | --- | --- | --- | --- |
> | **Average MSE** | **0.3222** | 0.3380 | 0.3487 | 0.3552 | 0.3418 | 0.3312 |
>
> ---
>
> ### Question #5
>
> In cases where short-term perturbations revert (e.g., transient scale spikes), how quickly does COSA roll back adaptation?
>
> ### Response #8
>
> **Figure 6 in Appendix D.2** visualizes the trajectory of pre-adaptation loss and learning rate for the iTransformer–ETTm1, \(L = 96\) case. When a short-term loss spike occurs, CALR immediately decreases the learning rate to minimize the impact of the perturbation, and once the loss enters a stable decreasing phase, CALR increases the learning rate again to promote rapid re-adaptation. This control mechanism suppresses excessive parameter drift **without requiring any roll-back**, allowing COSA to quickly recover its correction performance after the perturbation. The interaction between learning rate and loss demonstrates that, under non-stationary environments with short-term perturbations, COSA can respond and restore performance in a stable manner.

---

> > ### Comment · Reviewer_htEG · 2025-11-27
> >
> > I would like to thank the authors for their detailed response. Most of my concerns have been addressed, and I believe the additional experiments have further strengthened the validity of the paper's claims. Accordingly, I have raised my initial score.
> >
> > Additionally, I suggest some minor improvements in presentation to enhance readability. For instance, the font size in figures that contain legends is generally too small to be read easily. Additionally, the captions for certain tables (e.g., Table 3) would benefit from more detailed descriptions to help readers understand the context better without referring back to the main text.

---

> > > ### Author Response · Authors · 2025-11-28
> > >
> > > We sincerely appreciate your thoughtful feedback and are glad that our revisions have addressed your concerns. Following your suggestions, we have improved the presentation quality by (1) making caption of Table 3 more detailed to provide better context without requiring reference to the main text, and (2) increasing the font size of axis labels and legends across all figures for improved readability. We believe these changes will significantly enhance the accessibility of our results. Thank you again for your constructive comments and for raising your score.

---

### Official Review · Reviewer_hY4L · 2025-10-31

**Soundness:** 3
**Presentation:** 3
**Contribution:** 3
**Rating:** 8
**Confidence:** 4

**Summary:**

The paper proposes a simple yet effective method to adapt time-series forecasting models under distribution shifts. Instead of dual adapters that modify both inputs and outputs, COSA introduces a single output-space adapter that directly corrects predictions from a frozen base model using a linear residual correction modulated by a learnable gate. The adapter leverages a lightweight context vector summarizing recent ground-truth statistics and updates only its parameters during inference under a leakage-free streaming protocol. COSA achieves consistent accuracy gains across six datasets and multiple model types, while remaining architecture-agnostic and computationally efficient for deployment.

**Strengths:**

The paper presents a clear and elegant solution to the problem of test-time adaptation in time-series forecasting through a single output-space adapter, offering originality by simplifying previous dual-adapter frameworks. The methodology is technically sound, with a well-motivated design and a careful leakage-free adaptation protocol. Empirical results are extensive, covering multiple architectures, horizons, and datasets, demonstrating consistent and significant improvements. The paper is well written, with strong clarity in the problem formulation and experimental design. Its simplicity, effectiveness, and deployment-friendliness make it a valuable contribution with practical and theoretical significance for the TTA and time-series communities.

**Weaknesses:**

The context construction relies on simple aggregation statistics, which may not capture complex temporal dependencies or handle irregular or missing data effectively. The experiments focus mainly on standard LTSF datasets without testing on more challenging non-stationary or real-world drift scenarios. The computational analysis could better clarify the trade-offs between adaptation time and inference latency. Additionally, statistical significance reporting or variance analysis would strengthen the reliability of the reported performance gains.

**Questions:**

1. Can the authors clarify under which conditions an output-only adapter is sufficient, and when input-side calibration would still be necessary for stable adaptation?

2. How sensitive is COSA to the construction of the context vector? Would richer temporal encodings or normalization statistics improve its robustness and generalization?

3. Could the method be extended to jointly handle multivariate forecasting instead of decomposing into independent univariate tasks?

4. How would COSA perform under partial label availability or delayed target feedback, which are common in real-world streaming scenarios?

5. Would the proposed adaptive learning rate schedule still remain stable when applied to models with much longer adaptation windows or non-periodic datasets?

6. Could you fix the code link? It seems to have expired.

---

> ### Author Response · Authors · 2025-11-21
>
> We appreciate Reviewer hY4L for their positive and encouraging evaluation of our work. Below we address each weakness and question in turn, and have updated the manuscript and appendix accordingly. Added or revised content is marked in **blue**.
>
> ---
>
> ### Question #1
>
> Can the authors clarify under which conditions an output-only adapter is sufficient, and when input-side calibration would still be necessary for stable adaptation?
>
> ### **Response #1**
>
> We added experiments that combine COSA with the input-side calibration of TAFAS (GCM). On average, using COSA alone yields better accuracy and lower latency, because GCM suppresses fast or irregular but potentially informative distribution shifts and can therefore hurt adaptation.
>
> Output-only residual correction is sufficient when the input distribution is relatively stable and non-stationarity mainly appears as accumulated bias/scale drift in the outputs. In contrast, on datasets with strong input spikes/noise (ETTh1/h2/m2), input-side GCM smooths the inputs, stabilizes the base forecasts, and further improves COSA (**Table 13 in Appendix G.2**). We provide further discussion in **Appendix H**.
>
> |  | COSA-Only | COSA+GCM |
> | --- | --- | --- |
> | Avg. MSE | 0.3240 | 0.3298 |
> | Avg. Adaptation Time / Batch (ms) | 80.12 ± 13.58 | 86.64±12.93 |
> | Avg. Inference Time (ms) | 1.2500 ± 0.0984 | 1.38 ± 0.0953 |
> | Avg. Parameter | 1,211,287 | 1,276,478 |
>
> ---
>
> ### Question #2
>
> How sensitive is COSA to the construction of the context vector? Would richer temporal encodings or normalization statistics improve its robustness and generalization?
>
> ### Response #2
>
> To assess the sensitivity of COSA to context construction, we extended the analysis of context construction methods by adding encoder-based context vectors that extract temporal embeddings from the observed ground truth, in addition to the original statistic-based designs.
>
> We observed cases where encoder-based context, especially with an attention encoder, improved performance, indicating the potential of richer temporal encodings. However, on average, our original statistic-based context yielded the best overall accuracy. In non-stationary TSF-TTA, the distribution changes rapidly and the number of  adaptation steps is limited, making it difficult to reliably learn high-quality representations. As a result, encoder-based contexts can overfit on specific patterns or fail to track distribution shifts, degrading performance. Implementation details and the results are provided in **Table 12 in Appendix G.1.2, Table 13 in G.1.3** and further discussion in **Appendix H.**
>
> |  | Proposed | Rnn | LSTM | Attention |
> | --- | --- | --- | --- | --- |
> | Avg. MSE | 0.3240 | 0.3254 | 0.3260 | 0.3278 |
> | Avg. Adaptation Time / Batch (ms) | 80.12 ± 13.58 | 126.52 ± 22.88 | 134.29 ± 25.67 | 175.72 ± 16.88 |
> | Avg. Inference Time (ms) | 1.2500 ± 0.0984 | 1.6610 ± 0.2315 | 1.7419 ± 0.2778 | 1.9618 ± 0.3183 |
> | Avg. Parameter | 1,211,287 | 1,317,239 | 1,466,615 | 1,647,735 |

---

> ### Author Response · Authors · 2025-11-21
>
> ---
>
> ### Question #3
>
> Could the method be extended to jointly handle multivariate forecasting instead of decomposing into independent univariate tasks?
>
> ### Response #3
>
> COSA can be applied in a model-agnostic manner, as it fundamentally adopts an output-space residual adapter architecture. For fair comparison with prior TSF-TTA baselines (which all decompose multivariate series into per-variable univariate tasks), our main experiments adopt the univariate setting.
>
> Following the reviewer’s suggestion, we additionally implement a multivariate variant that uses cross-variable attention to exchange context across variables and mix shared and variable-specific patterns before applying COSA. This multivariate COSA generally outperforms the univariate version when cross-variable correlations are meaningful, while the Exchange Rate dataset still favors univariate modeling due to its weak inter-variable dependence. The added cross-variable module, however, introduces non-negligible complexity and latency. (**Table 17 in Appendix G.5**)
>
> We provide additional discussion on the future direction in the **Appendix H**.
>
> |  | Proposed (Muiltiple Independent Univariate) | Multivariate |
> | --- | --- | --- |
> | Avg. MSE | 0.3240 | 0.3071 |
> | Avg. Adaptation Time / Batch (ms) | 80.12 ± 13.58 | 186.28 ± 15.36 |
> | Avg. Inference Time (ms) | 1.25 ± 0.0984 | 6.35 ± 0.2452 |
> | Avg. Parameter | 1,211,287 | 1,212,446 |
>
> ---
>
> ### Question #4
>
> How would COSA perform under partial label availability or delayed target feedback, which are common in real-world streaming scenarios?
>
> ### Response #4
>
> Existing TSF-TTA methods, including COSA, assume that ground-truth labels eventually become available, and our experiments follow this setting. In real deployments, however, labels can be delayed or missing for certain time steps. To handle this, COSA updates its adapter parameters **only when** the corresponding ground-truth becomes available, and simply reuses the last updated correction for unlabeled steps, ensuring a leakage-free protocol (**Section 3.2**). Thus, under partial or delayed feedback, only the adaptation frequency changes, while prediction procedure of COSA remains stable and applicable.
>
> ---
>
> ### Question #5
>
> Would the proposed adaptive learning rate schedule still remain stable when applied to models with much longer adaptation windows or non-periodic datasets?
>
> ### Response #5
>
> CALR remains stable under longer adaptation windows because each step-wise update is uniformly bounded and the learning rate within a window is decayed to 0 by a cosine annealing schedule (**Appendix A**). Even if the number of adaptation steps per batch increases, later-step updates have vanishing impact, so the cumulative parameter change in an adaptation window stays bounded.
>
> Moreover, CALR does not rely on periodicity. It adjusts the learning rate solely based on the local loss behavior within each adaptation window. In additional experiments on forecasting horizons as short as $L \in \{24, 48\}$, where periodic patterns are hard to exploit, CALR still provides consistent performance gains across various benchmarks (**Table 7 in the Appendix F.2**).
>
> ---
>
> ### Question #6
>
> Could you fix the code link? It seems to have expired.
>
> ### Response #6
>
> There was temporary issues in Anonymous Github during the review period. We verified that the link is currently operating now.

---

> ### Author Response · Authors · 2025-11-22
>
> ### Weakness
>
> The context construction relies on simple aggregation statistics, which may not capture complex temporal dependencies or handle irregular or missing data effectively. The experiments focus mainly on standard LTSF datasets without testing on more challenging non-stationary or real-world drift scenarios. The computational analysis could better clarify the trade-offs between adaptation time and inference latency. Additionally, statistical significance reporting or variance analysis would strengthen the reliability of the reported performance gains.
>
> ### Response #7
>
> 1. Moivated by Reviewer’s comment, we additionally implement a variant that construct context by selecting top-K context values with importance scores. While this selective variant yields higher accuracy in some cases, the original local-context design performs better overall (**Table 12 in Appendix G.1.2**).
> We further discuss about the results and future directions in **Appendix H**.
>
>
>     |  | Proposed | Selective |
>     | --- | --- | --- |
>     | Avg. MSE | **0.3240** | 0.3270 |
>     | Avg. Adaptation Time / Batch (ms) | **80.12 ± 13.58** | 83.64 ± 15.71 |
>     | Avg. Inference Time (ms) | **1.25 ± 0.0984** | 1.26  ± 0.1039 |
>     | Avg. Parameter | **1,211,287** | 1,317,239 |
> 2. We conducted additional experiments on the Electricity dataset, which is a large-scale, real-world load monitoring benchmark with on average 33.77× more variates than our main datasets and exhibits **strong nonlinearity** (Avg. $R^2 = 0.0253$). COSA consistently outperforms all baselines, indicating that a linear output-space adapter is still sufficiently expressive for challenging non-stationary scenarios considered in this work **(Table 6 in Appendix F.1)**.
>
>
>     |  | No TTA | TAFAS | PETSA | COSA-F | COSA-P |
>     | --- | --- | --- | --- | --- | --- |
>     | Avg.MSE | 0.2121 | 0.2031 | 0.2023 | 0.2004 | **0.1998** |
> 3. We clarify that adaptation time and inference latency, although both affected by the number of adapter parameters, arise from different computation paths and should be treated as separate metrics. Adaptation time is dominated by backward passes and repeated adaptation steps, while inference latency is the cost of a single forward pass after adaptation. Empirically, adaptation time is more sensitive to batch size and the number of adaptation steps, whereas inference latency remains small and almost unchanged relative to the backbone (**Figures 4(a), 4(c), and Table 10 in Appendix F.5**).
> 4. Most of our numerical results focus on prediction accuracy (MSE), whose standard deviations are all below 0.001. For readability, we therefore omit “±” and explicitly note this in the main results. For quantities that exhibit more noticeable variation, such as adaptation time per batch and inference time, we additionally report dispersion (mean ± standard deviation).

---

### Official Review · Reviewer_8Mzj · 2025-11-01

**Soundness:** 2
**Presentation:** 3
**Contribution:** 2
**Rating:** 6
**Confidence:** 4

**Summary:**

This paper introduces COSA (Context-aware Output-Space Adapter), a Test-Time Adaptation (TTA) framework designed to address the performance degradation of time series forecasting models caused by distribution shifts after deployment. In contrast to existing TTA methods that often employ complex dual input-output adapters, COSA proposes a simpler and more direct strategy that operates solely in the output space. The core mechanism utilizes a lightweight context vector, which summarizes statistics from recently observed ground truth, along with the base model's original prediction. This information is fed through a linear layer, modulated by a gating mechanism, to compute a residual correction. During the test phase, only the adapter's parameters are updated, while the backbone model remains frozen. Extensive experiments demonstrate that COSA improves prediction accuracy while reducing inference latency.

---

**Strengths:**

1.  **Simplicity and Novelty:** The "output-space-only" design is a great innovation. It directly corrects predictions, offering a simpler, more intuitive, and safer alternative to complex dual-adapter methods that indirectly modify data distributions.
2.  **Inference Efficiency:** The method is quite fast, achieving a nearly order-of-magnitude speedup in inference time compared to prior TTA methods. This high efficiency makes it well-suited for practical, low-latency deployment scenarios.
3.  **Empirical Performance:** The method's effectiveness is validated by empirical results, where it consistently achieves best performance across six benchmark datasets and multiple backbone architectures, with particularly gains in long-horizon forecasting.
4.  **Comprehensive Validation:** The paper includes a thorough experimental analysis, featuring detailed ablation studies and sensitivity analyses on key hyperparameters. This evaluation validates the method's design choices and strengthens the credibility of the findings.

**Weaknesses:**

1. **Unconventional Experimental Settings:** The experimental setup has some questionable aspects: (1) The lookback window of 96 is relatively short. Most contemporary work uses longer history windows, such as 336 or 512. Predicting a long horizon of 720 from a short 96-step history is a somewhat impractical scenario and may weaken the persuasiveness of the method's effectiveness. (2) For multivariate time series, the mainstream approach (even for channel-independent models) is to process all variates concurrently. Splitting multivariate series into univariate ones for individual forecasting is inefficient, especially for widely-used datasets with a large number of variates like Electricity and Traffic.
2. **Missing Comparison with a Key Baseline and Motivation:** The authors motivate their work by claiming that existing TTA methods for time series forecasting use a dual-adapter architecture, and theirs is novel for being output-space-only. However, a recent paper, [SOLID](https://arxiv.org/abs/2310.14838), is also a TTA algorithm for time series that performs adaptation solely on the output prediction layer and involves an analysis of residuals. This paper is missing experimental comparison and analysis against SOLID, which weakens the novelty claim and leaves a gap in the evaluation.
3. **Potential Risk of Over-Correction and Instability:** The paper mentions that the adaptive learning rate scheduler, CALR, adopts an "aggressive" strategy for fast convergence. While the tanh(g) gating mechanism bounds the correction, an aggressive learning policy, when faced with noisy or anomalous batches, could still lead the adapter to learn unstable or oscillating correction patterns. This over-correction might amplify noise in the short term rather than capturing the true distribution shift, especially with a small batch size B.

**Questions:**

1. **Design of the Context Vector:** The context vector C is constructed by aggregating the K most recent batches, which implicitly assumes temporal locality. However, this assumption may not always hold. For series with strong periodicity, data points from the same phase in previous cycles might be more informative contexts. Conversely, for series with abrupt concept drifts, a context vector based on recent history could provide outdated or even detrimental information. Is a design that relies solely on recent locality optimal, or would it be more robust to consider other relationships such as periodicity?
2. **Clarification on Parameter Efficiency:** Given that COSA has far more parameters than PETSA, could the authors clarify the rationale behind choosing a linear layer with a relatively large parameterization (i.e., the weight W is L x (L+K))? Furthermore, could more parameter-efficient structures, such as low-rank factorization, be explored to potentially achieve similar performance while reducing the parameter count?
3. **Practical Use of Adaptive Batch Size B in COSA-P:** The paper proposes COSA-P, which uses the PAAS strategy to adapt the batch size B, and shows its superiority on the ETTh1 dataset, while COSA-F performs better on others. In a real-world deployment without clear periodic patterns, the PAAS strategy may not be effective. This raises two practical questions: 1) How should we as practitioners choose between COSA-F (fixed B) and COSA-P for a new real-world dataset? 2) For COSA-F, are there any guidelines or heuristics for selecting an optimal fixed value for B?

---

> ### Author Response · Authors · 2025-11-21
>
> We thank Reviewer 8Mzj for the thoughtful feedback. Below we address each weakness and question in turn, and have updated the manuscript and appendix accordingly. Added or revised content is marked in **blue**.
>
> ---
> ### Weakness #1: Unconventional Experimental Settings
>
> Our experimental setting was designed based on the experimental settings of major research on long-term time series forecasting tasks. The referenced studies are as follows:
>
> - Liu, et al. "itransformer: Inverted transformers are effective for time series forecasting." *arXiv* 2023.
> - Wang, et al. "Timexer: Empowering transformers for time series forecasting with exogenous variables." NeruIPS2024.
> - Zhou, et al. "Fedformer: Frequency enhanced decomposed transformer for long-term series forecasting." PMLR, 2022.
> - Zeng, et al. "Are transformers effective for time series forecasting?." AAAI 2023.
> - Kim, et al. "Battling the non-stationarity in time series forecasting via test-time adaptation." AAAI 2025.
>
> Additionally, we conducted longer look-back window experiments (192, 336) on 3 representative base models due to time constraints. COSA still achieving the largest performance improvement (**Table 7 in Appendix F.2**).
>
> |  | No TTA | TAFAS | PETSA | COSA-F | COSA-P |
> | --- | --- | --- | --- | --- | --- |
> | 192 → 192 | 0.2414 | 0.2779 | 0.2757 | **0.2361** | 0.2363 |
> | 192 → 336 | 0.2884 | 0.3404 | 0.3346 | **0.2817** | 0.2823 |
> | 192 → 720 | 0.4183 | 0.4881 | 0.4833 | 0.4072 | **0.4071** |
> | 336 → 720 | 0.3039 | 0.3520 | 0.3423 | **0.2963** | 0.2993 |
> | 336 → 720 | 0.4581 | 0.5081 | 0.5057 | **0.4250** | 0.4310 |
>
> ---
>
> ### Weakness #2: Missing Comparison with SOLID
>
> SOLID fine-tunes the prediction layer of a base forecaster, while other methods keeps the base model frozen. In the revised paper, we clarify this distinction and add a conceptual comparison to COSA. COSA consistently achieves higher accuracy and better efficiency than SOLID (**Table 8 in Appendix F.3**).
>
> |  | **SOLID** | **COSA-F** | **COSA-P** |
> | --- | --- | --- | --- |
> | **Prediction Accuracy(MSE)** | 0.3545 | **0.3100** | 0.3197 |
> | **Wall-Clock Time(Seconds)** | 306.55 ± 2.6183 | 9.73 ± 0.0840 | **7.44 ± 0.0488** |
>
> ---
>
> ### Weakness #3: Potential Risk of Over-Correction and Instability
>
> COSA aggressively adapts to incoming test-time data with only a few adaptation steps to quickly correct prediction errors. To prevent over-correction, we regularize this process with a learnable gating mechanism and gradient clipping. In addition, the proposed CALR scheduler adjusts the learning rate step-wise within each batch: when the loss spikes, CALR temporarily reduces the learning rate to soften updates; when the loss decreases steadily, CALR increases the learning rate to accelerate adaptation. Consequently, even if a transient noisy/anomalous batch causes a sharp loss increase, CALR immediately lowers the learning rate at the next step and rapidly damps the error amplification from that batch (**Appendix A**).
>
> As a result, smaller batch sizes, despite inducing more frequent adaptation steps and thus higher adaptation time, often yield better accuracy due to more fine-grained updates. As shown in **Table 10 in Appendix F.5, and Fig. 4(c)**, even in the most challenging setting with batch size 96, COSA achieves higher predictive accuracy and faster runtime than existing methods.
>
> | Avg. MSE | **No TTA** | **TAFAS** | **PETSA** | **COSA-P** | **COSA-F (B=8)** | **COSA-F (B=16)** | **COSA-F (B=24)** | **COSA-F (B=48)** | **COSA-F (B=96)** |
> | --- | --- | --- | --- | --- | --- | --- | --- | --- | --- |
> | **W=96** | 0.2545 | 0.2471 | 0.2480 | 0.2416 | **0.1908** | 0.2202 | 0.2330 | 0.2412 | 0.2445 |
> | **W=192** | 0.3144 | 0.3038 | 0.3046 | 0.2960 | **0.1900** | 0.2295 | 0.2524 | 0.2942 | 0.2986 |
> | **W=336** | 0.3839 | 0.3653 | 0.3664 | 0.3467 | **0.1837** | 0.2292 | 0.2526 | 0.3364 | 0.3463 |
> | **W=720** | 0.5539 | 0.5226 | 0.5232 | 0.4046 | **0.1811** | 0.2384 | 0.2672 | 0.3788 | 0.3958 |
>
> | Avg. Wall-clock time (Seconds) | **TAFAS** | **PETSA** | **COSA-P** | **COSA-F (B=8)** | **COSA-F (B=16)** | **COSA-F (B=24)** | **COSA-F (B=48)** | **COSA-F (B=96)** |
> | --- | --- | --- | --- | --- | --- | --- | --- | --- |
> | **W=96** | 5.9271 | 7.4129 | 7.0642 | 47.3922 | 25.6158 | 17.2297 | 9.6333 | **5.1258** |
> | **W=192** | 6.0383 | 7.3675 | 7.1419 | 49.4422 | 25.1222 | 18.0969 | 9.4344 | **5.5133** |
> | **W=336** | 6.1554 | 7.6588 | 7.4131 | 49.2211 | 25.1119 | 17.2906 | 9.5644 | **5.8081** |
> | **W=720** | **6.5300** | 7.9867 | 8.1578 | 46.3711 | 24.4683 | 17.6253 | 10.3036 | 6.5397 |

---

> ### Author Response · Authors · 2025-11-21
>
> ### Question #1: Design of the Context Vector
>
> COSA is designed to quickly and stably correct output bias in the local residual correction. In non-stationary TTA settings, the input distribution continuously drifts over time, so relying on distant past information can conflict with the current drift direction and degrade correction performance. For this reason, our original context vector is constructed from recent information only.
>
> Motivated by Reviewer 8Mzj’s question, we additionally implement a variant that construct context by selecting top-K context values with importance scores. While this selective variant yields higher accuracy in some cases, the original local-context design performs better overall (**Table 12 in Appendix G.1.2**).
>
> We further discuss about the results and future directions in **Appendix H**.
>
> |  | Proposed | Selective |
> | --- | --- | --- |
> | Avg. MSE | **0.3240** | 0.3270 |
> | Avg. Adaptation Time / Batch (ms) | **80.12 ± 13.58** | 83.64 ± 15.71 |
> | Avg. Inference Time (ms) | **1.25 ± 0.0984** | 1.26  ± 0.1039 |
> | Avg. Parameter | **1,211,287** | 1,317,239 |
>
> ---
>
> ### Question #2: Parameter Efficiency
>
> To provide sufficient capacity under this constraint, we adopt a full linear layer.
>
> The number parameters of COSA adapter account relatively small compared to the base model. While this is larger than PETSA adapter, it is still smaller than the TAFAS. Despite using more parameters than PETSA, COSA achieves faster inference, more stable behavior, and larger accuracy gains.
>
> Low-rank factorization can, in principle, be applied to COSA as well, but it trades parameter efficiency for reduced expressiveness and a sharper gradient landscape, which can overfit specific drift patterns or destabilize adaptation. As seen in PETSA, such approaches typically require additional architectural changes and auxiliary losses to remain stable. We therefore position COSA as complementary to low-rank designs and highlight parameter-efficient variants of COSA (e.g., via low-rank adapters) as an interesting direction for future work, which we now explicitly discuss in the **Appendix H** of revised paper.
>
> ---
>
> ### Question #3: Practical Use
>
> COSA-P uses a PAAS strategy that estimates the dominant frequency via Fast Fourier Transform and sets the batch size accordingly, enabling cycle-aligned adaptation on strongly periodic series. However, when no clear dominant frequency is detected within the input window, the batch size defaults to the full window length, which can hurt performance on weakly or very long-periodic data. Empirically, for datasets where periodicity is observable within the window (ETTh1/ETTh2), COSA-P achieves a smaller average batch size than COSA-F (B=48) and thus more frequent corrections, whereas on datasets with weak or long-range periodicity (Exchange Rate, Weather), COSA-F outperforms COSA-P (**Table 7 in Appendix F.2**).
>
> For COSA-F, there is an inherent trade-off between adaptation time and efficiency as batch size changes. We therefore recommend choosing the batch size according to the desired balance between accuracy and runtime. In the paper, guided by **Fig. 4(c)** and **Table 10 in Appendix F.5,** we adopt B=48 as a balanced setting that offers strong accuracy with competitive execution time.

---

### Public Comment · ~Luyao_Chen2 · 2025-11-27
**Discrepancies in Baseline Model Results Compared to Original Papers.**

Dear Authors,

I have a question about the results of the baseline methods (like iTransformer and DLinear) reported in your paper, as they seem different from the results in their original papers.

For example, according to Table 10 in the iTransformer paper, its MSE on the ETTm1 dataset for horizons 96, 192, 336, and 720 are 0.334, 0.377, 0.426, and 0.491, respectively. However, the corresponding numbers in your paper are 0.3823, 0.4423, 0.5093, and 0.6065. The difference is quite noticeable.

While one might explain the iTransformer difference by hyperparameter tuning, why are the results for DLinear – a much simpler linear model – also quite different from its original paper? For instance, on ETTh1 for the same horizons, Table 2 of the original DLinear paper reports results of 0.375, 0.405, 0.439, and 0.472. Alternatively, Table 10 in the iTransformer paper lists them as 0.386, 0.437, 0.481, and 0.519. Yet, your paper reports 0.4695, 0.5213, 0.5659, and 0.7117. How can the difference be this large?

Furthermore, for DLinear on ETTh1, even after adding your proposed COSA module, the performance is still worse than the results reported in the original DLinear paper without any adaptation.

My initial guess is that perhaps a different dataset splitting scheme was used. However, I didn't find this mentioned in your paper. Could you please clarify the reason for these discrepancies?

---

> ### Author Response · Authors · 2025-11-27
>
> Thank you for your careful checking and for bringing this to our attention. Additionally, we use a time-ordered 7:1:2 (train:val:test) split for all ETT datasets, and we have explicitly clarified this in Section 4.1. Under this unified split setup, ETTh1/ETTm1 indeed show lower accuracy than in the original papers, whereas ETTh2/ETTm2 actually achieve lower MSE, indicating that the effect is dataset-dependent. Despite these differences in absolute numbers, all TTA methods are evaluated on exactly the same pretrained baselines, so our conclusions about the relative gains of COSA over No TTA and other TTA methods remain valid.

---

### Meta-Review · Area_Chair_hfMT · 2025-12-23

**Summary:**

This paper presents COSA as a test-time adaptation strategy for time series forecasting, which performs residual correction modulated by gating and achieves significant promotion in the latest models. After the rebuttal, all the reviewers gave a positive score.

Since the current design is based on linear projection and a univariate forecasting paradigm, the shared concerns among reviewers are utilizing richer context encoders and extending to a multivariate setting. The authors have made the corresponding modification and include all the new results in Appendix G, which provides a solid and convincing response to the raised concerns.

There are still some remaining concerns, including the possibility of over-correction and the deficiency in output-only adaptation. Although the authors have provided some supporting experiments, I think these limitations should be discussed in the final version.

Considering the detailed rebuttal and the consistent positive judgments from reviewers, I would like to recommend acceptance for this paper.

**Reviewer Concerns:**

As described above, the resolved concerns and their corresponding rebuttal include:

-	Comparison to richer encoders.

-	Extensive to multivariate forecasting.

-	More experiments about new benchmarks (Electricity) and new baselines (SOLID and GCM) and an explanation of the inconsistent experimental setting.

-	Uncertainty analysis.

The following concerns remain:

-	Possibility of over-correction. Although the authors provide new experimental results on the robustness of the model, the discussion about extreme cases (noisy observations, abnormal events) needs further investigation.

-	The deficiency in output-only adaptation. The authors have provided some new experiments in Appendix G.2. Although the explanation is quite intuitive, some quantitative analysis is expected, such as experimenting with various abnormal or noisy degrees.

**Reviewer Scores:**

(1) Reviewer 8Mzj (initial score: 6). Since most of his/her concerns were answered in detail, except that the possibility of over-correctness is not comprehensively discussed, I think he/she will maintain the original positive score.

(2) Reviewer hY4L (initial score: 8). The authors provide a sufficient rebuttal; he/she would keep the original positive rating.

(3) Reviewer htEG (initial score: 4). During the rebuttal stage, this reviewer has already raised the score. The updated score would be 6.

(4) Reviewer jjeA (initial score: 6). The authors have provided the requested experiments and confidence interval. The reviewer mentioned that they have updated the score.

---

### Decision · Program_Chairs · 2026-01-26

Accept (Poster)